# Logical Characterizations of Recurrent Graph Neural Networks with Reals and Floats

**Veeti Ahvonen**[1]**, Damian Heiman**[1]**, Antti Kuusisto**[1]**, Carsten Lutz**[2,3]
[1]Tampere University, [2]Leipzig University, [3]ScaDS.AI, Dresden/Leipzig
[1]`firstname.lastname@tuni.fi`, [2,3]`clu@informatik.uni-leipzig.de`

## Abstract

In pioneering work from 2019, Barceló and coauthors identified logics that precisely match the expressive power of constant iteration-depth graph neural networks (GNNs) relative to properties definable in first-order logic. In this article, we give exact logical characterizations of recurrent GNNs in two scenarios: (1) in the setting with floating-point numbers and (2) with reals. For floats, the formalism matching recurrent GNNs is a rule-based modal logic with counting, while for reals we use a suitable infinitary modal logic, also with counting. These results give exact matches between logics and GNNs in the recurrent setting without relativising to a background logic in either case, but using some natural assumptions about floating-point arithmetic. Applying our characterizations, we also prove that, relative to graph properties definable in monadic second-order logic (MSO), our infinitary and rule-based logics are equally expressive. This implies that recurrent GNNs with reals and floats have the same expressive power over MSO-definable properties and shows that, for such properties, also recurrent GNNs with reals are characterized by a (finitary!) rule-based modal logic. In the general case, in contrast, the expressive power with floats is weaker than with reals. In addition to logic-oriented results, we also characterize recurrent GNNs, with both reals and floats, via distributed automata, drawing links to distributed computing models.

## 1 Introduction

Graph Neural Networks (GNNs) [11, 30, 34] have proven to be highly useful for processing graph data in numerous applications that span a remarkable range of disciplines including bioinformatics [35], recommender systems [33], traffic forecasting [21], and a multitude of others. The success of GNNs in applications has stimulated lively research into their theoretical properties such as expressive power. A landmark result is due to Barceló et al. [5] which was among the first to characterize the expressive power of GNNs in terms of *logic*, see [12, 8, 6, 26] and references therein for related results. More precisely, Barceló et al. show that a basic GNN model with a constant number of iterations has exactly the same expressive power as graded modal logic GML in restriction to properties definable in first-order logic FO.

In this article, we advance the analysis of the expressive power of GNNs in two directions. First, we study the relation between GNN models based on real numbers, as mostly studied in theory, and GNN models based on floating-point numbers, as mostly used in practice. And second, we focus on a family of basic recurrent GNNs while previous research has mainly considered GNNs with a constant number of iterations, with the notable exception of [26]. The GNNs studied in the current paper have a simple and natural termination (or acceptance) condition: termination is signaled via designated feature vectors and thus the GNN "decides" by itself about when to terminate. We remark that some of our results also apply to constant iteration GNNs and to recurrent GNNs with a termination condition based on fixed points, as used in the inaugural work on GNNs [11]; see the conclusion section for further details.

38th Conference on Neural Information Processing Systems (NeurIPS 2024).

We provide three main results. The first one is that recurrent GNNs with floats, or GNN[F]s, have the same expressive power as the *graded modal substitution calculus* GMSC [24, 1, 3]. This is a rule-based modal logic that extends the *modal substitution calculus* MSC [24] with counting modalities. MSC has been shown to precisely correspond to distributed computation models based on automata [24] and Boolean circuits [1]. GMSC is related to the graded modal $\mu$-calculus, but orthogonal in expressive power. The correspondence between GNN[F]s and GMSC is as follows.

**Theorem 3.2.** *The following have the same expressive power:* GNN[F]s, GMSC, *and R-simple aggregate-combine* GNN[F]s.

Here *R-simple aggregate-combine* GNN[F]s mean GNN[F]s that use basic aggregate-combine functions as specified by Barceló et al. [5] and the truncated $\mathrm{ReLU}$ as the non-linearity function, see Section 2.1 for the formalities. The theorem shows that an R-simple model of GNN[F]s suffices, and in fact GNN[F]s with a more complex architecture can be turned into equivalent R-simple ones. We emphasize that the characterization provided by Theorem 3.2 is *absolute*, that is, not relative to a background logic. In contrast, the characterization by Barceló et al. [5] is relative to first-order logic. Our characterization does rely, however, on an assumption about floating-point arithmetics. We believe that this assumption is entirely natural as it reflects practical implementations of floats.

Our second result shows that recurrent GNNs with reals, or GNN[$\mathbb{R}$]s, have the same expressive power as the infinitary modal logic $\omega$-GML that consists of infinite disjunctions of GML-formulas.

**Theorem 3.4.** GNN[$\mathbb{R}$]s have the same expressive power as $\omega$-GML.

Again, this result is absolute. As we assume no restrictions on the arithmetics used in GNN[$\mathbb{R}$]s, they are very powerful: with the infinitary logic $\omega$-GML it is easy to define even undecidable graph properties. We regard the theorem as an interesting theoretical upper bound on the expressivity of GNNs operating in an unrestricted, recurrent message passing scenario with messages flowing to neighbouring nodes. We note that GNN[$\mathbb{R}$]s can easily be shown more expressive than GNN[F]s.

Our third result considers GNN[$\mathbb{R}$]s and GNN[F]s relative to a very expressive background logic, probably the most natural choice in the recurrent GNN context: *monadic second-order logic* MSO.

**Theorem 4.3.** *Let* $\mathcal{P}$ *be a property expressible in* MSO. *Then* $\mathcal{P}$ *is expressible as a* GNN[$\mathbb{R}$] *if and only if it is expressible as a* GNN[F].

This result says that, remarkably, for the very significant and large class of MSO-expressible properties, using actual reals with unrestricted arithmetic gives no more expressive power than using floats, by Theorem 3.2 even in the R-simple aggregate-combine setting. Thus, for this class of properties, the theoretical analyses from the literature do not diverge from the practical implementation! Taken together, the above results also imply that in restriction to MSO-expressible properties, GNN[$\mathbb{R}$]s are equivalent to the (finitary!) graded modal substitution calculus GMSC.

We also develop characterizations of GNNs in terms of distributed automata. These are in fact crucial tools in our proofs, but the characterizations are also interesting in their own right as they build links between GNNs and distributed computing. We study a class of distributed automata called counting message passing automata (CMPAs) that may have a countably infinite number of states. Informally, these distributed automata update the state of each node according to the node's own state and the *multiset* of states received from its out-neighbours. We also study their restriction that admits only a finite number of states (FCMPAs) and, furthermore, bounded FCMPAs. In the bounded case the multiplicities of states in the received multisets are bounded by some constant $k \in \mathbb{N}$. A summary of our main results, both absolute and relative to MSO, is given in Table 1.

Table 1: A summary of our main results. The first row contains the results obtained without relativising to a background logic and the second row contains results relative to MSO. Here $x \equiv y$ means that $x$ and $y$ have the same expressive power while $x < y$ means that $y$ is strictly more expressive than $x$. Further, "bnd." stands for "bounded" and "AC" for "aggregate-combine".

| | |
|---|---|
| Absolute: | GNN[F] $\equiv$ R-simple AC-GNN[F] $\equiv$ GMSC $\equiv$ bnd. FCMPA $<$ GNN[$\mathbb{R}$] $\equiv$ $\omega$-GML $\equiv$ CMPA |
| MSO: | GNN[F] $\equiv$ R-simple AC-GNN[F] $\equiv$ GMSC $\equiv$ bnd. FCMPA $\equiv$ GNN[$\mathbb{R}$] $\equiv$ $\omega$-GML $\equiv$ CMPA |

**Related Work.** Barceló et al. [5] study aggregate-combine GNNs with a constant number of iterations. They characterize these GNNs—in restriction to properties expressible in first-order logic—in

terms of graded modal logic GML and show that a *global readout mechanism* leads to a model equivalent to the two-variable fragment of first-order logic with counting quantifiers $\text{FOC}^2$. Our work extends the former result to include recurrence in a natural way while we leave studying global readouts as future work; see the conclusion section for further details. Grohe [13] connects the *guarded fragment of first-order logic with counting* $\text{GFO} + \text{C}$ and polynomial-size bounded-depth circuits, linking (non-recurrent) GNNs to the circuit complexity class $\text{TC}^0$. Grohe's characterization utilises dyadic rationals rather than floating-point numbers. Benedikt et al. [6] use logics with a generalized form of counting via *Presburger quantifiers* to obtain characterizations for (non-recurrent) GNNs with a bounded activation function. The article also investigates questions of decidability concerning GNNs—a topic we will not study here. As a general remark on related work, it is worth mentioning that the expressive power of (basic) recurrent GNN-models is invariant under the Weisfeiler-Leman test. This link has been recently studied in numerous articles [25, 36, 12, 5].

Pfluger et al. [26] investigate recurrent GNNs with two kinds of termination conditions: one based on reaching a fixed point when iteratively generating feature vectors, and one where termination occurs after a number of rounds determined by the size of the input graph. They concentrate on the case of unrestricted aggregation and combination functions, even including all uncomputable ones. Their main result is relative to a logic LocMMFP introduced specifically for this purpose, extending first-order logic with a least fixed-point operator over unary monotone formulas. The characterization itself is given in terms of the graded two-way $\mu$-calculus. We remark that MSO significantly generalizes LocMMFP and that the graded two-way $\mu$-calculus is incomparable in expressive power to our GMSC. In contrast to our work and to Barcelo et al. [5], Pfluger et al. do not discuss the case where the aggregation and combination functions of the GNNs are R-simple or restricted in any other way. We view our work as complementing yet being in the spirit of both Barceló et al. [5] and Pfluger et al. [26].

GNNs are essentially distributed systems, and logical characterizations for distributed systems have been studied widely. A related research direction begins with Hella et al. [16], Kuusisto [24] and Hella et al. [17] by results linking distributed computation models to modal logics. The articles [16] and [17] give characterizations of constant-iteration scenarios with modal logics, and [24] lifts the approach to recurrent message-passing algorithms via showing that the modal substitution calculus MSC captures the expressivity of finite distributed message passing automata. This generalizes the result from [16] that characterized the closely related class $\text{SB}(1)$ of local distributed algorithms with modal logic. Later Ahvonen et al. [1] showed a match between MSC and circuit-based distributed systems. Building on the work on MSC, Reiter showed in [27] that the $\mu$-fragment of the modal $\mu$-calculus captures the expressivity of finite message passing automata in the asynchronous scenario.

## 2 Preliminaries

We let $\mathbb{N}$, $\mathbb{Z}_+$ and $\mathbb{R}$ denote the sets of non-negative integers, positive integers, and real numbers respectively. For all $n \in \mathbb{Z}_+$, we let $[n] := \{1, \ldots, n\}$ and for all $n \in \mathbb{N}$, we let $[0; n] := \{0, \ldots, n\}$. With $|X|$ we denote the cardinality of the set $X$, with $\mathcal{P}(X)$ the power set of $X$ and with $\mathcal{M}(X)$ the set of multisets over $X$, i.e., the set of functions $X \to \mathbb{N}$. With $\mathcal{M}_k(X)$ we denote the set of $k$-multisets over $X$, i.e., the set of functions $X \to [0; k]$. Given a $k$-multiset $M \in \mathcal{M}_k(X)$ and $x \in X$, intuitively $M(x) = n < k$ means that there are exactly $n$ copies of $x$ and $M(x) = k$ means that there are $k$ *or more* copies of $x$.

We work with node-labeled **directed graphs** (possibly with self-loops), and *simply refer to them as graphs*. Let LAB denote a countably infinite set of **node label symbols**, representing features. We denote finite sets of node label symbols by $\Pi \subseteq \text{LAB}$. Given any $\Pi \subseteq \text{LAB}$, a $\Pi$-**labeled graph** is a triple $G = (V, E, \lambda)$ where $V$ is a set of **nodes**, $E \subseteq V \times V$ is a set of **edges** and $\lambda \colon V \to \mathcal{P}(\Pi)$ is a **node labeling function**. Note that a node can carry multiple label symbols. A **pointed graph** is a pair $(G, v)$ with $v \in V$. Given a graph $(V, E, \lambda)$, the set of **out-neighbours** of $v \in V$ is $\{w \mid (v, w) \in E\}$. Unless stated otherwise, we only consider *finite* graphs, i.e., graphs where the set of nodes is finite. A **node property over** $\Pi$ is a class of pointed $\Pi$-labeled graphs. A **graph property over** $\Pi$ is a class of $\Pi$-labeled graphs. A graph property $\mathcal{G}$ over $\Pi$ corresponds to a node property $\mathcal{N}$ over $\Pi$ if the following holds for all $\Pi$-labeled graphs $G$: $G \in \mathcal{G}$ iff $(G, v) \in \mathcal{N}$ for every node $v$ of $G$. Henceforth a property means a node property. We note that many of our results hold even with infinite graphs and infinite sets of node label symbols. Our results easily extend to graphs that admit labels on both nodes and edges.

## 2.1 Graph neural networks

A graph neural network (GNN) is a neural network architecture for graph-structured data. It may be viewed as a distributed system where the nodes of the (directed, node-labeled) input graph calculate with real numbers and communicate synchronously in discrete rounds. More formally, a **recurrent graph neural network** $\text{GNN}[\mathbb{R}]$ over $(\Pi, d)$, with $\Pi \subseteq \text{LAB}$ and $d \in \mathbb{Z}_+$, is a tuple $\mathcal{G} = (\mathbb{R}^d, \pi, \delta, F)$. A recurrent graph neural network computes in a (node-labeled) directed graph as follows. In any $\Pi$-labeled graph $(V, E, \lambda)$, the **initialization function** $\pi \colon \mathcal{P}(\Pi) \to \mathbb{R}^d$ assigns to each node $v$ an initial **feature vector** or **state** $x_v^0 = \pi(\lambda(v))$.[1] In each subsequent round $t = 1, 2, \ldots$, every node computes a new feature vector $x_v^t$ using a **transition function** $\delta \colon \mathbb{R}^d \times \mathcal{M}(\mathbb{R}^d) \to \mathbb{R}^d, \delta(x, y) = \text{COM}(x, \text{AGG}(y))$, which is a composition of an **aggregation function** $\text{AGG} \colon \mathcal{M}(\mathbb{R}^d) \to \mathbb{R}^d$ (typically sum, min, max or average) and a **combination function** $\text{COM} \colon \mathbb{R}^d \times \mathbb{R}^d \to \mathbb{R}^d$ such that $x_v^t = \text{COM}\left(x_v^{t-1}, \text{AGG}\left(\{\!\{ x_u^{t-1} \mid (v, u) \in E \}\!\}\right)\right)$ where double curly brackets $\{\!\{...\}\!\}$ denote multisets.[2] The recurrent GNN $\mathcal{G}$ **accepts** a pointed $\Pi$-labeled graph $(G, v)$ if $v$ visits (at least once) a state in the set $F \subseteq \mathbb{R}^d$ of **accepting feature vectors**, i.e., $x_v^t \in F$ for some $t \in \mathbb{N}$. When we do not need to specify $d$, we may refer to a $\text{GNN}[\mathbb{R}]$ over $(\Pi, d)$ as a $\text{GNN}[\mathbb{R}]$ over $\Pi$. A **constant-iteration** $\text{GNN}[\mathbb{R}]$ is a pair $(\mathcal{G}, N)$ where $\mathcal{G}$ is a $\text{GNN}[\mathbb{R}]$ and $N \in \mathbb{N}$. It **accepts** a pointed graph $(G, v)$ if $x_v^N \in F$. Informally, we simply run a $\text{GNN}[\mathbb{R}]$ for $N$ iterations and accept (or do not accept) based on the last iteration. We say that $\mathcal{G}$ (resp., $(\mathcal{G}, N)$) **expresses** a node property $\mathcal{P}$ over $\Pi$, if for each pointed $\Pi$-labeled graph $(G, w)$: $(G, w) \in \mathcal{P}$ iff $\mathcal{G}$ (resp., $(\mathcal{G}, N)$) accepts $(G, w)$. A node property $\mathcal{P}$ over $\Pi$ is **expressible** as a $\text{GNN}[\mathbb{R}]$ (resp. as a constant-iteration $\text{GNN}[\mathbb{R}]$) if there is a $\text{GNN}[\mathbb{R}]$ (resp. constant-iteration $\text{GNN}[\mathbb{R}]$) expressing $\mathcal{P}$.

One common, useful and simple possibility for the aggregation and combination functions, which is also used by Barceló et al. (see [5], and also the papers [25, 15]) is defined by $\text{COM}\left(x_v^{t-1}, \text{AGG}\left(\{\!\{ x_u^{t-1} \mid (v, u) \in E \}\!\}\right)\right) = f(x_v^{t-1} \cdot C + \sum_{(v,u) \in E} x_u^{t-1} \cdot A + \mathbf{b})$, where $f \colon \mathbb{R}^d \to \mathbb{R}^d$ is a non-linearity function (such as the truncated ReLU also used by Barceló et al. in [5], defined by $\text{ReLU}^*(x) = \min(\max(0, x), 1)$ and applied separately to each vector element), $C, A \in \mathbb{R}^{d \times d}$ are matrices and $\mathbf{b} \in \mathbb{R}^d$ is a bias vector. We refer to GNNs that use aggregation and combination functions of this form and $\text{ReLU}^*$ as the non-linearity function as **R-simple aggregate-combine** GNNs (here 'R' stands for $\text{ReLU}^*$).

**Example 2.1.** Given $\Pi$ and $p \in \Pi$, *reachability of node label symbol $p$* is the property $\mathcal{P}$ over $\Pi$ that contains those pointed $\Pi$-labeled graphs $(G, v)$ where a path exists from $v$ to some $u$ with $p \in \lambda(u)$. An R-simple aggregate-combine $\text{GNN}[\mathbb{R}]$ over $(\Pi, 1)$ (where $C = A = 1$, $\mathbf{b} = 0$ and 1 is the only accepting feature vector) can express $\mathcal{P}$: In round 0, a node $w$'s state is 1 if $p \in \lambda(w)$ and else 0. In later rounds, $w$'s state is 1 if $w$'s state was 1 or it gets 1 from its out-neighbours; else the state is 0.

**Remark 2.2.** Notice that unrestricted $\text{GNN}[\mathbb{R}]$s can express, even in a single iteration, node properties such as that the number of immediate out-neighbours of a node is a prime number. In fact, for any $U \subseteq \mathbb{N}$, including any undecidable set $U$, a $\text{GNN}[\mathbb{R}]$ can express the property that the number $l$ of immediate out-neighbours is in the set $U$. See [6] for related undecidability results.

Informally, a floating-point system contains a finite set of rational numbers and arithmetic operations $\cdot$ and $+$. Formally, if $p \in \mathbb{Z}_+$, $n \in \mathbb{N}$ and $\beta \in \mathbb{Z}_+ \setminus \{1\}$, then a **floating-point system** is a tuple $S = ((p, n, \beta), +, \cdot)$ that consists of the set $D_S$ of all rationals accurately representable in the form $0.d_1 \cdots d_p \times \beta^e$ or $-0.d_1 \cdots d_p \times \beta^e$ where $0 \le d_i \le \beta - 1$ and $e \in \{-n, \ldots, n\}$. It also consists of arithmetic operations $+$ and $\cdot$ of type $D_S \times D_S \to D_S$. We adopt the common convention where $+$ and $\cdot$ are defined by taking a precise sum/product w.r.t. reals and then rounding to the closest float in $D_S$, with ties rounding to the float with an even least significant digit, e.g., $0.312 + 0.743$ evaluates to $1.06$ if the real sum $1.055$ is not in the float system. Thus, our float systems handle overflow by capping at the maximum value instead of wrapping around. We typically just write $S$ for $D_S$.

Consider GNNs using floats in the place of reals. In GNNs, sum is a common aggregation function (also used in R-simple GNNs), and the sum of floats can depend on the ordering of floats, since it

---

[1]In [5, 26] an initialization function is not explicitly included in GNNs; instead each node is labeled with an initial feature vector in place of node label symbols. However, these two approaches are essentially the same.

[2]In GNNs here, messages flow in the direction opposite to the edges of graphs, i.e., an edge $(v, u) \in E$ means that $v$ receives messages sent by $u$. This is only a convention that could be reversed via using a modal logic that scans the *inverse relation* of $E$ instead of $E$.

is not associative. In real-life implementations, the set $V$ of nodes of the graph studied can typically be associated with an implicit linear order relation $<^V$ (which is not part of the actual graph). It is then natural to count features of out-neighbours in the order $<^V$. However, this allows float GNNs to distinguish isomorphic nodes, which violates the desire that GNNs should be invariant under isomorphism. For example, summing $1$, $-1$ and $0.01$ in two orders in a system where the numbers must be representable in the form $0.d_1 d_2 \times 10^e$ or $-0.d_1 d_2 \times 10^e$: first $(1 + (-1)) + 0.01 = 0 + 0.01 = 0.01$ while $(1 + 0.01) + (-1) = 1 + (-1) = 0$, since $1.01$ is not representable in the system. A float GNN could distinguish two isomorphic nodes with such ordering of out-neighbours.

To ensure isomorphism invariance for GNNs with floats that use sum, it is natural to order the floats instead of the nodes. For example, adding floats in increasing order (of the floats) is a natural and simple choice. Summing in this increasing order is also used widely in applications, being a reasonable choice w.r.t. accuracy (see, e.g., [32],[29], [18]). Generally, floating-point sums in applications have been studied widely, see for example [19]. Summing multisets of floats in increasing order leads to a bound in the multiplicities of the elements of the sum; see Proposition 2.3 for the formal statement. Before discussing its proof, we define the $k$-**projection** of a multiset $M$ as $M_{|k}$ where $M_{|k}(x) = \min\{M(x), k\}$. Given a multiset $N$ of floats in float system $S$, we let $\mathrm{SUM}_S(N)$ denote the output of the sum $f_1 + \cdots + f_\ell$ where **(1)** $f_i$ appears $N(f_i)$ times (i.e., its multiplicity) in the sum, **(2)** the floats appear and are summed in increasing order and **(3)** $+$ is according to $S$.

**Proposition 2.3.** *For all floating-point systems $S$, there exists a $k \in \mathbb{N}$ such that for all multisets $M$ over floats in $S$, we have $\mathrm{SUM}_S(M) = \mathrm{SUM}_S(M_{|k})$.*

*Proof.* (Sketch) See also in Appendix A.2. Let $u = 0.0\cdots01 \times \beta^e$ and $v = 0.10\cdots0 \times \beta^{e+1}$. Now notice that for a large enough $\ell$, summing $u$ to itself $m > \ell$ times will always give $v$. $\qquad\square$

Due to Proposition 2.3, GNNs with floats using sum in increasing order are bounded in their ability to fully count out-neighbours. Thus, it is natural to assume that floating-point GNNs are **bounded** GNNs, i.e., the aggregation function can be written as $\mathcal{M}_k(U^d) \to U^d$ for some **bound** $k \in \mathbb{N}$, i.e., for every multiset $M \in \mathcal{M}(U^d)$, we have $\mathrm{AGG}(M) = \mathrm{AGG}(M_{|k})$, where $M_{|k}$ is the $k$-projection of $M$ (and $U^d$ is the set of states of the GNN). We finally give a formal definition for floating-point GNNs: a **floating-point graph neural network** (GNN[F]) is simply a bounded GNN where the set of states and the domains and co-domains of the functions are restricted to some floating-point system $S$ instead of $\mathbb{R}$ (note that $S$ can be any floating-point system). In R-simple GNN[F]s, $\mathrm{SUM}_S$ replaces the sum of reals as the aggregation function, and their bound is thus determined by the choice of $S$. A GNN[F] obtained by removing the condition on boundedness is called an **unrestricted** GNN[F]. Note that by default and unless otherwise stated, a GNN[$\mathbb{R}$] is unbounded, whereas a GNN[F] is bounded. Now, it is immediately clear that unrestricted GNN[F]s (with an unrestricted aggregation function) are more expressive than GNN[F]s: expressing the property that a node has an even number of out-neighbours is easy with unrestricted GNN[F]s, but no bounded GNN[F] with bound $k$ can distinguish the centers of two star graphs with $k$ and $k+1$ out-neighbours.

## 2.2 Logics

We then define the logics relevant for this paper. For $\Pi \subseteq \mathrm{LAB}$, the set of $\Pi$-**formulae of graded modal logic** (GML) is given by the grammar $\varphi ::= \top \mid p \mid \neg\varphi \mid \varphi \wedge \varphi \mid \Diamond_{\geq k}\varphi$, where $p \in \Pi$ and $k \in \mathbb{N}$. The connectives $\vee, \to, \leftrightarrow$ are considered abbreviations in the usual way. Note that node label symbols serve as propositional symbols here. A $\Pi$-formula of GML is interpreted in pointed $\Pi$-labeled graphs. In the context of modal logic, these are often called (pointed) Kripke models. Let $G = (V, E, \lambda)$ be a $\Pi$-labeled graph and $w \in V$. The truth of a formula $\varphi$ in a pointed graph $(G, w)$ (denoted $G, w \models \varphi$) is defined as usual for the Boolean operators and $\top$, while for $p \in \Pi$ and $\Diamond_{\geq k}\varphi$, we define that $G, w \models p$ iff $p \in \lambda(w)$, and $G, w \models \Diamond_{\geq k}\varphi$ iff $G, v \models \varphi$ for at least $k$ out-neighbours $v$ of $w$. We use the abbreviations $\Diamond\varphi := \Diamond_{\geq 1}\varphi$, $\Box\varphi := \neg\Diamond\neg\varphi$ and $\Diamond_{=n}\varphi := \Diamond_{\geq n}\varphi \wedge \neg\Diamond_{\geq n+1}\varphi$. The set of $\Pi$-**formulae of** $\omega$-GML is given by the grammar $\varphi ::= \psi \mid \bigvee_{\psi \in \Psi} \psi$, where $\psi$ is a $\Pi$-formula of GML and $\Psi$ is an at most countable set of $\Pi$-formulae of GML. The truth of infinite disjunctions is defined in the obvious way: $G, w \models \bigvee_{\psi \in \Psi} \psi \iff G, w \models \psi$ for some $\psi \in \Psi$.

We next introduce the graded modal substitution calculus (or GMSC), which extends the modal substitution calculus [24, 1, 3] with counting capabilities. Define the set $\mathrm{VAR} = \{V_i \mid i \in \mathbb{N}\}$ of **schema variable symbols**. Let $\mathcal{T} = \{X_1, \ldots, X_n\} \subseteq \mathrm{VAR}$. The set of $(\Pi, \mathcal{T})$-**schemata** of

GMSC is defined by the grammar $\varphi ::= \top \mid p \mid X_i \mid \neg\varphi \mid \varphi \wedge \varphi \mid \Diamond_{\geq k}\varphi$ where $p \in \Pi$, $X_i \in \mathcal{T}$ and $k \in \mathbb{N}$. A $(\Pi, \mathcal{T})$-**program** $\Lambda$ of GMSC consists of two lists of expressions

$$X_1(0) :- \varphi_1 \quad \cdots \quad X_n(0) :- \varphi_n \qquad\qquad X_1 :- \psi_1 \quad \cdots \quad X_n :- \psi_n$$

where $\varphi_1, \ldots, \varphi_n$ are $\Pi$-formulae of GML and $\psi_1, \ldots, \psi_n$ are $(\Pi, \mathcal{T})$-schemata of GMSC. Moreover, each program is associated with a set $\mathcal{A} \subseteq \mathcal{T}$ of **appointed predicates**. A program of **modal substitution calculus** MSC is a program of GMSC that may only use diamonds $\Diamond$ of the standard modal logic. The expressions $X_i(0) :- \varphi_i$ are called **terminal clauses** and $X_i :- \psi_i$ are called **iteration clauses**. The schema variable $X_i$ in front of the clause is called the **head predicate** and the formula $\varphi_i$ (or schema $\psi_i$) is called the **body** of the clause. The terminal and iteration clauses are the rules of the program. When we do not need to specify $\mathcal{T}$, we may refer to a $(\Pi, \mathcal{T})$-program as a $\Pi$-**program** of GMSC. Now, the $n$**th iteration formula** $X_i^n$ of a head predicate $X_i$ (or the iteration formula of $X_i$ in **round** $n \in \mathbb{N}$) (w.r.t. $\Lambda$) is defined as follows. The 0th iteration formula $X_i^0$ is $\varphi_i$ and the $(n+1)$st iteration formula $X_i^{n+1}$ is $\psi_i$ where each head predicate $Y$ in $\psi_i$ is replaced by the formula $Y^n$. We write $G, w \models \Lambda$ and say that $\Lambda$ **accepts** $(G, w)$ iff $G, w \models X^n$ for some appointed predicate $X \in \mathcal{A}$ and some $n \in \mathbb{N}$. Moreover, for all $(\Pi, \mathcal{T})$-schemata $\varphi$ that are not head predicates and for $n \in \mathbb{N}$, we let $\varphi^n$ denote the formula (w.r.t. $\Lambda$) where each $Y \in \mathcal{T}$ in $\varphi$ is replaced by $Y^n$.

Recall that **monadic second-order logic** MSO is obtained as an extension of **first-order logic** FO by allowing quantification of unary relation variables $X$, i.e., if $\varphi$ is an MSO-formula, then so are $\forall X\varphi$ and $\exists X\varphi$, see e.g. [9] for more details. Given a set $\Pi \subseteq$ LAB of node label symbols, an FO- or MSO-formula $\varphi$ over $\Pi$ is an FO- or MSO-formula over a vocabulary which contains exactly a unary predicate for each $p \in \Pi$ and the edge relation symbol $E$. Equality is admitted.

Let $\varphi$ be an $\omega$-GML-formula, GMSC-schema, GMSC-program, or a rule of a program. The **modal depth** (resp. the **width**) of $\varphi$ is the maximum number of nested diamonds in $\varphi$ (resp. the maximum number $k \in \mathbb{N}$ that appears in a diamond $\Diamond_{\geq k}$ in $\varphi$). If an $\omega$-GML-formula has no maximum depth (resp., width), its modal depth (resp., width) is $\infty$. If an $\omega$-GML-formula has finite modal depth (resp., width), it is **depth-bounded** (resp., **width-bounded**). The **formula depth** of a GML-formula or GMSC-schema is the maximum number of nested operators $\neg$, $\wedge$ and $\Diamond_{\geq k}$. Given a $\Pi$-program of GMSC or a $\Pi$-formula of $\omega$-GML $\varphi$ (respectively, a formula $\psi(x)$ of MSO or FO over $\Pi$, where the only free variable is the first-order variable $x$), we say that $\varphi$ (resp., $\psi(x)$) **expresses** a node property $\mathcal{P}$ over $\Pi$, if for each pointed $\Pi$-labeled graph $(G, w)$: $(G, w) \in \mathcal{P}$ iff $G, w \models \varphi$ (or resp. $G \models \psi(w)$). A node property $\mathcal{P}$ over $\Pi$ is **expressible** in GMSC (resp., in $\omega$-GML, FO or MSO) if there is a $\Pi$-program of GMSC (resp., a $\Pi$-formula of $\omega$-GML, FO or MSO) expressing $\mathcal{P}$.

**Example 2.4.** Recall the property *reachability of node label symbol* $p$ over $\Pi$ defined in Example 2.1. It is expressed by the $\Pi$-program of GMSC $X(0) :- p$, $X :- \Diamond X$, where $X$ is an appointed predicate. The $i$th iteration formula of $X$ is $X^i = \Diamond \cdots \Diamond p$ where there are exactly $i$ diamonds.

**Example 2.5** ([24]). A pointed $\Pi$-labeled graph $(G, w)$ has the *centre-point property* $\mathcal{P}$ over $\Pi$ if there exists an $n \in \mathbb{N}$ such that each directed path starting from $w$ leads to a node with no out-neighbours in exactly $n$ steps. It is easy to see that the $\Pi$-program $X(0) :- \Box\bot$, $X :- \Diamond X \wedge \Box X$, where $X$ is an appointed predicate, expresses $\mathcal{P}$. In [24], it was established that the centre-point property is not expressible in MSO and that there are properties expressible in $\mu$-calculus and MSO that are not expressible in MSC (e.g., non-reachability), with the same proofs applying to GMSC.

**Proposition 2.6.** *Properties expressible in* GMSC *are expressible in* $\omega$-GML*, but not vice versa.*

*Proof.* A property over $\Pi$ expressed by a $\Pi$-program $\Gamma$ of GMSC is expressible in $\omega$-GML by the $\Pi$-formula $\bigvee_{X_i \in \mathcal{A},\, n \in \mathbb{N}} X_i^n$, where $X_i^n$ is the $n$th iteration formula of appointed predicate $X_i$ of $\Gamma$. Like GNN[$\mathbb{R}$]s, $\omega$-GML can express undecidable properties (cf. Remark 2.2). Clearly GMSC-programs $\Lambda$ cannot, as configurations defined by $\Lambda$ in a finite graph eventually loop, i.e., the truth values of iteration formulae start repeating cyclically. $\qquad\square$

While GMSC is related to the graded modal $\mu$-calculus ($\mu$GML), which originates from [22] and is used in [26] to characterize a recurrent GNN model, $\mu$GML and GMSC are orthogonal in expressivity. Iteration in GMSC need *not* be over a monotone function and does not necessarily yield a fixed point, and there are no syntactic restrictions that would, e.g., force schema variables to be used only positively as in $\mu$GML. The centre-point property from Example 2.5 is a simple property not expressible in $\mu$GML (as it is not even expressible in MSO, into which $\mu$GML translates). Conversely, GMSC offers neither greatest fixed points nor fixed point alternation. In particular, natural

properties expressible in the $\nu$-fragment of $\mu$GML such as non-reachability are not expressible in GMSC; this is proved in [24] for the non-graded version, and the same proof applies to GMSC. However, the $\mu$-fragment of the graded modal $\mu$-calculus translates into GMSC (by essentially the same argument as the one justifying Proposition 7 in [24]). We also note that GMSC translates into partial fixed-point logic with choice [28], but it is not clear whether the same holds without choice.

## 3   Connecting GNNs and logics via automata

In this section we establish exact matches between classes of GNNs and our logics. The first main theorem is Theorem 3.2, showing that GNN[F]s, R-simple aggregate-combine GNN[F]s and GMSC are equally expressive. Theorem 3.4 is the second main result, showing that GNN[$\mathbb{R}$]s and $\omega$-GML are equally expressive. We begin by defining the concept of distributed automata which we will mainly use as a tool for our arguments but they also lead to nice additional characterizations. Informally, we consider a model of distributed automata called counting message passing automata and its variants which operate similarly to GNNs. These distributed automata update the state of each node according to the node's own state and the *multiset* of states of its out-neighbours.

Formally, given $\Pi \subseteq \mathrm{LAB}$, a **counting message passing automaton** (CMPA) over $\Pi$ is a tuple $(Q, \pi, \delta, F)$ where $Q$ is an at most countable set of *states* and $\pi$, $\delta$ and $F$ are defined in a similar way as for GNNs: $\pi\colon \mathcal{P}(\Pi) \to Q$ is an *initialization function*, $\delta\colon Q \times \mathcal{M}(Q) \to Q$ a *transition function* and $F \subseteq Q$ a set of *accepting states*. Computation of a CMPA over $\Pi$ is defined in a $\Pi$-labeled graph $G$ analogously to GNNs: for each node $w$ in $G$, the initialization function gives the initial state for $w$ based on the node label symbols true in $w$, and the transition function is applied to the previous state of $w$ and the multiset of states of out-neighbours of $w$. Acceptance is similar to GNNs: the CMPA **accepts** $(G, w)$ if the CMPA visits (at least once) an accepting state at $w$ in $G$. A **bounded** CMPA is a CMPA whose transition function can be written as $\delta\colon Q \times \mathcal{M}_k(Q) \to Q$ for some $k \in \mathbb{N}$ (i.e., $\delta(q, M) = \delta(q, M_{|k})$ for each multiset $M \in \mathcal{M}(Q)$ and state $q \in Q$). A **finite** CMPA (FCMPA) is a CMPA with finite $Q$. We define bounded FCMPAs similarly to bounded CMPAs. A CMPA $\mathcal{A}$ over $\Pi$ **expresses** a node property $\mathcal{P}$ over $\Pi$ if $\mathcal{A}$ accepts $(G, w)$ iff $(G, w) \in \mathcal{P}$. We define whether a node property $\mathcal{P}$ is expressible by a CMPA in a way analogous to GNNs.

For any $\Pi$, a $\Pi$-object refers to a GNN over $\Pi$, a CMPA over $\Pi$, a $\Pi$-formula of $\omega$-GML or a $\Pi$-program of GMSC. Let $\mathcal{C}$ be the class of all $\Pi$-objects for all $\Pi$. Two $\Pi$-objects in $\mathcal{C}$ are **equivalent** if they express the same node property over $\Pi$. Subsets $A, B \subseteq \mathcal{C}$ **have the same expressive power**, if each $x \in A$ has an equivalent $y \in B$ and vice versa. It is easy to obtain the following.

**Proposition 3.1.** *Bounded* FCMPA*s have the same expressive power as* GMSC.

*Proof.* (Sketch) Details in Appendix B.2. To obtain a bounded FCMPA equivalent to a GMSC-program $\Lambda$, we first turn $\Lambda$ into an equivalent program $\Gamma$ where the modal depth of terminal clauses (resp., iteration clauses) is $0$ (resp., at most $1$). Then from $\Gamma$ with head predicate set $\mathcal{T}'$, we construct an equivalent FCMPA $\mathcal{A}$ as follows. The set of states of $\mathcal{A}$ is $\mathcal{P}(\Pi \cup \mathcal{T}')$ and $\mathcal{A}$ enters in round $n \in \mathbb{N}$ in node $w$ into a state that contains precisely the node label symbols true in $w$ and the predicates $X$ whose iteration formula $X^n$ is true at $w$. For the converse, we create a head predicate for each state in $\mathcal{A}$, and let predicates for accepting states be appointed. The terminal clauses simulate $\pi$ using disjunctions of conjunctions of non-negated and negated node label symbols. The iteration clauses simulate $\delta$ using, for each pair $(q, q')$, a subschema specifying the multisets that take $q$ to $q'$. □

We are ready to show equiexpressivity of GMSC and GNN[F]s. This applies *without relativising to any background logic*. The direction from GNN[F]s to GMSC is trivial. The other direction is more challenging, in particular when going all the way to R-simple GNN[F]s. While size issues were not a concern in this work, we observe that the translation from GNN[F]s to GMSC involves only polynomial blow-up in size; the related definitions and proofs are in appendices A.1, B.2 and B.3. We also conjecture that a polynomial translation from GMSC to R-simple GNN[F]s is possible by the results and techniques in [3], taking into account differences between GNN[F]s and R-simple GNN[F]s w.r.t. the definition of size. A more serious examination of blow-ups would require a case-by-case analysis taking other such details into account.

**Theorem 3.2.** *The following have the same expressive power:* GNN[F]*s,* GMSC*, and R-simple aggregate-combine* GNN[F]*s.*

*Proof.* (Sketch) Details in Appendix B.3. By definition, a GNN[F] is just a bounded FCMPA and translates to a GMSC-program by Proposition 3.1. To construct an R-simple aggregate-combine GNN[F] $\mathcal{G}$ for a GMSC-program $\Lambda$ with formula depth $D$, we first turn $\Lambda$ into an equivalent program $\Gamma$, where each terminal clause has the body $\bot$, the formula depth of each body of an iteration clause is $D'$ (linear in $D$) and for each subschema of $\Gamma$ that is a conjunction, both conjuncts have the same formula depth if neither conjunct is $\top$. We choose a floating-point system that can express all integers up to the width of $\Gamma$. We define binary feature vectors that are split into two halves: the 1st half intuitively calculates the truth values of the head predicates and subschemata of $\Gamma$ one formula depth at a time in the style of Barceló et al. [5]. The 2nd half records the current formula depth under evaluation. $\mathcal{G}$ simulates one round of $\Gamma$ in $D' + 1$ rounds, using the 2nd half of the features to accept nodes only every $D' + 1$ rounds: the truth values of head predicates are correct in those rounds. Note that the choice of floating-point system in $\mathcal{G}$ depends on $\Lambda$ and thus no single floating-point system is used by all GNNs resulting from the translation. In fact, fixing a single floating-point system would trivialize the computing model as only finitely many functions could be defined. $\qquad\square$

To show that GNN[$\mathbb{R}$]s and $\omega$-GML are equally expressive, we first prove a useful theorem.

**Theorem 3.3.** CMPA*s have the same expressive power as $\omega$-GML.*

*Proof.* (Sketch) Details in Appendix B.4. To construct a CMPA for each $\omega$-GML-formula $\chi$, we define a GML-formula called the "*full graded type of modal depth* $n$" for each pointed graph $(G, w)$, which expresses all the local information of the neighbourhood of $w$ up to depth $n$ (with the maximum out-degree of $G$ plus 1 sufficing for width). We show that each $\omega$-GML-formula $\chi$ is equivalent to an infinite disjunction $\psi_\chi$ of types. We then define a CMPA that computes the type of modal depth $n$ of each node in round $n$. Its accepting states are the types in the type-disjunction $\psi_\chi$. For the converse, to show that each CMPA has an equivalent $\omega$-GML-formula, we first show that two pointed graphs satisfying the same full graded type of modal depth $n$ have identical states in each round $\ell \leq n$ in each CMPA. The $\omega$-GML-formula is the disjunction containing every type $T$ such that some $(G, w)$ satisfying $T$ is accepted by the automaton in round $n$, where $n$ is the depth of $T$. $\qquad\square$

Next we characterize GNN[$\mathbb{R}$]s with $\omega$-GML *without relativising to a background logic*. As the theorems above and below imply that GNN[$\mathbb{R}$]s and CMPAs are equally expressive, it follows that GNN[$\mathbb{R}$]s can be discretized to CMPAs having—by definition—an only countable set of states.

**Theorem 3.4.** GNN[$\mathbb{R}$]*s have the same expressive power as $\omega$-GML.*

*Proof.* (Sketch) Details in Appendix B.5. For any GNN[$\mathbb{R}$], we build an equivalent $\omega$-GML-formula using the same method as in the proof of Theorem 3.3, where we show that for each CMPA, we can find an equivalent $\omega$-GML-formula. For the converse, we first translate an $\omega$-GML-formula to a CMPA $\mathcal{A}$ by Theorem 3.3 and then translate $\mathcal{A}$ to an equivalent CMPA $\mathcal{A}'$ with maximal ability to distinguish nodes. Then we build an equivalent GNN[$\mathbb{R}$] for $\mathcal{A}'$ by encoding states into integers. $\qquad\square$

**Remark 3.5.** It is easy to show that unrestricted GNN[F]s have the same expressive power as FCMPAs. Moreover, the proof of Theorem 3.4 is easily modified to show that bounded GNN[$\mathbb{R}$]s have the same expressive power as width-bounded $\omega$-GML. See Appendix B.6 for the proofs.

In Appendix B.7 we show that our model of constant-iteration GNN[$\mathbb{R}$]s is equivalent to the one in Barceló et al. [5]. Thus Theorem 3.6 (proven in Appendix B.8) generalizes the result in [5] saying that any property $\mathcal{P}$ expressible by FO is expressible as a constant-iteration GNN[$\mathbb{R}$] iff it is expressible in GML. Furthermore, any such $\mathcal{P}$ is expressible in GML iff it is expressible in $\omega$-GML.

**Theorem 3.6.** *Constant-iteration* GNN[$\mathbb{R}$]*s have the same expressive power as depth-bounded $\omega$-GML.*

## 4 Characterizing GNNs over MSO-expressible properties

In this section we consider properties expressible in MSO. The first main result is Theorem 4.1, where we show that for properties expressible in MSO, the expressive power of GNN[$\mathbb{R}$]s is captured by a *finitary* logic. In fact, this logic is GMSC and by Theorem 3.2, it follows that relative to MSO, GNN[$\mathbb{R}$]s have the same expressive power as GNN[F]s (Theorem 4.3 below). Our arguments in this section work uniformly for any finite set $\Sigma_N$ of node label symbols.

**Theorem 4.1.** *Let $\mathcal{P}$ be a property expressible in* MSO. *Then $\mathcal{P}$ is expressible as a* GNN[$\mathbb{R}$] *if and only if it is expressible in* GMSC.

Theorem 4.1 is proved later in this section. The recent work [26] relates to Theorem 4.1; see the introduction for a discussion. The proof of Theorem 4.1 can easily be adapted to show that a property $\mathcal{P}$ expressible in MSO is expressible as a constant-iteration GNN[$\mathbb{R}$] iff it is expressible in GML. This relates to [5] which shows the same for FO in place of MSO. However, in this particular case the MSO version is not an actual generalization as based on [10], we show the following in Appendix C.1.

**Lemma 4.2.** *Any property expressible in* MSO *and as a constant-iteration* GNN[$\mathbb{R}$] *is also* FO-*expressible.*

Uniting Theorems 4.1 and 3.2, we see that (recurrent) GNN[$\mathbb{R}$]s and GNN[F]s coincide relative to MSO. It is easy to get a similar result for constant-iteration GNNs (the details are in Appendix C.3).

**Theorem 4.3.** *Let $\mathcal{P}$ be a property expressible in* MSO. *Then $\mathcal{P}$ is expressible as a* GNN[$\mathbb{R}$] *if and only if it is expressible as a* GNN[F]. *The same is true for constant-iteration* GNN*s.*

To put Theorem 4.3 into perspective, we note that by Example 2.5, the centre-point property is expressible in GMSC, but not in MSO. From Theorem 3.2, we thus obtain the following.

**Proposition 4.4.** *There is a property $\mathcal{P}$ that is expressible as a* GNN[F] *but not in* MSO.

Theorem 3.6 shows that constant-iteration GNN[$\mathbb{R}$]s and depth-bounded $\omega$-GML are equally expressive. The proof of Proposition 2.6 shows that already depth-bounded $\omega$-GML can express properties that GMSC cannot, in particular undecidable ones. Thus, by Theorem 3.2 we obtain the following.

**Proposition 4.5.** *There is a property expressible as a constant-iteration* GNN[$\mathbb{R}$] *but not as a* GNN[F].

We next discuss the proof of Theorem 4.1. We build upon results due to Janin and Walukiewicz [31, 20], reusing an automaton model from [31] that captures the expressivity of MSO on tree-shaped graphs. With a tree-shaped graph we mean a *potentially infinite* graph that is a directed tree. The out-degree of nodes is unrestricted (but nevertheless finite), different nodes may have different degree, and both leaves and infinite paths are admitted in the same tree.

We next introduce the mentioned automaton model. Although we are only going to use it on tree-shaped $\Sigma_N$-labeled graphs, in its full generality it is actually defined on unrestricted such graphs. We nevertheless call them **tree automata** as they belong to the tradition of more classical forms of such automata. In particular, a run of an automaton is tree-shaped, even if the input graph is not. Formally, a **parity tree automaton (PTA)** is a tuple $\mathcal{A} = (Q, \Sigma_N, q_0, \Delta, \Omega)$, where $Q$ is a finite set of **states**, $\Sigma_N \subseteq$ LAB is a finite set of node label symbols, $q_0 \in Q$ is an **initial state**, $\Delta \colon Q \times \mathcal{P}(\Sigma_N) \to \mathcal{F}$ is a transition relation with $\mathcal{F}$ being the set of all **transition formulas** for $\mathcal{A}$ defined below, and $\Omega \colon Q \to \mathbb{N}$ is a **priority function**. A transition formula for $\mathcal{A}$ is a disjunction of FO-formulas of the form

$$\exists x_1 \cdots \exists x_k \left( \mathsf{diff}(x_1, \ldots, x_k) \wedge q_1(x_1) \wedge \cdots \wedge q_k(x_k) \wedge \forall z \left( \mathsf{diff}(z, x_1, \ldots, x_k) \to \psi \right) \right)$$

where $k \geq 0$, $\mathsf{diff}(y_1, \ldots, y_n)$ shortens an FO-formula declaring $y_1, \ldots, y_n$ as pairwise distinct, $q_i \in Q$ are states used as unary predicates and $\psi$ is a disjunction of conjunctions of atoms $q(z)$, with $q \in Q$. A PTA $\mathcal{A}$ accepts a language $L(\mathcal{A})$ consisting of (possibly infinite) graphs $G$. We have $G \in L(\mathcal{A})$ if there is an accepting **run** of $\mathcal{A}$ on $G$, and runs are defined in the spirit of alternating automata. While details are in Appendix C.2, we mention that transition formulas govern transitions in the run: a transition of a PTA currently visiting node $v$ in state $q$ consists of sending copies of itself to out-neighbours of $v$, potentially in states other than $q$. It is not required that a copy is sent to every out-neighbour, and multiple copies (in different states) can be sent to the same out-neighbour. However, we must find some $(q, \lambda(v), \vartheta) \in \Delta$ such that the transition satisfies $\vartheta$ in the sense that $\vartheta$ is satisfied in the graph with one element for each out-neighbour of $v$ and unary predicates (states of $\mathcal{A}$) are interpreted according to the transition. This specific form of PTAs is interesting to us due to the following.

**Theorem 4.6.** *Let $\mathcal{P}$ be a property expressible in* MSO *and in* $\omega$-GML. *Then there is a PTA $\mathcal{A}$ such that for any graph $G$:* $(G, w) \in \mathcal{P}$ *iff the unraveling of $G$ at $w$ is in $L(\mathcal{A})$.*

*Proof.* Let $\varphi(x)$ be the MSO-formula over $\Sigma_N$ that expresses $\mathcal{P}$. Theorem 9 in [20] states that for every MSO-sentence $\psi$, there is a PTA $\mathcal{A}$ such that for every tree-shaped $\Sigma_N$-labeled graph $G$ we have that $G \in L(\mathcal{A})$ iff $G \models \psi$. To obtain a PTA for $\varphi(x)$, we start from the MSO-sentence $\psi := \exists x \, (\varphi(x) \wedge \neg \exists y E(y, x))$ and build the corresponding PTA $\mathcal{A}$. Then $\mathcal{A}$ is as desired. In fact, $(G, w) \in \mathcal{P}$ iff $G \models \varphi(w)$ iff $U \models \varphi(w)$ with $U$ the unraveling of $G$ at $w$ since $\mathcal{P}$ is expressible by an $\omega$-GML-formula which is invariant under unraveling (defined in Appendix C.1). Moreover, $U \models \varphi(w)$ iff $U \models \psi$ iff $U \in L(\mathcal{A})$. $\qquad\square$

Since GMSC-programs are invariant under unraveling, we may now prove Theorem 4.1 by considering PTAs obtained by Theorem 4.6 and constructing a $\Sigma_N$-program $\Lambda$ of GMSC for each such PTA $\mathcal{A} = (Q, \Sigma_N, q_0, \Delta, \Omega)$ so that the following holds: For every tree-shaped $\Sigma_N$-labeled graph $G$ with root $w$, we have $G \in L(\mathcal{A})$ iff $G, w \models \Lambda$. We then say that $\mathcal{A}$ and $\Lambda$ are **tree equivalent**.

For a state $q \in Q$, let $\mathcal{A}_q$ be the PTA defined like $\mathcal{A}$ but with $q$ as its initial state. For a tree-shaped graph $T$, set $Q_T = \{q \in Q \mid T \in L(\mathcal{A}_q)\}$. We say that $\mathcal{Q} \subseteq \mathcal{P}(Q)$ is **the universal set for** $P \subseteq \Sigma_N$ if $\mathcal{Q}$ consists precisely of the sets $Q_T$, for all tree-shaped graphs $T$, whose root is labeled exactly with the node label symbols in $P$. Let $T = (V, E, \lambda)$ be a tree-shaped graph with root $w$ and $k \in \mathbb{N}$, and let $V_k$ be the restriction of $V$ to the nodes on level at most $k$ (the root being on level 0). A $k$-**prefix decoration** of $T$ is a mapping $\mu : V_k \to \mathcal{P}(\mathcal{P}(Q))$ such that the following conditions hold: **(1)** for each $S \in \mu(w)$: $q_0 \in S$; **(2)** for all $v \in V$ on the level $k$, $\mu(v)$ is the universal set for $\lambda(v)$; **(3)** for each $v \in V$ on some level smaller than $k$ that has out-neighbours $u_1, \ldots, u_n$ and all $S_1 \in \mu(u_1), \ldots, S_n \in \mu(u_n)$, $\mu(v)$ contains the set $S$ that contains a state $q \in Q$ iff we have $\Delta(q, \lambda(v)) = \vartheta$ such that $\vartheta$ is satisfied in the following graph: the universe is $\{u_1, \ldots, u_n\}$ and each unary predicate $q' \in Q$ is interpreted as $\{u_i \mid q' \in S_i\}$. Intuitively, a $k$-prefix decoration of $T$ represents a set of accepting runs of $\mathcal{A}$ on the prefix $T_k$ of $T$. As these runs start in universal sets, for each extension of $T_k$ obtained by attaching trees to nodes on level $k$, we can find a run among the represented ones that can be extended to an accepting run of $\mathcal{A}$ on that extension. In fact $T$ is such an extension. The following crucial lemma is proved in Appendix C.2.

**Lemma 4.7.** *For every tree-shaped $\Sigma_N$-labeled graph $T$: $T \in L(\mathcal{A})$ if and only if there is a $k$-prefix decoration of $T$, for some $k \in \mathbb{N}$.*

Using the above, we sketch the proof of Theorem 4.1; the full proof is in Appendix C.2.

*Proof of Theorem 4.1.* By Lemma 4.7, given a PTA $\mathcal{A}$ obtained from Lemma 4.6, we get a tree equivalent GMSC-program $\Lambda$ by building $\Lambda$ to accept the root of a tree-shaped graph $G$ iff $G$ has a $k$-prefix decoration for some $k$. This requires care, but is possible; the details are in Appendix C.2. A crucial part of the proof is transferring the set of transition formulas of $\mathcal{A}$ into the rules of $\Lambda$. $\quad\square$

## 5 Conclusion

We have characterized the expressivity of recurrent GNNs with floats and reals in terms of modal logics, both in general and relative to MSO. Particularly, in restriction to MSO, GNNs with floats and with reals have the same expressiveness. We mention two interesting directions for future research. The first one is to extend our GNN model, e.g., with global readouts. On the GML side, these correspond to the (counting) global modality [7]. Interestingly, it can be proved—via showing that fixed points can be defined—that this extension makes non-reachability expressible, cf. Example 2.5. The second direction is to consider other acceptance conditions for recurrent GNNs. Intuitively, it should be possible to establish Theorem 3.2 for virtually any GNN acceptance condition, as long as the acceptance condition of GMSC is changed accordingly. Both of these directions are explored in [4]. In particular, it is shown there (and in Appendix D.1) that Theorem 3.2 also holds for recurrent GNNs where termination is based on fixed points, studied under the name "RecGNN" in [26]. These fixed points require only that a round is reached where the examined node does not exit the set of accepting states which is closely related to but different from the fixed point condition in the inaugural works [30, 11] which requires all nodes in the network to reach a stable state.

**Limitations:** For our acceptance condition (and for many others such as those based on fixed points), training and applying a recurrent GNN brings questions of termination. These are very important from a practical perspective, but not studied in this paper. A particularly interesting question is whether termination can be learned in a natural way during the training phase.

## Acknowledgements

The list of authors on the first page is given based on the alphabetical order. Veeti Ahvonen was supported by the Vilho, Yrjö and Kalle Väisälä Foundation of the Finnish Academy of Science and Letters. Damian Heiman was supported by the Magnus Ehrnrooth Foundation. Antti Kuusisto was supported by the Research Council of Finland consortium project Explaining AI via Logic (XAILOG), grant number 345612, and the Research Council of Finland project Theory of computational logics, grant numbers 352419, 352420, 353027, 324435, 328987; Damian Heiman was also supported by the same project, grant number 353027. Carsten Lutz was supported by the Federal Ministry of Education and Research of Germany (BMBF) and by Sächsisches Staatsministerium für Wissenschaft, Kultur und Tourismus in the program Center of Excellence for AI-research "Center for Scalable Data Analytics and Artificial Intelligence Dresden/Leipzig", project identification number: ScaDS.AI. Lutz is also supported by BMBF in DAAD project 57616814 (SECAI, School of Embedded Composite AI) as part of the program Konrad Zuse Schools of Excellence in Artificial Intelligence.

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

# A  Appendix: Preliminaries

## A.1  Notions of size

We start this appendix by presenting definitions for the sizes of GMSC-programs, GNN[F]s and bounded FCMPAs.

The **size** of a $\Pi$-program of GMSC is here defined as the size of $\Pi$ plus the number of occurrences of node label symbols, head predicates and logical operators in the program, with each $\Diamond_{\geq k}$ adding $k$ to the sum instead of just 1.

The **size** of a GNN[F] or bounded FCMPA over $\Pi$ is defined here as follows. For FCMPAs define $U := Q$, and for GNN[F]s define $U := S^d$ where $S$ is the floating-point system and $d$ the dimension of the GNN[F]. In both cases, let $n := |U|$ and let $k$ be the bound. Note that the cardinality of a multiset $\mathcal{M}_k(U)$ is the number $(k+1)^n$ of different functions of type $U \to [0;k]$. Then the size of the GNN[F] or bounded FCMPA is $|\mathcal{P}(\Pi)|$ plus $n \cdot (k+1)^n$, i.e., the sum of the cardinality $|\mathcal{P}(\Pi)|$ of its initialization function $\mathcal{P}(\Pi) \to U$ and the cardinality $n \cdot (k+1)^n$ of its transition function $U \times \mathcal{M}_k(U) \to U$ as look-up tables.

We note that there is no single way to define the sizes of the objects above. For example, counting each node label symbol, head predicate or state into the size just once may be naïve, as there may not be enough distinct symbols in practice. To account for this, we could also define that the size of each node label symbol, head predicate and state is instead $\log(k)$, where $k$ is the number of node label symbols, head predicates or states respectively. Likewise, the size of a diamond $\Diamond_{\geq \ell}$ could be considered $\log(k)$, where $k$ is the maximum $\ell$ appearing in a diamond in the program. However, this happens to not affect the size of our translation from GNN[F]s to GMSC. As another example, defining the size of a single floating-point number as one may be naïve; one could define that the size of a single floating-point number is the length of the string that represents it in the floating-point system instead of one.

## A.2  Proof of Proposition 2.3

We recall Proposition 2.3 and give a more detailed proof.

**Proposition 2.3.** *For all floating-point systems $S$, there exists a $k \in \mathbb{N}$ such that for all multisets $M$ over floats in $S$, we have* $\mathrm{SUM}_S(M) = \mathrm{SUM}_S(M_{|k})$.

*Proof.* Consider a floating-point system $S = ((p,n,\beta),+,\cdot)$ and a multiset $M \in \mathcal{M}(S)$. We assume $f = 0.d_1 \cdots d_p \times \beta^e \in S$ (the case where $f = -0.d_1 \cdots d_p \times \beta^e$ is symmetric). Assume that the sum of all numbers smaller than $f$ in $M$ in increasing order is $s \in S$. Let $s + f^0$ denote $s$ and let $s + f^{\ell+1}$ denote $(s+f^\ell)+f$ for all $\ell \in \mathbb{N}$. It is clear that $s+f^{\ell+1} \geq s+f^\ell$, and furthermore, if $s + f^{\ell+1} = s + f^\ell$, then $s + f^{\ell'} = s + f^\ell$ for all $\ell' \geq \ell$. If $s + f^{\ell+1} = s + f^\ell$, then we say that the sum $s + f^\ell$ has *stabilized*.

Let $\beta' := \beta - 1$. We may assume that $s \geq -0.\beta' \cdots \beta' \times \beta^n$. Since $s + f^{\ell+1} \geq s + f^\ell$ for each $\ell \in \mathbb{N}$ and $S$ is finite, there must exist some $k \in \mathbb{Z}_+$ such that either $s + f^k = s + f^{k-1}$ or $s + f^k = 0.\beta' \cdots \beta' \times \beta^n$ but then $s + f^{k+1} = s + f^k$. Clearly, $k = |S|$ is sufficient for each $s$ and $f$, and thus satisfies the condition of the proposition.

A smaller bound can be found as follows. Assume without loss of generality that $d_1 \neq 0$ (almost all floating-point numbers in $S$ can be represented in this form and the representation does not affect the outcome of a sum; we consider the case for floats not representable in this way separately). Now, consider the case where $s = -0.\beta' \cdots \beta' \times \beta^{e'}$ for some $e' \in [-n, n]$, as some choice of $e'$ clearly maximizes the number of times $f$ can be added. Now it is quite easy to see that that $s + f^{\beta^p - \beta^{p-1}} \geq -0.10 \cdots 0 \times \beta^{e'}$, unless the sum stabilizes sooner. This is because there are exactly $\beta^p - \beta^{p-1}$ numbers in $S$ from $-0.\beta' \cdots \beta' \times \beta^{e'}$ to $-0.10 \cdots 0 \times \beta^{e'}$. After this another $2\beta^{p-1}$ additions ensures that $s + f^{\beta^p + \beta^{p-1}} \geq 0$. (Consider the case where $p = 3$ and $\beta = 10$ and we add the number $f = 0.501 \times 10^0$ to $-0.100 \times 10^3$ repeatedly. We first get $-0.995 \times 10^2$, then $-0.990 \times 10^2$, etc. Each two additions of $f$ is more than adding $f' = 0.001 \times 10^3$ once, and $f'$ can be added exactly $\beta^{p-1}$ times to $-0.100 \times 10^3$ until the sum surpasses 0, meaning that $f$ can

be added at most $2\beta^{p-1}$ times. On the other hand, if $f$ were $0.500 \times 10^0$ or smaller, then the sum would have never reached $-0.100 \times 10^3$ in the first place.)

Let $\ell := \beta^p + \beta^{p-1}$. Let $s' := s + f^\ell$ and assume the sum $s'$ has not yet stabilized. Next, it is clear that for any $i \in [p]$ we have $s' + f^{\beta^i} \geq 0.10\cdots0 \times \beta^{e+i}$ because $d_1 \geq 1$. For $d_1 \leq \frac{\beta}{2}$, we have that $s' + f^{\beta^p} = 0.10\cdots0 \times \beta^{e+p}$ which is where the sum stabilizes. If on the other hand $d_1 > \frac{\beta}{2}$, we have that $s' + f^{\beta^{p+1}} = 0.10\cdots0 \times \beta^{e+p+1}$, which is where the sum stabilizes (this is because $d_1 > \frac{\beta}{2}$ causes the sum to round upwards until the exponent reaches the value $e + p + 1$). Thus, we get a threshold of $\beta^{p+1} + \beta^p + \beta^{p-1}$.

The case where $d_1 = 0$ for all representations of $f$ is analogous; we simply choose one representation and shift our focus to the first $i$ such that $d_i \neq 0$ and perform the same examination. The position of $i$ does not affect the overall analysis.

We conjecture that a smaller bound than $\beta^{p+1} + \beta^p + \beta^{p-1}$ is possible, but involves a closer analysis. $\qquad\square$

# B  Appendix: Connecting GNNs and logics via automata

## B.1  An extended definition for distributed automata

We give a more detailed definition for distributed automata. Given $\Pi \subseteq \mathrm{LAB}$, a **counting message passing automaton** (or CMPA) over $\Pi$ is a tuple $(Q, \pi, \delta, F)$, where

- $Q$ is an at most countable non-empty set of **states**,
- $\pi \colon \mathcal{P}(\Pi) \to Q$ is an **initialization function**,
- $\delta \colon Q \times \mathcal{M}(Q) \to Q$ is a **transition function**, and
- $F \subseteq Q$ is a set of **accepting states**.

If the set of states is finite, then we say that the counting message passing automaton is finite and call it an FCMPA. Note that in the context of FCMPAs, finiteness refers specifically to the number of states. We also define a subclass of counting message passing automata: a $k$-**counting message passing automaton** ($k$-CMPA) is a tuple $(Q, \pi, \delta, F)$, where $Q$, $\pi$ and $F$ are defined analogously to CMPAs and the transition function $\delta$ can be written in the form $Q \times \mathcal{M}_k(Q) \to Q$, i.e., for all multisets $M \in \mathcal{M}(Q)$ we have $\delta(q, M) = \delta(q, M_{|k})$ for all $q \in Q$. A $k$-FCMPA naturally refers to a $k$-CMPA whose set of states is finite; this is truly a finite automaton. For any $k \in \mathbb{N}$, each $k$-CMPA (respectively, each $k$-FCMPA) is called a **bounded** CMPA (resp., a **bounded** FCMPA).

Let $G = (V, E, \lambda)$ be a $\Pi$-labeled graph. We define the computation of a CMPA formally. A CMPA (or resp. $k$-CMPA) $(Q, \pi, \delta, F)$ over $\Pi$ and a $\Pi$-labeled graph $G = (V, E, \lambda)$ define a distributed system, which executes in $\omega$-rounds as follows. Each **round** $n \in \mathbb{N}$ defines a **global configuration** $g_n \colon V \to Q$ which essentially tells in which state each node is in round $n$. If $n = 0$ and $w \in V$, then $g_0(w) = \pi(\lambda(w))$. Now assume that we have defined $g_n$. Informally, a new state for $w$ in round $n + 1$ is computed by $\delta$ based on the previous state of $w$ and the multiset of previous states of its immediate out-neighbours. More formally, let $w$ be a node and $v_1, \ldots, v_m$ its immediate out-neighbours (i.e., the nodes $v \in V$ s.t. $(w, v) \in E$). Let $S$ denote the corresponding multiset of the states $g_n(v_1), \ldots, g_n(v_m)$. We define $g_{n+1}(w) = \delta(g_n(w), S)$. We say that $g_n(w)$ is the **state of $w$ at round $n$**.

A CMPA (or resp. $k$-CMPA) **accepts** a pointed graph $(G, w)$ if and only if $g_n(w) \in F$ for some $n \in \mathbb{N}$. If $g_n(w) \in F$, then we say that the CMPA **accepts** $(G, w)$ **in round** $n$. The computation for FCMPAs is defined analogously (since each FCMPA is a CMPA).

## B.2  Proof of Proposition 3.1

Note that extended definitions for automata are in Appendix B.1.

First we recall Proposition 3.1.

**Proposition 3.1.** *Bounded* FCMPA*s have the same expressive power as* GMSC.

To prove Proposition 3.1 (in the end of this subsection), we first prove an auxiliary Lemma B.1 that informally shows that we can translate any GMSC-program into an equivalent program, where the modal depth of the terminal clauses is zero and the modal depth of the iteration clauses is at most one. Then in Lemma B.2 we show with the help of Lemma B.1 that for each GMSC-program we can construct an equivalent bounded FCMPA. Finally, we show in Lemma B.3 that for each bounded FCMPA we can construct an equivalent GMSC-program.

We start with an auxiliary result.

**Lemma B.1.** *For every* $\Pi$*-program of* GMSC*, we can construct an equivalent* $\Pi$*-program* $\Gamma$ *of* GMSC *such that the modal depth of each terminal clause of* $\Gamma$ *is* 0*, and the modal depth of each iteration clause of* $\Gamma$ *is at most* 1*.*

*Proof.* The proof is similar to the proof of Theorem 11 in [1] (or the proof of Theorem 5.4 in [2]). We consider an example and conclude that the strategy mentioned in the cited papers can be generalized to our framework.

For simplicity, in the example below we consider a program where the modal depth of each terminal clause is zero, since it is easy to translate any program of GMSC into an equivalent GMSC-program, where the modal depth of the terminal clauses is zero. Then from the program of GMSC, where the modal depth of each terminal clause is zero, it is easy to obtain an equivalent GMSC-program where the modal depth of each terminal clause is zero and the modal depth of each iteration clause is at most one.

Consider a $\{q\}$-program $\Lambda$, with the rules $X(0) :- \bot$, $X :- \Diamond_{\geq 3}(\neg\Diamond_{\geq 2}\Diamond_{\geq 1}X \wedge \Diamond_{\geq 3}q)$. The modal depth of the iteration clause of $\Lambda$ is 3. The program $\Lambda$ can be translated into an equivalent program of GMSC where the modal depth of the iteration clauses is at most one as follows. First, we define the following subprogram called a "clock":

$$T_1(0) :- \top \qquad T_1 :- T_3$$
$$T_2(0) :- \bot \qquad T_2 :- T_1$$
$$T_3(0) :- \bot \qquad T_3 :- T_2.$$

We split the evaluation of each subformula between corresponding head predicates $X_{1,1}$, $X_{1,2}$, $X_2$ and $X_3$ and define their terminal clauses and iteration clauses as follows. The body for each terminal clause is $\bot$ and the iteration clauses are defined by

- $X_{1,1} :- (T_1 \wedge \Diamond_{\geq 1}X_3) \vee (\neg T_1 \wedge X_{1,1})$,

- $X_{1,2} :- (T_1 \wedge \Diamond_{\geq 3}q) \vee (\neg T_1 \wedge X_{1,2})$,

- $X_2 :- (T_2 \wedge \Diamond_{\geq 2}X_{1,1}) \vee (\neg T_2 \wedge X_2)$,

- $X_3 :- (T_3 \wedge \Diamond_{\geq 3}(\neg X_2 \wedge X_{1,1})) \vee (\neg T_3 \wedge X_3)$.

The appointed predicate of the program is $X_3$.

Let us analyze how the obtained program works. The program works in a periodic fashion: a single iteration round of $\Lambda$ is simulated in a 3 step period by the obtained program. The clock (i.e. the head predicates $T_1$, $T_2$ and $T_3$) makes sure that each level of the modal depth is evaluated once during each period in the correct order. For example, if $T_2$ is true, then the truth of $X_2$ depends on the truth of $\Diamond_{\geq 2}X_{1,1}$ and when $T_2$ is false, then the truth of $X_2$ stays the same. The head predicate $X_3$ essentially simulates the truth of $X$ in $\Lambda$ in the last round of each period. It is easy to show by induction on $n \in \mathbb{N}$ that for all pointed $\{q\}$-labeled graphs $(G, w)$ and for all $k \in [0; 2]$: $G, w \models X^n$ iff $G, w \models X_3^{n3+k}$. $\qquad\square$

With Lemma B.1, it is easy to construct an equivalent bounded FCMPA for any GMSC-program.

**Lemma B.2.** *For each* $\Pi$*-program of* GMSC*, we can construct an equivalent bounded* FCMPA *over* $\Pi$*.*

*Proof.* Let $\Lambda$ be a $(\Pi, \mathcal{T})$-program of GMSC. By Lemma B.1 we obtain an equivalent $(\Pi, \mathcal{T}')$-program $\Gamma$ where the modal depth is 0 for each terminal clause and at most 1 for each iteration clause. Let $k$ be the width of $\Gamma$. We create an equivalent $k$-FCMPA $\mathcal{A}_\Gamma$ as follows.

We create a state for each subset $q \subseteq \Pi \cup \mathcal{T}'$; these form the state set $Q = \mathcal{P}(\Pi \cup \mathcal{T}')$.

We formulate the initialization function $\pi$ as follows: $\pi(P) = q$ if $\Pi \cap q = P$ and for each head predicate $X \in \mathcal{T}'$ (with the terminal clause $X(0) :- \varphi$) we have that $X \in q$ if and only if $\varphi$ is true when exactly the node label symbols in $P$ are true. (Here it is vital that the terminal clauses have modal depth 0.)

Before we define the transition function, we define an auxiliary relation $\Vdash$ for schemata with modal depth at most 1. Let $N \in \mathcal{M}(Q)$ be a multiset of states and let $q$ be a state. Given a $(\Pi, \mathcal{T})$-schema $\psi$ with modal depth at most 1, we define the relation $(q, N) \Vdash \psi$ recursively as follows: $(q, N) \Vdash \top$ always, $(q, N) \Vdash p$ iff $p \in q$, $(q, N) \Vdash X$ iff $X \in q$, $(q, N) \Vdash \neg\varphi$ iff $(q, N) \not\Vdash \varphi$, $(q, N) \Vdash \varphi \wedge \theta$ iff $(q, N) \Vdash \varphi$ and $(q, N) \Vdash \theta$, and $(q, N) \Vdash \Diamond_{\geq \ell}\varphi$ iff there is a set $Q' \subseteq Q$ of states such that $\sum_{q' \in Q'} N(q') \geq \ell$ and $(q', \emptyset) \Vdash \varphi$ for all $q' \in Q'$ (note that $\varphi$ has modal depth 0).

Now we shall formulate the transition function $\delta$ as follows. Let $N \in \mathcal{M}(Q)$ be a multiset of states and let $q$ be a state. For each node label symbol $p \in \Pi$, $p \in \delta(q, N)$ iff $(q, N) \Vdash p$ (i.e. $p \in q$). For each head predicate $X \in \mathcal{T}'$ with the iteration clause $X :- \psi$ we have that $X \in \delta(q, N)$ if and only if $(q, N) \Vdash \psi$. (Here it is crucial that the iteration clauses have modal depth at most 1.)

The set of accepting states is defined as follows. If $X \in q$ for some appointed predicate $X$ of $\Gamma$ then $q \in F$; otherwise $q \notin F$.

If $G$ is $\Pi$-labeled graph and $v$ is a node in $G$, then we let $v^n$ denote the state of $\mathcal{A}_\Gamma$ at $v$ in round $n$. Moreover, if $N$ is the set of out-neighbours of $v$, we let $N_v^n$ denote the multiset $\{\!\{u^n \mid u \in N\}\!\}$. Then we prove by induction on $n \in \mathbb{N}$ that for any $(\Pi, \mathcal{T}')$-schema $\varphi$ of modal depth 1 and for each pointed $\Pi$-labeled graph $(G, w)$, we have $(w^n, N_w^n) \Vdash \varphi$ iff $G, w \models \varphi^n$.

If $n = 0$ we prove by induction on the structure of $\varphi$ that $(w^0, N_w^0) \Vdash \varphi$ iff $G, w \models \varphi^0$.

- Case $\varphi = p \in \Pi$: Trivial, since $(w^0, N_w^0) \Vdash p$ iff $p \in w^0$ iff $G, w \models p$ iff $G, w \models p^0$.

- Case $\varphi = X \in \mathcal{T}'$: Let $\psi_X$ be the body of the terminal clause of $X$. Now, $(w^0, N_w^0) \Vdash X$ iff $X \in w^0$ iff $G, w \models \psi_X$ iff $G, w \models X^0$.

Now, assume that the induction hypothesis holds for $(\Pi, \mathcal{T}')$-schemata $\psi$ and $\theta$ with modal depth at most 1.

- Case $\varphi := \neg\psi$: Now, $(w^0, N_w^0) \Vdash \neg\psi$ iff $(w^0, N_w^0) \not\Vdash \psi$. By the induction hypothesis, this is equivalent to $G, w \not\models \psi^0$ which is equivalent to $G, w \models \neg\psi^0$.

- Case $\varphi := \psi \wedge \theta$: Now, $(w^0, N_w^0) \Vdash \psi \wedge \theta$ iff $(w^0, N_w^0) \Vdash \psi$ and $(w^0, N_w^0) \Vdash \theta$. By the induction hypothesis, this is equivalent to $G, w \models \psi^0$ and $G, w \models \theta^0$, which is equivalent to $G, w \models \psi^0 \wedge \theta^0$.

- Case $\varphi := \Diamond_{\geq \ell}\psi$: First it is easy to show that for every $(\Pi, \mathcal{T}')$-schema $\psi$ of modal depth 0 and for every state $q$ of $\mathcal{A}_\Gamma$ that $(q, \emptyset) \Vdash \psi$ iff $(q, N) \Vdash \psi$ for every multiset $N$ of states of $\mathcal{A}_\Gamma$. Now, $(w^0, N_w^0) \Vdash \Diamond_{\geq \ell}\psi$ iff there is a set $Q' \subseteq Q$ of states such that $\sum_{q' \in Q'} N_w^0(q') \geq \ell$ and $(q', \emptyset) \Vdash \psi$ for every $q' \in Q'$ iff there is a set $N$ of out-neighbours of $w$ such that $|N| \geq \ell$ and $(u^0, N_u^0) \Vdash \psi$ for every $u \in N$. By the induction hypothesis this is true iff there is a set $N$ of out-neighbours of $w$ such that $|N| \geq \ell$ and $G, u \models \psi^0$ for every $u \in N$, which is equivalent to $G, w \models \Diamond_{\geq \ell}\psi^0$.

Now, assume that the induction hypothesis holds for $n$ and let us prove the case for $n+1$. Similarly to the case $n = 0$, we prove by induction on structure of $\varphi$ that $(w^{n+1}, N_w^{n+1}) \Vdash \varphi$ iff $G, w \models \varphi^{n+1}$. The proof by induction is almost identical, except in the case where $\varphi = X \in \mathcal{T}'$. Let $\chi_X$ be the body of the iteration clause of $X$. Then $(w^{n+1}, N_w^{n+1}) \Vdash X$ iff $X \in w^{n+1}$ iff $(w^n, N_w^n) \Vdash \chi_X$. By the induction hypothesis, this is equivalent to $G, w \models \chi_X^n$, which is equivalent to $G, w \models X^{n+1}$.

Thus, the result above implies that for all pointed $\Pi$-labeled graphs $(G, w)$, the automaton $\mathcal{A}_\Gamma$ is in the state $q$ at $w$ in round $n$ iff $G, w \models X^n$ and $G, w \models p$ for every $X, p \in q$, and $G, w \not\models X^n$ and $G, w \not\models p$ for every $p, X \notin q$. Therefore, by the definition of the set of accepting states we see that $\mathcal{A}_\Gamma$ is equivalent to $\Gamma$. $\qquad \square$

The converse direction is easier. Note that the definitions for sizes of GMSC-programs and bounded FCMPAs can be found in Appendix A.1.

**Lemma B.3.** *For each bounded* FCMPA *over* $\Pi$*, we can construct an equivalent* $\Pi$*-program of* GMSC. *The size of the constructed* GMSC*-program is polynomial in the size of the* FCMPA.

*Proof.* The proof below is analogous to the proof of Theorem 1 in [24]. Let $\Pi$ be a set of node label symbols, and let $\mathcal{A} = (Q, \pi, \delta, F)$ be a $k$-FCMPA over $\Pi$. For each state $q \in Q$, we define a head predicate $X_q$ and the corresponding rules as follows. The terminal clause for $X_q$ is defined by

$$X_q(0) :- \bigvee_{P \subseteq \Pi,\ \pi(P)=q} \Big( \bigwedge_{p \in P} p \wedge \bigwedge_{p \in \Pi \setminus P} \neg p \Big).$$

To define the iteration clause for each $X_q$ we first define some auxiliary formulae. Given $q, q' \in Q$, we let $M_k(q, q') = \{ S \in \mathcal{M}_k(Q) \mid \delta(q, S) = q' \}$. Notice that the number of multisets $S$ specified by the set $M_k(q, q')$ is finite. Now, for each $S \in M_k(q, q')$ we define

$$\varphi_S := \bigwedge_{q \in Q,\ S(q)=n,\ n<k} \Diamond_{=n} X_q \wedge \bigwedge_{q \in Q,\ S(q)=k} \Diamond_{\geq k} X_q.$$

Then the iteration clause for $X_q$ is defined by $X_q :- \bigwedge_{q' \in Q} \Big( X_{q'} \rightarrow \bigvee_{S \in M_k(q',q)} \varphi_S \Big)$. It is easy to show by induction on $n \in \mathbb{N}$ that for every pointed $\Pi$-labeled graph $(G, w)$ it holds that $G, w \models X_q^n$ if and only if $w$ is in the state $q$ in round $n$ (see [24] for the details; the only difference is swapping sets for multisets).

The size of $\mathcal{A}$ is $|\mathcal{P}(\Pi)| + |Q|(k+1)^{|Q|}$ by definition. The size of all terminal clauses of the constructed program is altogether $|Q| + \mathcal{O}(|\Pi| \cdot |\mathcal{P}(\Pi)|)$, as there are $|Q|$ terminal clauses that altogether encode each element of $\mathcal{P}(\Pi)$ and each encoding is of size $\mathcal{O}(|\Pi|)$. The size of all iteration clauses of the constructed program is altogether $\mathcal{O}(|Q|^2 + k|Q|^2(k+1)^{|Q|}) = \mathcal{O}(k|Q|^2(k+1)^{|Q|})$, as there are $|Q|$ iteration clauses with $|Q|$ conjuncts each, and they altogether encode every element of $\mathcal{M}_k(Q)$ exactly $|Q|$ times and each encoding is of size $\mathcal{O}(k|Q|)$. Note that the cardinality of $\mathcal{M}_k(Q)$ is $(k+1)^{|Q|}$, i.e., the number of functions of type $Q \rightarrow [0; k]$. Thus, the size of the program is

$$|\Pi| + |Q| + \mathcal{O}(|\Pi| \cdot |\mathcal{P}(\Pi)|) + \mathcal{O}(k|Q|^2(k+1)^{|Q|}) = \mathcal{O}(|\Pi| \cdot |\mathcal{P}(\Pi)| + k|Q|^2(k+1)^{|Q|})$$

which is clearly less than $\mathcal{O}((|\mathcal{P}(\Pi)| + |Q|(k+1)^{|Q|})^2)$ and therefore polynomial in the size of $\mathcal{A}$. If head predicates and states were encoded in binary, the result would still be polynomial, as it would simply add a factor of $\log(|\Pi|)$ and $\log(|Q|)$ respectively to the sizes of terminal and iteration clauses. $\square$

*Proof of Proposition 3.1.* Note that the proof uses auxiliary results that are introduced in this subsection. Lemma B.2 shows that for every $\Pi$-program of GMSC we can construct an equivalent bounded FCMPA over $\Pi$. Lemma B.3 shows that for every bounded FCMPA over $\Pi$ we can construct an equivalent $\Pi$-program of GMSC. Thus, we conclude that GMSC has the same expressive power as bounded FCMPAs. $\square$

## B.3 Proof of Theorem 3.2

First we recall Theorem 3.2.

**Theorem 3.2.** *The following have the same expressive power:* GNN[F]*s,* GMSC, *and R-simple aggregate-combine* GNN[F]*s.*

In the end of this appendix section, we formally prove Theorem 3.2. Informally, this is done in the following steps. First, we note that it is easy to obtain an equivalent bounded FCMPA for a given GNN[F] and by Proposition 3.1 we can translate the bounded FCMPA into an equivalent GMSC-program. The converse direction is much more interesting. In Lemma B.5 we show that for each GMSC-program we can construct an equivalent R-simple aggregate-combine GNN[F].

Before we show how to obtain an equivalent R-simple GNN[F] for each GMSC-program, we first want to modify the programs so that they are easier to handle via R-simple GNN[F]s. In particular,

the terminal clauses of a program may involve diamonds, the rules of a program may have differing amounts of nested logical operators, and so may the two conjuncts of each conjunction. Thus we prove a lemma which intuitively shows how to "balance" GMSC-programs such that this is not the case. We let $\mathrm{fdepth}(\varphi)$ denote the formula depth of a schema $\varphi$ (see the definition of formula depth in Section 2.2). The formula depth of a terminal clause (resp., of an iteration clause) refers to the formula depth of the body of the clause. The formula depth of a GMSC-program is the maximum formula depth of its clauses.

**Lemma B.4.** *For each $\Pi$-program $\Lambda$ of* GMSC *with formula depth $D$, we can construct an equivalent $\Pi$-program $\Gamma$ of* GMSC *which has the following properties.*

1. *Each terminal clause is of the form $X(0) :- \bot$.*

2. *Each iteration clause has the same formula depth $\max(3, D + 2)$.*

3. *If $\varphi \wedge \theta$ is a subschema of $\Lambda$ such that neither $\varphi$ nor $\theta$ is $\top$, then $\varphi$ and $\theta$ have the same formula depth.*

*Proof.* First we define a fresh auxiliary predicate $I$ with the rules $I(0) :- \bot$ and $I :- \top$. Informally, we will modify the rules of $\Lambda$ such that the terminal clauses are simulated by the iteration clauses with the help of $I$. Then for each head predicate of $\Lambda$ with terminal clause $X(0) :- \varphi$ and iteration clause $X :- \psi$ we define new rules in $\Gamma$ as follows: $X(0) :- \bot$ and $X :- (\neg I \wedge \varphi) \vee (I \wedge \psi)$. Now, the formula depth of $\Gamma$ is $\max(3, D + 2)$.

First, we "balance" the formula depth of each iteration clause in $\Gamma$ according to the second point in the statement of the lemma. Let $d$ be the formula depth of an iteration clause $X :- \psi$, let $D' = \max(3, D + 2)$ and let $n = D' - d$. If $n$ is even, then the new iteration clause for $X$ is $X :- (\neg)^n \psi$, where $(\neg)^n = \neg \cdots \neg$ denotes $n$ nested negations. If $n$ is odd, then the new iteration clause for $X$ is $X :- (\neg)^{n-1}(\psi \wedge \top)$. In either case, the new iteration clause for $X$ has formula depth $D'$.

Then we show how to "balance" each subschema of $\Gamma$ according to the third point in the statement of the lemma as follows. Let $\varphi \wedge \psi$ be a subschema of $\Gamma$ such that neither $\varphi$ nor $\psi$ is $\top$, and w.l.o.g. assume that $\mathrm{fdepth}(\varphi) < \mathrm{fdepth}(\psi)$ and let $n = \mathrm{fdepth}(\psi) - \mathrm{fdepth}(\varphi)$. If $n$ is even, then we replace $\varphi \wedge \psi$ in $\Gamma$ with $(\neg)^n \varphi \wedge \psi$. If $n$ is odd, then we replace $\varphi \wedge \psi$ in $\Gamma$ with $(\neg)^{n-1}(\varphi \wedge \top) \wedge \psi$.

We have now obtained the desired $\Gamma$. It is easy to see that $\Gamma$ is equivalent to $\Lambda$ as follows. If $X$ is a head predicate that appears in both $\Lambda$ and $\Gamma$, then for each $n \in \mathbb{N}$ and for each pointed $\Pi$-labeled graph, we have $G, w \models X^n$ w.r.t. $\Lambda$ iff $G, w \models X^{n+1}$ w.r.t. $\Gamma$. In $\Gamma$, the auxiliary predicate $I$ makes sure that each head predicate $X$ in $\Gamma$ that appears in $\Lambda$ simulates in round 1 the corresponding terminal clause of $X$ in $\Lambda$ and in subsequent rounds simulates the corresponding iteration clause of $X$ in $\Lambda$. Furthermore, for each head predicate $X$ in $\Gamma$ and for each pointed $\Pi$-labeled graph $(G, w)$, we have $G, w \not\models X^0$ w.r.t. $\Gamma$. Thus, $\Gamma$ and $\Lambda$ are equivalent. $\qquad\square$

Now we are ready to show the translation from GMSC to R-simple aggregate-combine GNN[F]s.

**Lemma B.5.** *For each $\Pi$-program of* GMSC *we can construct an equivalent R-simple aggregate-combine* GNN[F] *over $\Pi$.*

*Proof.* Let $\Lambda$ be a $\Pi$-program of GMSC. Informally, an equivalent R-simple aggregate-combine GNN[F] for $\Lambda$ is constructed as follows. First from $\Lambda$ we construct an equivalent $\Pi$-program $\Gamma$ of GMSC with Lemma B.4. Let $D$ be the formula depth of $\Gamma$.

Intuitively, we build an R-simple aggregate-combine GNN[F] $\mathcal{G}_\Gamma$ for $\Gamma$ which periodically computes a single iteration round of $\Gamma$ in $D + 1$ rounds. The feature vectors used by $\mathcal{G}_\Gamma$ are *binary* vectors $\mathbf{v} = \mathbf{uw}$ where:

- $\mathbf{u}$ has one bit per each (distinct) subschema of a body of an iteration clause in $\Gamma$ as well as for each head predicate, and $\mathbf{v}_1$ keeps track of their truth values, and

- $\mathbf{w}$ has $D + 1$ bits and it keeps track of the formula depth that is currently being evaluated.

Therefore, $\mathcal{G}_\Gamma$ is a GNN[F] over $(\Pi, N + D + 1)$, where $N$ is the number of (distinct) subschemata of the bodies of the iteration clauses of $\Gamma$ as well as head predicates, and $D$ is the maximum formula depth of the bodies of the iteration clauses. In order to be able to compute values, the floating-point system for $\mathcal{G}_\Gamma$ is chosen to be high enough. More precisely, we choose a floating-point system $S$ which express all integers from 0 at least up to $K_{\max}$, where $K_{\max}$ is the width of $\Gamma$. Note that 0s and 1s are also represented in the floating-point system $S$. Note that although feature vectors exist that have elements other than 1s and 0s, they are not used.

Next we define the functions $\pi$ and COM, and the set $F$ of accepting states for $\mathcal{G}_\Gamma$. (Note that AGG is just the sum in increasing order.) We assume an enumeration $\mathrm{SUB}(\Gamma) := (\varphi_1, \ldots, \varphi_N)$ of subschemata and head predicates in $\Gamma$ such that if $\varphi_k$ is a subschema of $\varphi_\ell$, then $k \leq \ell$. The initialization function $\pi$ of $\mathcal{G}_\Gamma$ with input $P \subseteq \Pi$ outputs a feature vector $\mathbf{v} \in \{0,1\}^{N+D+1}$, where the value of each component corresponding to a node label symbol is defined as follows: the component for $p$ is 1 iff $p \in P$. The other components are 0s (excluding the possible subschema $\top$ which is assigned 1, as well as the very last $(N + D + 1)$th bit which is also assigned 1).

Recall that $S$ is the floating-point system for $\mathcal{G}_\Gamma$. The combination function (as per the definition of R-simple GNNs) is $\mathrm{COM}(\mathbf{x}, \mathbf{y}) = \sigma(\mathbf{x} \cdot C + \mathbf{y} \cdot A + \mathbf{b})$ where $\sigma$ is the truncated ReLU (ReLU$^*$) defined by $\mathrm{ReLU}^*(x) = \min(\max(0, x), 1)$, $\mathbf{b} \in S^{N+D+1}$ and $C, A \in S^{(N+D+1)\times(N+D+1)}$, where $\mathbf{b}, C$ and $A$ are defined as follows. For $k, \ell \leq N + D + 1$, we let $C_{k,\ell}$ (resp. $A_{k,\ell}$) denote the element of $C$ (resp. $A$) at the $k$th row and $\ell$th column. Similarly for $\ell \leq N + D + 1$ and for any vector $\mathbf{v} \in S^{N+D+1}$ including $\mathbf{b}$, we let $\mathbf{v}_\ell$ denote the $\ell$th value of $\mathbf{v}$. Now, we define the top-left $N \times N$ submatrices of $C$ and $A$ and the first $N$ elements of $\mathbf{b}$ in the same way as Barceló et al. [5]. For all $\ell \leq N$ we define as follows.

- If $\varphi_\ell \in \Pi \cup \{\top\}$, then $C_{\ell,\ell} = 1$.

- If $\varphi_\ell$ is a head predicate $X$ with the iteration clause $X :\!- \varphi_k$, then $C_{k,\ell} = 1$.

- If $\varphi_\ell = \varphi_j \wedge \varphi_k$, then $C_{j,\ell} = C_{k,\ell} = 1$ and $\mathbf{b}_\ell = -1$.

- If $\varphi_\ell = \neg \varphi_k$, then $C_{k,\ell} = -1$ and $\mathbf{b}_\ell = 1$.

- If $\varphi_\ell = \Diamond_{\geq K} \varphi_k$, then $A_{k,\ell} = 1$ and $\mathbf{b}_\ell = -K + 1$.

Next, we define that the bottom-right $(D + 1) \times (D + 1)$ submatrix of $C$ is the $(D + 1) \times (D + 1)$ identity matrix, except that the rightmost column is moved to be the leftmost column. More formally, for all $N + 1 \leq \ell < N + D + 1$ we have $C_{\ell,\ell+1} = 1$ and also $C_{N+D+1,N+1} = 1$. Lastly, we define that all other elements in $C$, $A$ and $\mathbf{b}$ are 0s. Finally, we define that $\mathbf{v} \in F$ if and only if $\mathbf{v}_\ell = 1$ (i.e., the $\ell$th value of $\mathbf{v}$ is 1) for some appointed predicate $\varphi_\ell$ and also $\mathbf{v}_{N+D+1} = 1$.

Recall that the formula depth of $\Gamma$ is $D$. Let $\mathbf{v}(w)_i^t$ denote the value of the $i$th component of the feature vector in round $t$ at node $w$. It is easy to show by induction that for all $n \in \mathbb{N}$, for all pointed $\Pi$-labeled graphs $(G, w)$, and for every schema $\varphi_\ell$ in $\mathrm{SUB}(\Gamma)$ of formula depth $d$, we have

$$\mathbf{v}(w)_\ell^{n(D+1)+d} = 1 \text{ if } G, w \models \varphi_\ell^n \text{ and } \mathbf{v}(w)_\ell^{n(D+1)+d} = 0 \text{ if } G, w \not\models \varphi_\ell^n.$$

Most of the work is already done by Barceló et al. (see [5] for the details), but we go over the proof as there are some additional considerations related to recurrence.

For the base case, let $n = 0$. We prove the case by induction over the formula depth $d$ of $\varphi_\ell$. First, let $d = 0$.

- Case 1: $\varphi_\ell \in \Pi \cup \{\top\}$. Now $\mathbf{v}_\ell^0 = 1$ and $G, w \models \top^0$ if $\varphi_\ell = \top$. If $\varphi_\ell = p \in \Pi$, then by the definition of the initialization function $\pi$ we have $\mathbf{v}(w)_\ell^0 = 1$ iff $p \in \lambda(w)$ iff $G, w \models p^0$, and $\mathbf{v}(w)_\ell^0 = 0$ iff $p \notin \lambda(w)$ iff $G, w \not\models p^0$.

- Case 2: $\varphi_\ell = X$, where $X$ is a head predicate of $\Gamma$. Now $\mathbf{v}(w)_\ell^0 = 0$ due to the definition of the initialization function $\pi$, and $G, w \not\models X^0$ because each head predicate of $\Gamma$ has the terminal clause $\bot$.

Now, assume the claim holds for $n = 0$ for any formulae $\varphi_j, \varphi_k$ with formula depth $d - 1$. We show that the claim holds for $n = 0$ for formulae $\varphi_\ell$ of formula depth $d$.

- Case 3: $\varphi_\ell = \varphi_j \wedge \varphi_k$ for some $\varphi_j, \varphi_k$ (recall that $\Gamma$ only contains conjunctions where both conjuncts have the same formula depth if neither of them are $\top$). This means we have $C_{j,\ell} = C_{k,\ell} = 1$ and $\mathbf{b}_\ell = -1$. Moreover, $C_{m,\ell} = 0$ for all $m \neq j, k$ and $A_{m,\ell} = 0$ for all $m$. Now

$$\mathbf{v}(w)_\ell^d = \mathrm{ReLU}^* \left( \mathbf{v}(w)_j^{d-1} + \mathbf{v}(w)_k^{d-1} - 1 \right).$$

Thus $\mathbf{v}(w)_\ell^d = 1$ iff $\mathbf{v}(w)_j^{d-1} = \mathbf{v}(w)_k^{d-1} = 1$. By the induction hypothesis this is equivalent to $G, w \models \varphi_j^0$ and $G, w \models \varphi_k^0$ which is equivalent to $G, w \models (\varphi_j \wedge \varphi_k)^0$. Likewise, $\mathbf{v}(w)_\ell^d = 0$ iff $\mathbf{v}(w)_j^{d-1} = 0$ or $\mathbf{v}(w)_k^{d-1} = 0$. By the induction hypothesis this is equivalent to $G, w \not\models \varphi_j^0$ or $G, w \not\models \varphi_k^0$ which is equivalent to $G, w \not\models (\varphi_j \wedge \varphi_k)^0$.

- Case 4: $\varphi_\ell = \neg \varphi_k$ for some $\varphi_k$. This means we have $C_{k,\ell} = -1$ and $\mathbf{b}_\ell = 1$. Moreover, $C_{m,\ell} = 0$ for all $m \neq k$ and $A_{m,\ell} = 0$ for all $m$. Now

$$\mathbf{v}(w)_\ell^d = \mathrm{ReLU}^* \left( -\mathbf{v}(w)_k^{d-1} + 1 \right).$$

Thus $\mathbf{v}(w)_\ell^d = 1$ iff $\mathbf{v}(w)_k^{d-1} = 0$. By the induction hypothesis this is equivalent to $G, w \not\models \varphi_k^0$ which is further equivalent to $G, w \models (\neg \varphi_k)^0$. Likewise, $\mathbf{v}(w)_\ell^d = 0$ iff $\mathbf{v}(w)_k^{d-1} = 1$. By the induction hypothesis this is equivalent to $G, w \models \varphi_k^0$ which is equivalent to $G, w \not\models (\neg \varphi_k)^0$.

- Case 5: $\varphi_\ell = \Diamond_{\geq K} \varphi_k$ for some $\varphi_k$. This means we have $A_{k,\ell} = 1$ and $\mathbf{b}_\ell = -K + 1$. Moreover, $C_{m,\ell} = 0$ for all $m$ and $A_{m,\ell} = 0$ for all $m \neq k$. Now

$$\mathbf{v}(w)_\ell^d = \mathrm{ReLU}^* \left( \mathrm{SUM}_S \left( \{\{ \mathbf{v}(v)_k^{d-1} \mid (w,v) \in E \}\} \right) - K + 1 \right),$$

where $\mathrm{SUM}_S \colon \mathcal{M}(S) \to S$ is the sum of floating-point numbers in $S$ in increasing order (see Section 2.1 for more details) and $E$ is the set of edges of $G$. Thus $\mathbf{v}(w)_\ell^d = 1$ iff there are at least $K$ out-neighbours $v$ of $w$ such that $\mathbf{v}(v)_k^{d-1} = 1$. By the induction hypothesis, this is equivalent to there being at least $K$ out-neighbours $v$ of $w$ such that $G, v \models \varphi_k^0$ which is further equivalent to $G, w \models (\Diamond_{\geq K} \varphi_k)^0$. Likewise, $\mathbf{v}(w)_\ell^d = 0$ iff there are fewer than $K$ out-neighbours $v$ of $w$ such that $\mathbf{v}(v)_k^{d-1} = 1$. By the induction hypothesis, this is equivalent to there being fewer than $K$ out-neighbours $v$ of $w$ such that $G, v \models \varphi_k^0$ which is equivalent to $G, w \not\models (\Diamond_{\geq K} \varphi_k)^0$.

Next, assume the claim holds for $n$ for all formulae of any formula depth. We show that it also holds for $n + 1$. We once again prove the claim by structure of $\varphi_\ell$. Cases 3, 4 and 5 are handled analogously to how they were handled in the case $n = 0$, so we only consider cases 1 and 2.

- Case 1: $\varphi_\ell \in \Pi \cup \{\top\}$. This means we have $C_{\ell,\ell} = 1$ and $\mathbf{b}_\ell = 0$. Moreover, $C_{m,\ell} = 0$ for all $m \neq \ell$ and $A_{m,\ell} = 0$ for all $m$. Now

$$\mathbf{v}(w)_\ell^{(n+1)(D+1)} = \mathrm{ReLU}^* \left( \mathbf{v}(w)_\ell^{n(D+1)+D} \right).$$

Thus we see that $\mathbf{v}(w)_\ell^{(n+1)(D+1)} = \mathbf{v}(w)_\ell^{n(D+1)+D}$ and a trivial induction shows that also $\mathbf{v}(w)_\ell^{(n+1)(D+1)} = \mathbf{v}(w)_\ell^{n(D+1)}$. Now $\mathbf{v}(w)_\ell^{(n+1)(D+1)} = 1$ iff $\mathbf{v}(w)_\ell^{n(D+1)} = 1$. By the induction hypothesis this is equivalent to $G, w \models \varphi_\ell^n$ which is equivalent to $G, w \models \varphi_\ell^{n+1}$ because $\varphi_\ell^{n+1} = \varphi_\ell^n$. Likewise, $\mathbf{v}(w)_\ell^{(n+1)(D+1)} = 0$ iff $\mathbf{v}(w)_\ell^{n(D+1)} = 0$. By the induction hypothesis this is equivalent to $G, w \not\models \varphi_\ell^n$ which is equivalent to $G, w \not\models \varphi_\ell^{n+1}$.

- Case 2: $\varphi_\ell = X$, where $X$ is a head predicate of $\Gamma$ with the iteration clause $\varphi_k$ of formula depth $D$ (recall that in $\Gamma$, each iteration clause has formula depth $D$). This means we have $C_{k,\ell} = 1$ and $\mathbf{b}_\ell = 0$. Moreover, $C_{m,\ell} = 0$ for all $m \neq k$ and $A_{m,\ell} = 0$ for all $m$. Now

$$\mathbf{v}(w)_\ell^{(n+1)(D+1)} = \mathrm{ReLU}^* \left( \mathbf{v}(w)_k^{n(D+1)+D} \right).$$

Thus $\mathbf{v}(w)_\ell^{(n+1)(D+1)} = 1$ iff $\mathbf{v}(w)_k^{n(D+1)+D} = 1$. By the induction hypothesis this is equivalent to $G, w \models \varphi_k^n$ which is equivalent to $G, w \models X^{n+1}$. Likewise, $\mathbf{v}(w)_\ell^{(n+1)(D+1)} = 0$ iff $\mathbf{v}(w)_k^{n(D+1)+D} = 0$. By the induction hypothesis this is equivalent to $G, w \not\models \varphi_k^n$ which is equivalent to $G, w \not\models X^{n+1}$.

This concludes the induction.

We also know for all $n \geq 1$ and all $N + 1 \leq \ell < N + D + 1$ that $\mathbf{v}(w)^n_{\ell+1} = \mathbf{v}(w)^{n-1}_\ell$ and $\mathbf{v}(w)^n_{N+1} = \mathbf{v}(w)^{n-1}_{N+D+1}$. This is because $C_{\ell,\ell+1} = 1$, $C_{\ell',\ell+1} = 0$ for all $\ell' \neq \ell$ and $\mathbf{b}_{\ell+1} = 0$ and thus

$$\mathbf{v}(w)^n_{\ell+1} = \mathrm{ReLU}^* \left( \mathbf{v}(w)^{n-1}_\ell \right),$$

and also $C_{N+D+1,N+1} = 1$, $C_{\ell',N+1} = 0$ for all $\ell' \neq N + D + 1$ and $\mathbf{b}_{N+1} = 0$ and thus

$$\mathbf{v}(w)^n_{N+1} = \mathrm{ReLU}^* \left( \mathbf{v}(w)^{n-1}_{N+D+1} \right).$$

By the initialization this means for all $1 \leq \ell, \ell' \leq D + 1$ and all $n \in \mathbb{N}$ that $\mathbf{v}(w)^{n(D+1)+\ell'}_{N+\ell} = 1$ iff $\ell' = \ell$ and 0 otherwise. Specifically, this means that $\mathbf{v}(w)^{n(D+1)+\ell'}_{N+D+1} = 1$ iff $\ell' = D + 1$ and $\mathbf{v}(w)^{n(D+1)+\ell'}_{N+D+1} = 0$ otherwise.

Thus if $\varphi_\ell$ is an appointed predicate $X$ of $\Gamma$, then we know for all $n \in \mathbb{N}$ that $G, w \models X^n$ iff $\mathbf{v}(w)^{n(D+1)}_\ell = 1$ and we also know that $\mathbf{v}(w)^{n(D+1)}_{N+D+1} = 1$ and thereby $\mathbf{v}(w)^{n(D+1)} \in F$. Thus $\mathcal{G}_\Gamma$ is equivalent to $\Gamma$.

Since $\mathcal{G}_\Gamma$ is equivalent to $\Gamma$ which is equivalent to $\Lambda$, we are done. $\qquad\square$

We note that the proof of Lemma B.5 generalizes for other types of GNN[F]s, such as the type described below. Let $S = ((p, n, \beta), +, \cdot)$ be the floating-point system, where $p = 1$, $n = 1$ and $\beta = K_{\max} + 1$, where $K_{\max}$ is again the maximum width of the rules of $\Gamma$ in the proof above. We believe that the proof generalizes for other floating-point systems which can represent all non-negative integers up to $K_{\max}$. Recall that $K_{\max}$ is the width of $\Gamma$ in the proof of Lemma B.5. Consider a class of GNN[F]s over $S$ with dimension $2(N + D + 1)$ (twice that in the proof above) where the aggregation function is $\mathrm{SUM}_S \colon \mathcal{M}(S) \to S$ (the sum of floating-point numbers in $S$ in increasing order applied separately to each element of feature vectors) and whose combination function $\mathrm{COM} \colon S^{2(N+D+1)} \times S^{2(N+D+1)} \to S^{2(N+D+1)}$ is defined by

$$\mathrm{COM}\,(x, y) = \mathrm{ReLU}(x \cdot C + y \cdot A + \mathbf{b}),$$

where $C, A \in S^{2(N+D+1) \times 2(N+D+1)}$ are matrices, $\mathbf{b} \in S^{2(N+D+1)}$ is a bias vector and ReLU is the rectified linear unit defined by $\mathrm{ReLU}(x) := \max(0, x)$ as opposed to $\mathrm{ReLU}^*$. The last $2(D+1)$ elements of feature vectors function as they do above, counting the $2(D+1)$ steps required each time an iteration of $\Gamma$ is simulated (here the computation takes twice as long compared to above). The first $2N$ elements calculate the truth values of subformulae as before, but now each calculation takes two steps instead of one as each subschema is assigned two elements instead of one. The first of these elements calculates the truth value of the subschema as before. However, due to the use of ReLU instead of $\mathrm{ReLU}^*$ the value of the element might be more than 1 if the subschema is of the form $\Diamond_{\geq K} \varphi_k$. Thus, the second element normalizes the values by assigning a weight of $K_{\max}$ ensuring that each positive value becomes $K_{\max}$ and then assigning the bias $-K_{\max} + 1$ to bring them all down to 1. More formally, for each subschema $\varphi_\ell$, let $\ell$ be the first and $N + \ell$ the second element associated with $\varphi_\ell$. In the initialization step we define that if $\varphi_\ell \in P \subseteq \Pi$ or $\varphi_\ell = \top$, then $\pi(P)_{N+\ell} = 1$ and other elements of $\pi(P)$ are 0s. Now we define as follows for all $1 \leq \ell \leq N$.

- If $\varphi_\ell \in \Pi \cup \{\top\}$, then $C_{N+\ell,\ell} = 1$.
- If $\varphi_\ell$ is a head predicate $X$ with the iteration clause $X :- \varphi_k$, then $C_{N+k,\ell} = 1$.
- If $\varphi_\ell = \varphi_j \wedge \varphi_k$, then $C_{N+j,\ell} = C_{N+k,\ell} = 1$ and $\mathbf{b}_\ell = -1$.
- If $\varphi_\ell = \neg \varphi_k$, then $C_{N+k,\ell} = -1$ and $\mathbf{b}_\ell = 1$.
- If $\varphi_\ell = \Diamond_{\geq K} \varphi_k$, then $A_{N+k,\ell} = 1$ and $\mathbf{b}_\ell = -K + 1$.

In each of the above cases we also define that $C_{\ell,N+\ell} = K_{\max}$ and $\mathbf{b}_{N+\ell} = -K_{\max} + 1$. Lastly for all $2N + 1 \leq \ell < 2(N+D+1)$ we define (as before) that $C_{\ell,\ell+1} = 1$ and also $C_{2(N+D+1),2D+1} = 1$. All other elements of $C$, $A$ and $\mathbf{b}$ are 0s. The set $F$ of accepting feature vectors is defined such that $\mathbf{v} \in F$ iff $\mathbf{v}_{N+\ell} = 1$ for some appointed predicate $\varphi_\ell$ and also $\mathbf{v}_{2(N+D+1)} = 1$. It is then straightforward to prove for all $n \in \mathbb{N}$, for all pointed $\Pi$-labeled graphs $(G, w)$ and for every schema $\varphi_\ell \in \mathrm{SUB}(\Gamma)$ of formula depth $d$ that

$$\mathbf{v}(w)^{2n(D+1)+2d-1}_\ell \geq 1 \text{ and } \mathbf{v}(w)^{2n(D+1)+2d}_{N+\ell} = 1 \text{ if } G, w \models \varphi^n_\ell$$

and
$$\mathbf{v}(w)_\ell^{2n(D+1)+2d-1} = \mathbf{v}(w)_{N+\ell}^{2n(D+1)+2d} = 0 \text{ if } G, w \not\models \varphi_\ell^n.$$

The proof is similar to the above, but we also need to show that normalization works as intended, i.e., $\mathbf{v}(w)_\ell^{2n(D+1)+2d-1} \geq 1$ implies $\mathbf{v}(w)_{N+\ell}^{2n(D+1)+2d} = 1$ and $\mathbf{v}(w)_\ell^{2n(D+1)+2d-1} = 0$ implies $\mathbf{v}(w)_{N+\ell}^{2n(D+1)+2d} = 0$. To see this, observe that $C_{\ell,N+\ell} = K_{\max}$ and $\mathbf{b}_{N+\ell} = -K_{\max} + 1$ and also $C_{\ell',N+\ell} = 0$ for all $\ell' \neq \ell$ and $A_{\ell',N+\ell} = 0$ for all $\ell'$. Thus

$$\mathbf{v}(w)_{N+\ell}^{2n(D+1)+2d} = \mathrm{ReLU}\left( K_{\max} \cdot \mathbf{v}(w)_\ell^{2n(D+1)+2d-1} - K_{\max} + 1 \right),$$

which is 0 if $\mathbf{v}(w)_\ell^{2n(D+1)+2d-1} = 0$ and 1 if $\mathbf{v}(w)_\ell^{2n(D+1)+2d-1} \geq 1$ (because by the choice of $S$ we have $K_{\max} \cdot x = K_{\max}$ for all $x \in S$, $x \geq 1$).

Having proved Lemma B.5, we are now ready to conclude the main theorem of this appendix section expressed below.

*Proof of Theorem 3.2.* Note that the proof uses auxiliary results introduced in this subsection. Each GNN[F] is trivial to translate into an equivalent bounded FCMPA with linear blow-up in size by the fact that the definitions of GNN[F]s and bounded FCMPAs are almost identical. Lemma B.5 shows that each GMSC-program can be translated into an equivalent R-simple aggregate-combine GNN[F]. The translation from GNN[F]s to GMSC-programs causes only polynomial blow-up in size by Lemma B.3. □

## B.4 Proof of Theorem 3.3

Note that $k$-CMPAs are defined in Appendix B.1. First we recall Theorem 3.3.

**Theorem 3.3.** CMPA*s have the same expressive power as* $\omega$-GML.

The proof of Theorem 3.3 is in the end of this subsection but we need some auxiliary results first. We show in Lemma B.7 that we can construct an equivalent counting type automaton over $\Pi$ for each $\Pi$-formula of $\omega$-GML. Informally, to do this, we first define a $\Pi$-formula of GML called the "full graded type of modal depth $n$" for each pointed graph, which expresses all the local information of its neighbourhood up to depth $n$. We show in Proposition B.6 that for each $\omega$-GML-formula there is a logically equivalent disjunction of types. We also define counting type automata that compute the type of modal depth $n$ of each node in every round $n$. The accepting states of the resulting automaton are exactly those types that appear in the disjunction of types.

Then we show in Lemma B.9 that for each CMPA over $\Pi$ we can construct an equivalent $\Pi$-formula of $\omega$-GML. Informally, to do this we first prove in Lemma B.8 that two pointed graphs that satisfy the same full graded type of modal depth $n$ also have identical states in each round $\ell \leq n$ in each CMPA. For each $n \in \mathbb{N}$, we consider exactly those full graded types of modal depth $n$ which are satisfied by some pointed graph that is accepted by the automaton in round $n$. We obtain the desired $\omega$-GML-formula by taking the disjunction of all these types across all $n \in \mathbb{N}$.

By similar arguments, we also obtain Theorem B.10 which is analogous to Theorem 3.3 but restricted to bounded CMPAs and width-bounded $\omega$-GML, which involves defining analogous width-bounded concepts.

Now, we start formalizing the proof of Theorem 3.3. To show that for each $\omega$-GML-formula we can construct an equivalent CMPA, we need to define the concepts of graded $\Pi$-types, full graded $\Pi$-types and counting type automata. The graded $\Pi$-type of modal depth $n$ and width $k$ of a pointed $\Pi$-labeled graph $(G, w)$ is a $\Pi$-formula of GML that contains all the information from the $n$-neighbourhood of $w$ (according to outgoing edges), with the exception that at each distance from $w$ we can only distinguish between at most $k$ identical branches. The full graded $\Pi$-type of modal depth $n$ lifts this limitation and contains all the information from the neighbourhood of depth $n$ of $w$. A counting type automaton of width $k$ is a $k$-CMPA that calculates the graded $\Pi$-type of modal depth $n$ and width $k$ of a node in each round $n$. Likewise, we define counting type automata which calculate the full graded $\Pi$-type of modal depth $n$ of a node in each round $n$.

Let $\Pi$ be a set of node label symbols, let $(G, w)$ be a pointed $\Pi$-labeled graph and let $k, n \in \mathbb{N}$. The **graded $\Pi$-type of width** $k$ **and modal depth** $n$ of $(G, w)$ (denoted $\tau_{k,n}^{(G,w)}$) is defined recursively as

follows. For $n = 0$ we define that

$$\tau_{k,0}^{(G,w)} := \bigwedge_{p_i \in \lambda(w)} p_i \wedge \bigwedge_{p_i \notin \lambda(w)} \neg p_i.$$

Assume we have defined the graded $\Pi$-type of width $k$ and modal depth $n$ of all pointed $\Pi$-labeled graphs, and let $T_{k,n}$ denote the set of such types. The graded $\Pi$-type of $(G, w)$ of width $k$ and modal depth $n + 1$ is defined as follows:

$$\tau_{k,n+1}^{(G,w)} := \tau_{k,0}^{(G,w)} \wedge \bigwedge_{\ell=0}^{k-1} \{ \Diamond_{=\ell}\tau \mid \tau \in T_{k,n} \text{ and } G, w \models \Diamond_{=\ell}\tau \}$$
$$\wedge \{ \Diamond_{\geq k}\tau \mid \tau \in T_{k,n} \text{ and } G, w \models \Diamond_{\geq k}\tau \}.$$

Canonical bracketing and ordering is used to ensure that no two types are logically equivalent, and thus each pointed $\Pi$-labeled graph has exactly one graded $\Pi$-type of each modal depth and width.

A **counting type automaton of width** $k$ over $\Pi$ is a $k$-CMPA defined as follows. The set $Q$ of states is the set $\bigcup_{n \in \mathbb{N}} T_{k,n}$ of all graded $\Pi$-types of width $k$ (of any modal depth). The initialization function $\pi \colon \mathcal{P}(\Pi) \to Q$ is defined such that $\pi(P) = \tau_{k,0}^{(G,w)}$ where $(G, w)$ is any pointed $\Pi$-labeled graph satisfying exactly the node label symbols in $P \subseteq \Pi$. Let $N \colon T_{k,n} \to \mathbb{N}$ be a multiset of graded $\Pi$-types of width $k$ and modal depth $n$, and let $\tau$ be one such type (note that $T_{k,n}$ is finite for any $k, n \in \mathbb{N}$). Let $\tau_0$ be the unique type of modal depth $0$ in $T_{k,n}$ that does not contradict $\tau$. We define the transition function $\delta \colon Q \times \mathcal{M}(Q) \to Q$ such that

$$\delta(\tau, N) = \tau_0 \wedge \bigwedge_{\ell=0}^{k-1} \{ \Diamond_{=\ell}\sigma \mid N(\sigma) = \ell \} \wedge \{ \Diamond_{\geq k}\sigma \mid N(\sigma) \geq k \},$$

For other $N$ and $\tau$ that do not all share the same modal depth, we may define the transition as we please such that $\delta(q, N) = \delta(q, N_{|k})$ for all $q$ and $N$.

We similarly define the **full graded $\Pi$-type of modal depth** $n$ of a pointed $\Pi$-labeled graph $(G, w)$ (denoted by $\tau_n^{(G,w)}$) which contains all the local information of the $n$-depth neighbourhood of $(G, w)$ with no bound on width. For $n = 0$, we define that $\tau_0^{(G,w)} = \tau_{k,0}^{(G,w)}$ for any $k \in \mathbb{N}$. Assume that we have defined the full graded $\Pi$-type of modal depth $n$ of all pointed $\Pi$-labeled graphs, and let $T_n$ be the set of such full types. The full graded $\Pi$-type of modal depth $n + 1$ of $(G, w)$ is defined as follows:

$$\tau_{n+1}^{(G,w)} := \tau_0^{(G,w)} \wedge \bigwedge_{\ell \geq 1} \{ \Diamond_{=\ell}\tau \mid \tau \in T_n, (G, w) \models \Diamond_{=\ell}\tau \} \wedge \Diamond_{=|\mathcal{N}(w)|}\top,$$

where $\mathcal{N}(w)$ is the set of out-neighbours of $w$. The formula tells exactly how many out-neighbours of a node satisfy each full graded type of the previous modal depth; to keep the formulae finite (over finite graphs), instead of containing conjuncts $\Diamond_{=0}\tau$ it tells exactly how many out-neighbours a node has.

A **counting type automaton** over $\Pi$ is a CMPA defined as follows. The set of states is the set of all full graded $\Pi$-types. The initialization function $\pi \colon \mathcal{P}(\Pi) \to Q$ is defined such that $\pi(P) = \tau_0^{(G,w)}$ where $(G, w)$ is any pointed $\Pi$-labeled graph satisfying exactly the node label symbols in $P$. Let $N \colon T_n \to \mathbb{N}$ be a multiset of full graded $\Pi$-types of some modal depth $n$, and let $\tau$ be one such type. Let $\tau_0$ be the unique full type of modal depth $0$ in $T_n$ that does not contradict $\tau$. We define the transition function $\delta \colon Q \times \mathcal{M}(Q) \to Q$ such that

$$\delta(\tau, N) = \tau_0 \wedge \bigwedge_{\ell \geq 1} \{ \Diamond_{=\ell}\sigma \mid N(\sigma) = \ell \} \wedge \Diamond_{=|N|}\top,$$

For other $N$ and $\tau$ that do not all share the same modal depth, we may define the transition as we please.

We prove the following useful property.

**Proposition B.6.** *Each $\Pi$-formula $\varphi$ of modal depth $n$ and width $k$ of* GML *has*

1. *a logically equivalent countably infinite disjunction of full graded $\Pi$-types of modal depth $n$ and*

2. *a logically equivalent finite disjunction of graded $\Pi$-types of width $k$ and modal depth $n$.*

*Proof.* First we prove the case for graded types of width $k$, then we prove the case for full graded types. Let $T_{k,n}$, as above, denote the set of all graded $\Pi$-types of width $k$ and modal depth $n$. Let $\Phi = \{\, \tau \in T_{k,n} \mid \tau \models \varphi \,\}$ and $\neg\Phi = \{\, \tau \in T_{k,n} \mid \tau \models \neg\varphi \,\}$, and let $\bigvee \Phi$ denote the disjunction of the types in $\Phi$. Note that this disjunction is finite since the set $T_{k,n}$ is finite. Obviously we have that $\Phi \cap \neg\Phi = \emptyset$ and $\bigvee \Phi \models \varphi$. To show that $\varphi \models \bigvee \Phi$, it suffices to show that $\Phi \cup \neg\Phi = T_{k,n}$. Assume instead that $\tau \in T_{k,n} \setminus (\Phi \cup \neg\Phi)$. Then there exist pointed $\Pi$-labeled graphs $(G, w)$ and $(H, v)$ that satisfy $\tau$ such that $G, w \models \varphi$ and $H, v \models \neg\varphi$. Since $(G, w)$ and $(H, v)$ satisfy the same graded $\Pi$-type of modal depth $n$ and width $k$, there can be no $\Pi$-formula of GML of modal depth at most $n$ and width at most $k$ that distinguishes $(G, w)$ and $(H, v)$, but $\varphi$ is such a formula, which is a contradiction. Ergo, $\bigvee \Phi$ and $\varphi$ are logically equivalent.

In the case of full graded types we first observe that the set $T_n$ of full graded $\Pi$-types of modal depth $n$ is countable. Thus the sets $\Phi = \{\, \tau \in T_n \mid \tau \models \varphi \,\}$ and $\neg\Phi = \{\, \tau \in T_n \mid \tau \models \neg\varphi \,\}$ are also countable. Now, with the same proof as for graded types of width $k$, we can show that $\bigvee \Phi$ and $\varphi$ are logically equivalent. Therefore, $\varphi$ is also logically equivalent to a countably infinite disjunction of full graded $\Pi$-types of modal depth $n$. $\qquad\square$

Now, we are ready to show the translation from $\omega$-GML to CMPAs.

**Lemma B.7.** *For each $\Pi$-formula of $\omega$-GML, we can construct an equivalent counting type automaton over $\Pi$. If the formula has finite width $k$, we can also construct an equivalent counting type automaton of width $k$.*

*Proof.* Assume the class $\mathcal{K}$ of pointed $\Pi$-labeled graphs is expressed by the countable disjunction $\psi := \bigvee_{\varphi \in S} \varphi$ of $\Pi$-formulae $\varphi$ of GML.

First, we prove the case without the width bound. By Proposition B.6, each $\varphi \in S$ is logically equivalent with a countably infinite disjunction $\varphi^*$ of full graded $\Pi$-types of GML such that the modal depth of $\varphi^*$ is the same as the modal depth of $\varphi$. We define a counting type automaton $\mathcal{A}$ whose set $F$ of accepting states is the set of types that appear as disjuncts of $\varphi^*$ for any $\varphi \in S$. Now $(G, w) \in \mathcal{K}$ if and only if $G, w \models \psi$ if and only if $G, w \models \tau$ for some $\tau \in F$ if and only if the state of $(G, w)$ is $\tau$ in round $n$ in $\mathcal{A}$, where $n$ is the modal depth of $\tau$. Ergo, $\mathcal{A}$ accepts exactly the pointed $\Pi$-labeled graphs in $\mathcal{K}$.

The case for the width bound is analogous. First with Proposition B.6 we transform each disjunct $\varphi \in S$ to a finite disjunction $\varphi^+$ of graded $\Pi$-types with the same width and modal depth as $\varphi$. Then, for $\psi$ we construct an equivalent type automaton of width $k$, where the set of accepting states is the set of graded types of width $k$ that appear as disjuncts of $\varphi^+$ for any $\varphi \in S$. $\qquad\square$

Before we prove Lemma B.9 we prove another helpful (and quite obvious) lemma.

**Lemma B.8.** *Two pointed $\Pi$-labeled graphs $(G, w)$ and $(H, v)$ satisfy exactly the same full graded $\Pi$-type of modal depth $n$ (respectively, graded $\Pi$-type of width $k$ and modal depth $n$) if and only if they share the same state in each round up to $n$ for each CMPA (resp., each $k$-CMPA) over $\Pi$.*

*Proof.* We prove the claim by induction over $n$ without the width bound, since again the case for the width bound is analogous. Let $n = 0$. Two pointed $\Pi$-labeled graphs $(G, w)$ and $(H, v)$ share the same full graded $\Pi$-type of modal depth 0 if and only if they satisfy the exact same node label symbols if and only if each initialization function $\pi$ assigns them the same initial state.

Now assume the claim holds for $n$. Two pointed $\Pi$-labeled graphs $(G, w)$ and $(H, v)$ satisfy the same full graded $\Pi$-type of modal depth $n + 1$ if and only if **1)** they satisfy the same full graded $\Pi$-type of modal depth 0 and **2)** for each full graded $\Pi$-type $\tau$ of modal depth $n$, they have the same number of neighbors that satisfy $\tau$. Now **1)** holds if and only if $(G, w)$ and $(H, v)$ satisfy the same node label symbols, and by the induction hypothesis **2)** is equivalent to the neighbors of $(G, w)$ and $(H, v)$ sharing (pair-wise) the same state in each round (up to $n$) for each CMPA. This is equivalent to $(G, w)$ and $(H, v)$ satisfying the same full graded $\Pi$-type of modal depth 0, and receiving the

same multiset of states as messages in round $n$ in each CMPA. By the definition of the transition function, this is equivalent to $(G, w)$ and $(H, v)$ sharing the same state in round $n + 1$ for each CMPA. □

Now, we are able to show the translation from CMPAs to $\omega$-GML.

**Lemma B.9.** *For each* CMPA *(respectively, each $k$-CMPA) over* $\Pi$*, we can construct an equivalent* $\Pi$*-formula of* $\omega$*-GML (resp., of width $k$).*

*Proof.* We prove the claim without the width bound, since the case with the width bound is analogous. Assume that the class $\mathcal{K}$ of pointed $\Pi$-labeled graphs is expressed by the counting message passing automaton $\mathcal{A}$ over $\Pi$. Let $\mathcal{T}$ be the set of all full graded $\Pi$-types and let

$$\Phi = \{\, \tau_n^{(G,w)} \in \mathcal{T} \mid \mathcal{A} \text{ accepts the pointed } \Pi\text{-labeled graph } (G, w) \in \mathcal{K} \text{ in round } n \,\}.$$

We define the countable disjunction $\bigvee_{\tau \in \Phi} \tau$ and show that $G, w \models \bigvee_{\tau \in \Phi} \tau$ if and only if $\mathcal{A}$ accepts $(G, w)$. Note that $\Phi$ is countable since $\mathcal{T}$ is countable.

If $G, w \models \bigvee_{\tau \in \Phi} \tau$, then $G, w \models \tau_n^{(H,v)}$ for some pointed $\Pi$-labeled graph $(H, v)$ accepted by $\mathcal{A}$ in round $n$. This means that $(G, w)$ and $(H, v)$ satisfy the same full graded $\Pi$-type of modal depth $n$. By Lemma B.8, this means that $(G, w)$ and $(H, v)$ share the same state in $\mathcal{A}$ in each round $\ell \leq n$. Since $\mathcal{A}$ accepts $(H, v)$ in round $n$, $\mathcal{A}$ also accepts $(G, w)$ in round $n$. Conversely, if $\mathcal{A}$ accepts $(G, w)$ in round $n$, then $\tau_n^{(G,w)} \in \Phi$ and thus $G, w \models \bigvee_{\tau \in \Phi} \tau$. □

Finally, we prove Theorem 3.3.

*Proof of Theorem 3.3.* Note that the proof uses auxiliary results that are introduced in this subsection. Lemma B.7 shows that for each $\Pi$-formula of $\omega$-GML we can construct an equivalent CMPA over $\Pi$ and Lemma B.9 shows that for each CMPA over $\Pi$ we can construct an equivalent $\Pi$-formula of $\omega$-GML. □

By Lemma B.7 and Lemma B.9, it is also straightforward to conclude a similar result for bounded CMPAs and width-bounded $\omega$-GML-formulae.

**Theorem B.10.** *Bounded* CMPA*s have the same expressive power as width-bounded $\omega$-GML-formulae.*

## B.5 Proof of Theorem 3.4

First we recall Theorem 3.4.

**Theorem 3.4.** GNN$[\mathbb{R}]$*s have the same expressive power as $\omega$-GML.*

Informally, Theorem 3.4 is proved as follows (in the end of this section we give a formal proof). We first prove Lemma B.12 which shows that we can translate any GNN$[\mathbb{R}]$ into an equivalent $\omega$-GML-formula. To prove Lemma B.12, we need to first prove an auxiliary result Lemma B.11, which shows that if two pointed graphs satisfy the same full graded type of modal depth $n$, then those two pointed graphs share the same feature vector in each round (up to $n$) with any GNN$[\mathbb{R}]$. For these results, we need graded types which were introduced in Section B.4.

Informally, the converse direction (Lemma B.13) is proved as follows. We first translate the $\Pi$-formula of $\omega$-GML into an equivalent CMPA over $\Pi$ by Theorem 3.3. Then we translate the equivalent CMPA into an equivalent counting type automaton (again, see Section B.4 for the definition of counting type automata). Then we prove that we can construct an equivalent GNN$[\mathbb{R}]$ for each counting type automaton. Informally, we encode each state of the counting type automaton into an integer and the GNN$[\mathbb{R}]$ can use them to mimic the type automaton in every step.

In both directions, we also consider the case where GNN$[\mathbb{R}]$s are bounded and the $\omega$-GML-formulae are width-bounded.

Before proving Lemma B.12, we first establish the following useful lemma.

**Lemma B.11.** *Two pointed $\Pi$-labeled graphs $(G, w)$ and $(H, v)$ satisfy exactly the same full graded $\Pi$-type of modal depth $n$ (respectively the same graded $\Pi$-type of width $k$ and modal depth $n$) if and only if they share the same state in each round (up to $n$) for each unbounded $\text{GNN}[\mathbb{R}]$ (resp., $\text{GNN}[\mathbb{R}]$ with bound $k$) over $\Pi$.*

*Proof.* The proof is analogous to that of Lemma B.8.  □

With the above lemma, we are ready to prove Lemma B.12.

**Lemma B.12.** *For each $\text{GNN}[\mathbb{R}]$ $\mathcal{G}$ over $\Pi$, we can construct an equivalent $\Pi$-formula of $\omega$-GML. Moreover, if $\mathcal{G}$ is bounded with the bound $k$, we can construct an equivalent $\Pi$-formula of $\omega$-GML of width $k$.*

*Proof.* Assume that $\mathcal{G}$ is a $\text{GNN}[\mathbb{R}]$ over $(\Pi, d)$. Let $\mathcal{K}$ be the class of pointed $\Pi$-labeled graphs expressed by $\mathcal{G}$. Now let

$$\Phi = \{\, \tau_n^{(G,w)} \mid \mathcal{G} \text{ accepts the pointed } \Pi\text{-labeled graph } (G, w) \in \mathcal{K} \text{ in round } n \,\},$$

where $\tau_n^{(G,w)}$ is the full graded $\Pi$-type of modal depth $n$ of $(G, w)$ (see Section B.4). Note that there are only countably many formulae in $\Phi$ since there are only countably many full graded $\Pi$-types. Consider the counting type automaton $\mathcal{A}$ over $\Pi$, where $\Phi$ is the set of accepting states. We will show that $\mathcal{A}$ accepts $(G, w)$ if and only if $\mathcal{G}$ accepts $(G, w)$.

If $(G, w)$ is accepted by $\mathcal{A}$, then $G, w \models \tau_n^{(H,v)}$ for some pointed $\Pi$-labeled graph $(H, v)$ accepted by $\mathcal{G}$ in round $n$. This means that $(G, w)$ and $(H, v)$ satisfy the same full graded $\Pi$-type of modal depth $n$. By Lemma B.11, this means that $(G, w)$ and $(H, v)$ share the same state in $\mathcal{G}$ in each round $\ell \le n$. Since $\mathcal{G}$ accepts $(H, v)$ in round $n$, $\mathcal{G}$ also accepts $(G, w)$ in round $n$. Conversely, if $\mathcal{G}$ accepts $(G, w)$ in round $n$, then $\tau_n^{(G,w)} \in \Phi$ and thus $(G, w)$ is accepted by $\mathcal{A}$ by the definition of counting type automata. Thus $\mathcal{A}$ and $\mathcal{G}$ are equivalent. By Theorem 3.3 we can translate the type automaton $\mathcal{A}$ into an equivalent $\Pi$-formula of $\omega$-GML. Therefore, for $\mathcal{G}$ we can obtain an equivalent $\omega$-GML-formula.

Next consider the case where $\mathcal{G}$ is bounded. We follow the same steps as above with the following modification: instead of constructing a set of full graded types, we construct a set of graded types of width $k$: $\Phi = \{\, \tau_{k,n}^{(G,w)} \mid \mathcal{G} \text{ accepts the pointed } \Pi\text{-labeled graph } (G, w) \in \mathcal{K} \text{ in round } n \,\}$. The reasoning of the second paragraph is then modified to refer to graded types of width $k$ and GNNs with the bound $k$ respectively using Lemma B.11. By Theorem B.10 we obtain an equivalent width-bounded $\omega$-GML-formula for the counting type automaton of width $k$.  □

We then show the other direction of Theorem 3.4 in the next lemma. We first define a graph neural network model that is used as a tool in the proof that follows. A **recurrent graph neural network over natural numbers** $\text{GNN}[\mathbb{N}]$ over $(\Pi, d)$, is a $\text{GNN}[\mathbb{R}]$ over $(\Pi, d)$ where the feature vectors and the domains and co-domains of the functions are restricted to $\mathbb{N}^d$ instead of $\mathbb{R}^d$.

**Lemma B.13.** *For each $\Pi$-formula of $\omega$-GML, we can construct an equivalent $\text{GNN}[\mathbb{R}]$ over $(\Pi, 1)$. Moreover, for each $\Pi$-formula of $\omega$-GML of width $k$, we can construct an equivalent bounded $\text{GNN}[\mathbb{R}]$ over $(\Pi, 1)$ with the bound $k$.*

*Proof.* We first give an informal description, and then we give a formal proof. We show that each counting type automaton over $\Pi$ can be translated into an equivalent $\text{GNN}[\mathbb{N}]$ over $(\Pi, 1)$, which suffices since counting type automata have the same expressive power as $\omega$-GML and CMPAs by Lemma B.9 and by Lemma B.7. Informally, each counting type automaton can be simulated by a graph neural network as follows. Each full type has a minimal tree graph that satisfies the full type. Each such tree can be encoded into a binary string (or more precisely into an integer) in a standard way. These binary strings are essentially used to simulate the computation of the counting type automaton. At each node and in each round, the $\text{GNN}[\mathbb{N}]$ simulates the counting type automaton by combining the multiset of binary strings obtained from its out-neighbours and the local binary string into a new binary string that corresponds to the minimal tree that satisfies the type that would be obtained by the counting type automaton in the same round at the same node.

Now, we formally prove the statement. Let $\mathcal{A}$ be a counting type automaton. Now we create an equivalent GNN[$\mathbb{N}$] $\mathcal{G}$ over $(\Pi, 1)$ as follows. Each full graded type is converted into the unique finite rooted tree graph $(\mathcal{T}, r)$ of the same depth as the modal depth of the full type. This graph is encoded into a binary string (in a standard way) as follows.

- The first $n$ bits of the string are 1s, telling the number of nodes in the tree; we choose some ordering $v_1, \ldots, v_n$ for the nodes. (A natural ordering would be to start with the root, then list its children, then its grandchildren, etc.. The children of each node can be ordered in increasing order of magnitude according to the encodings of their generated subtrees. Another option is to assume that the domain of the tree is always $[n]$ for some $n \in \mathbb{N}$ and use the standard ordering of integers.) This is followed by a 0.

- The next $nk$ bits are a bit string $\mathbf{b}$ that tells which node label symbol is true in which node; for each $\ell \in [n]$ and $i \in [k]$, the $((\ell - 1)k + i)$:th bit of $\mathbf{b}$ is 1 if and only if $\mathcal{T}, v_\ell \models p_i$.

- Finally, the last $n^2$ bits form a bit string $\mathbf{b}'$, where for each $i, j \in [n]$, the $((i-1)n + j)$:th bit of $\mathbf{b}'$ is 1 if and only if $(v_i, v_j) \in E$, i.e., there is an edge from $v_i$ to $v_j$.

The GNN $\mathcal{G}$ operates on these binary strings (we can either use the integers encoded in binary by the binary strings, or interpret the binary strings as decimal strings). In round 0, the initialization function maps each set of node label symbols to the encoding of the unique one-node tree graph where the node satisfies exactly the node label symbols in question. In each subsequent round $n$, each node receives a multiset $N$ of binary encodings of full graded types of modal depth $n - 1$. The aggregation function calculates a binary string that encodes the following graph:

- The graph contains a node $r$ where every node label symbol is false.

- For each copy of each element in the multiset $N$, the graph contains a unique copy of the rooted tree encoded by that element.

- There is an edge from $r$ to the root of each of these rooted trees; the result is itself a rooted tree where $r$ is the root.

The combination function receives a binary string corresponding to the node's full graded type $\tau$ of modal depth $n - 1$, as well as the string constructed above. It takes the above string and modifies the bits corresponding to the node label symbols at the root; it changes them to be identical to how they were in the root in $\tau$. The obtained binary encoding is the feature of the node in round $n$.

It is clear that the constructed GNN[$\mathbb{N}$] $\mathcal{G}$ calculates a node's full graded type of modal depth $n$ in round $n$. This is because our construction is an embedding from full graded types to natural numbers. Any other such embedding would also suffice, as the images of the aggregation and combination function are always full graded types and GNN[$\mathbb{R}$]s do not limit the aggregation and combination functions. We can choose the accepting states to be the numbers that encode the accepting states in the corresponding counting type automaton $\mathcal{A}$, in which case the GNN accepts exactly the same pointed graphs as $\mathcal{A}$.

Now consider the case where $\mathcal{A}$ is a counting type automaton of width $k$. We follow similar steps as above with the following modifications.

- We convert each graded type of width $k$ and modal depth $n$ to a corresponding rooted tree. We choose the tree with depth $n$ and the least amount of branches; in other words, each node has at most $k$ identical out-neighbours.

- The aggregation function of $\mathcal{G}$ constructs a graph in the same way as above, except that it creates at most $k$ copies of a graph encoded by an element in the multiset.

It is clear that the resulting GNN is bounded, as the aggregation function is bounded by $k$.

We have now shown that for each counting type automaton we can construct an equivalent GNN[$\mathbb{R}$] (or more precisely an equivalent GNN[$\mathbb{N}$]) and respectively for each counting type automaton of width $k$ we can construct an equivalent bounded GNN[$\mathbb{R}$] (or more precisely an equivalent bounded

GNN[$\mathbb{N}$]). Therefore, by Lemma B.9 and by Lemma B.7 for each formula of $\omega$-GML (and respectively for each CMPA) we can construct an equivalent GNN[$\mathbb{R}$]. Analogously, for each width-bounded formula of $\omega$-GML (and resp., for each bounded CMPA) we can construct an equivalent bounded GNN[$\mathbb{R}$]. □

We are now ready to formally prove Theorem 3.4.

*Proof of Theorem 3.4.* Note that the proof uses auxiliary results that are introduced in this subsection. By Lemma B.12 we can construct an equivalent $\Pi$-formula of $\omega$-GML for each GNN[$\mathbb{R}$] over $\Pi$. By Lemma B.13 we can construct an equivalent GNN[$\mathbb{R}$] over $\Pi$ for each $\Pi$-formula of $\omega$-GML. □

## B.6 Proof of Remark 3.5

In this section we formally prove the claims in Remark 3.5, expressed in Theorems B.15 and B.16. Note that we do not require boundedness for FCMPAs in the statement of the lemma below.

First we prove an auxiliary result.

**Lemma B.14.** *For each* FCMPA *over* $\Pi$ *with state set $Q$, we can construct an equivalent unrestricted* GNN[F] *over* $(\Pi, |Q|^2)$.

*Proof.* We encode the FCMPA fully into the GNN[F]. First we give an informal description. The feature vectors encode which state a node is in using one-hot encoding, i.e., only the bit corresponding to the occupied state is 1 and others are 0. For a received multiset $M$ of feature vectors, the aggregation function encodes the function $\delta_M \colon Q \to Q, \delta_M(q) = \delta(q, M)$ into a vector, i.e., it tells for each state $q_i$ which state $q_j$ satisfies $\delta(q_i, M) = q_j$. To encode such a function we need $|Q|^2$ components in the feature vector. Finally, the combination function receives (the encodings of) $q$ and $\delta_M$ and computes (the encoding of) $\delta_M(q)$.

Now we define the construction formally. Let $\Pi$ be a set of node label symbols. Let $\mathcal{A} = (Q, \pi, \delta, F)$ be an FCMPA over $\Pi$. Assume some arbitrary ordering $<_Q$ between the states $Q$. Let $q_1, \dots, q_{|Q|}$ enumerate the states of $Q$ w.r.t. $<_Q$. Given a multiset $M \in \mathcal{M}(Q)$ the function $\delta_M \colon Q \to Q$, $\delta_M(q) = \delta(q, M)$ specified by $M$ is possible to encode to the binary string $\mathbf{d}_M \in \{0, 1\}^{|Q|^2}$ as follows. If $\delta_M(q_i) = q_j$, then the $((i - 1)|Q| + j)$:th bit of $\mathbf{d}_M$ is 1; the other bits are 0. That is, $\mathbf{d}_M$ encodes the function $\delta_M$ in binary.

We construct a GNN over $(\Pi, |Q|^2)$ over a floating-point system $S$ that includes at least 0 and 1 (that is to say, small floating-point systems suffice). For all $i \le |Q|$, we let $\mathbf{d}_i \in S^{|Q|^2}$ denote the one-hot string where exactly the $i$th bit is 1 and the other bits are 0s.

- Let $P \subseteq \Pi$. The initialization function $\pi'$ is defined as $\pi'(P) = \mathbf{d}_i$ where $\pi(P) = q_i$.

- The aggregation function AGG is defined as follows. Assume that the multiset $M$ contains only one-hot strings $\mathbf{d}_i$, where $i \le |Q|$. Now $M$ corresponds to a multiset $M^*$, where each $\mathbf{d}_i$ is replaced by $q_i$. Then AGG($M$) is $\mathbf{d}_{M^*}$. Otherwise, the aggregation function is defined in an arbitrary way.

- The combination function COM is defined as follows. Let $\mathbf{d}, \mathbf{d}' \in S^{|Q|^2}$ such that $\mathbf{d} = \mathbf{d}_i$ is a one-hot string for some $i \le |Q|$ and $\mathbf{d}' = \mathbf{d}_M$ for some multiset $M \in \mathcal{M}(Q)$. Then we define that COM($\mathbf{d}, \mathbf{d}'$) = COM($\mathbf{d}_i, \mathbf{d}_M$) = $\mathbf{d}_j$ if and only if $\delta_M(q_i) = q_j$. Otherwise COM is defined in an arbitrary way.

- The set $F'$ of accepting states is defined as follows. If $\mathbf{d} = \mathbf{d}_j$ is a one-hot string, then $\mathbf{d}_j \in F'$ if and only if $q_j \in F$. Otherwise $F'$ is defined in an arbitrary way.

It is easy to show by induction over $n \in \mathbb{N}$ that for any pointed $\Pi$-labeled graph $(G, w)$, the state of $\mathcal{A}$ at $w$ in round $n$ is $q_i \in Q$ if and only if the state of $\mathcal{G}$ at $w$ in round $n$ is $\mathbf{d}_i$.

We note that this proof is based only on providing two embeddings to $S^d$ where $S$ is the floating-point system and $d$ is the dimension of the GNN[F]. The first is an embedding from the set $Q$

and the second is an embedding from the set of functions $Q \to Q$, and the two embeddings need not be related in any way. Thus, any choice of floating-point system $S$ and dimension $d$, such that there are at least $|Q|^2$ expressible feature vectors, would suffice and we would be able to define such embeddings and the subsequent aggregation and combination functions. In particular, dimension $|Q|^2$ suffices for any floating-point system $S$. $\qquad\square$

The first claim in Remark 3.5 is stated as follows.

**Theorem B.15.** FCMPA*s have the same expressive power as unrestricted* GNN[F]*s.*

*Proof.* It is straightforward to translate an unrestricted GNN[F] into an equivalent FCMPA. By Lemma B.14 we can also construct an equivalent unrestricted GNN[F] for each FCMPA. $\qquad\square$

The second claim in Remark 3.5 is stated as follows.

**Theorem B.16.** *Bounded* GNN[$\mathbb{R}$]*s have the same expressive power as width-bounded* $\omega$-GML.

*Proof.* Note that this proof uses auxiliary results that are in Appendix B.5. By Lemma B.12 we can construct an equivalent width-bounded $\Pi$-formula of $\omega$-GML for each bounded GNN[$\mathbb{R}$] over $\Pi$. By Lemma B.13 we can construct an equivalent bounded GNN[$\mathbb{R}$] over $\Pi$ for each width-bounded $\Pi$-formula of $\omega$-GML. $\qquad\square$

## B.7 Proof that multiple GNN layers can be simulated using a single layer

In the literature, GNNs are often defined as running for a constant number of iterations unlike our recurrent GNN model, see for example [5, 13, 12]. Each iteration of the GNN is considered its own layer, and each layer has its own aggregation and combination function. More formally for any $N \in \mathbb{N}$, an $N$-**layer** GNN[$\mathbb{R}$] $\mathcal{G}_N$ over $(\Pi, d)$ is a tuple $(\mathbb{R}^d, \pi, (\delta^i)_{i \in [N]}, F)$, where $\pi \colon \mathcal{P}(\Pi) \to \mathbb{R}^d$ is the *initialization function*, $\delta^i \colon \mathbb{R}^d \times \mathcal{M}(\mathbb{R}^d) \to \mathbb{R}^d$ is the **transition function of layer** $i$ of the form $\delta^{(i)}(q, M) = \mathrm{COM}^i(q, \mathrm{AGG}^i(M))$ (where $\mathrm{AGG}^i \colon \mathcal{M}(\mathbb{R}^d) \to \mathbb{R}^d$ is the **aggregation function of layer** $i$ and $\mathrm{COM}^i \colon \mathbb{R}^d \times \mathbb{R}^d \to \mathbb{R}^d$ is the **combination function of layer** $i$) and $F \subseteq \mathbb{R}^d$ is the set of *accepting feature vectors*. We define the computation of $\mathcal{G}_N$ in a $\Pi$-labeled graph $G = (V, E, \lambda)$ as follows. In round 0, the feature vector of a node $v$ is $x_v^0 = \pi(\lambda(v))$. In round $i \in [N]$, the feature vector of a node is

$$x_v^i = \delta^i(x_v^{i-1}, \{\{ x_u^{i-1} \mid (v, u) \in E \}\}) = \mathrm{COM}^i(x_v^{i-1}, \mathrm{AGG}^i(\{\{ x_u^{i-1} \mid (v, u) \in E \}\})).$$

We say that $\mathcal{G}_N$ **accepts** a pointed graph $(G, w)$ if and only if $x_w^N \in F$. Concepts concerning node properties, equivalence and same expressive power are defined as for other models of GNNs.

**Proposition B.17.** *For each $N$-layer* GNN[$\mathbb{R}$]*, we can construct an equivalent constant-iteration* GNN[$\mathbb{R}$].

*Proof.* Intuitively, we add a clock to the feature vectors of the $N$-layer GNN[$\mathbb{R}$] that tells the transition function of the constant-iteration GNN[$\mathbb{R}$] which layer to simulate.

Let $\mathcal{G}_N = (\mathbb{R}^d, \pi, (\delta^i)_{i \in [N]}, F)$ be an $N$-layer GNN[$\mathbb{R}$] over $(\Pi, d)$. We construct a constant-iteration GNN[$\mathbb{R}$] $(\mathcal{G}, N) = ((\mathbb{R}^{d+N}, \pi', \delta', F'), N)$ over $(\Pi, d + N)$ as follows. In all feature vectors used, exactly one of the last $N$ elements is 1 and the others are 0s. For the initialization function $\pi'$ we define for all $P \subseteq \Pi$ that $\pi'(P) = (\pi(P)_1, \dots, \pi(P)_d, 1, 0, \dots, 0) \in \mathbb{R}^{d+N}$, where $\pi(P)_i$ denotes the $i$th element of $\pi(P) \in \mathbb{R}^d$. Before specifying the transition function, we define the following:

- For each feature vector $x = (x_1, \dots, x_{d+N}) \in \mathbb{R}^{d+N}$, we let $x' = (x_1, \dots, x_d)$ and $x'' = (x_{d+1}, \dots, x_{d+N})$.

- Likewise, for each $M \in \mathcal{M}(\mathbb{R}^{d+N})$, let $M'$ be $M$ where each $x \in M$ is replaced with $x'$.

- Let $f \colon \mathbb{R}^N \to \mathbb{R}^N$ be a function such that we have $f(1, 0, \dots, 0) = (0, 1, 0, \dots, 0)$, $f(0, 1, 0, \dots, 0) = (0, 0, 1, 0, \dots, 0)$, and so forth until $f(0, \dots, 0, 1) = (0, \dots, 0, 1)$.

Now, assuming that $x \in \mathbb{R}^{d+N}$ and $M \in \mathcal{M}(\mathbb{R}^{d+N})$ such that exactly the $i$th element of $x''$ and each $y''$ in $M$ is 1 and others are 0s, we define that $\delta'(x, M) = (\delta^i(x', M'), f(x''))$ (i.e., we concatenate $\delta^i(x', M')$ and $f(x'')$). For other inputs, we define $\delta'$ arbitrarily. The set $F'$ of accepting feature vectors is the set of feature vectors $(x_1, \ldots, x_d, 0, \ldots, 0, 1)$, where $(x_1, \ldots, x_d)$ is an accepting feature vector of $\mathcal{G}_N$. It is easy to show that $(\mathcal{G}, N)$ accepts a pointed $\Pi$-labeled graph $(G, w)$ if and only if $\mathcal{G}_N$ accepts $(G, w)$ (note that for both $N$-layer GNN[$\mathbb{R}$]s and constant-iteration GNN[$\mathbb{R}$]s, only the feature vector of a node in round $N$ counts for acceptance). $\qquad\square$

## B.8 Proof of Theorem 3.6

First we recall Theorem 3.6.

**Theorem 3.6.** *Constant-iteration GNN[$\mathbb{R}$]s have the same expressive power as depth-bounded $\omega$-GML.*

*Proof.* Note that we *heavily* use the proofs of Lemma B.12 and Lemma B.13.

Assume that $(\mathcal{G}, N)$ is a constant-iteration GNN[$\mathbb{R}$] over $(\Pi, d)$. Let $\mathcal{K}$ be the class of pointed $\Pi$-labeled graphs expressed by $(\mathcal{G}, N)$. Now let

$$\Phi = \{\, \tau_N^{(G,w)} \mid \mathcal{G} \text{ accepts } (G, w) \in \mathcal{K} \text{ in round } N \,\}$$

where $\tau_N^{(G,w)}$ is the full graded $\Pi$-type of modal depth $N$ of $(G, w)$ (see Section B.4). Consider the counting type automaton $\mathcal{A}$ over $\Pi$, where $\Phi$ is the set of accepting states. It is easy to show with an analogous argument as in the proof of Lemma B.12 that $\mathcal{A}$ accepts $(G, w)$ if and only if $(\mathcal{G}, N)$ accepts $(G, w)$.

For the converse, assume that $\psi$ is a depth-bounded $\omega$-GML-formula over $\Pi$ of modal depth $D$. First by Lemma B.6 we translate $\psi$ into an equivalent formula $\psi^*$ which is a disjunction of full graded $\Pi$-types such that the modal depth of $\psi^*$ is the same as $\psi$. By Lemma B.7, $\psi^*$ is equivalent to a counting type automaton $\mathcal{A}$ over $\Pi$ whose accepting states are the types that appear as disjuncts of $\psi^*$. Since the depth of each type is bounded by $D$, each pointed graph accepted by $\mathcal{A}$ is accepted in some round $r \leq D$. By the proof of Lemma B.13 we can construct an equivalent GNN[$\mathbb{R}$] $\mathcal{G}$ over $\Pi$ such that for all pointed $\Pi$-labeled graphs $(G, w)$ and for all $n \in \mathbb{N}$: $\mathcal{A}$ accepts $(G, w)$ in round $n$ iff $\mathcal{G}$ accepts $(G, w)$ in round $n$. Now, it is easy to modify $\mathcal{G}$ such that if $\mathcal{G}$ accepts a pointed graph in round $m$ then it also accepts that pointed graph in every round $m' > m$. Therefore, for all pointed graphs $(G, w)$ we have that $(\mathcal{G}, D)$ accepts $(G, w)$ iff $G, w \models \psi^*$ iff $G, w \models \psi$. $\qquad\square$

# C Appendix: Characterizing GNNs over MSO-expressible properties

## C.1 Proof of Lemma 4.2

In the end of this subsection we give the formal proof of Lemma 4.2. First we give some preliminary definitions.

Let $G = (V, E, \lambda)$ be a graph and let $w_0 \in V$ be a node in $G$. A **walk in** $G$ **starting at** $w_0$ is a sequence $p = w_0, \ldots, w_n$ of elements of $V$ such that $(w_i, w_{i+1}) \in E$ for all $i \leq n - 1$. We use $\mathsf{tail}(p)$ to denote $w_n$. Now, the **unraveling** of $G$ at $w_0$ is the graph $U = (V', E', \lambda')$ defined as follows:

$$V' = \text{the set of all walks in } G \text{ starting at } w_0$$
$$E' = \{\, (p, p') \in V' \times V' \mid p' = pw \text{ for some } w \in V \,\}$$
$$\lambda'(p) = \lambda(\mathsf{tail}(p)) \text{ for all } p \in V'.$$

We say that a formula $\varphi(x)$ is **invariant under unraveling** if for every graph $G = (V, E, \lambda)$ and every $w \in V$, we have $G \models \varphi(w)$ iff $U \models \varphi(w)$, with $U$ the unraveling of $G$ at $w$. Invariance under unraveling is defined in the same way also for GNNs. The following is easy to prove, see for example [7].

**Lemma C.1.** *The following are invariant under unraveling: $\omega$-GML, GMSC, GNN[F]s, GNN[$\mathbb{R}$]s, and their constant iteration depth versions.*

We recall Lemma 4.2 and prove it.

**Lemma 4.2.** *Any property expressible in* MSO *and as a constant-iteration* GNN[$\mathbb{R}$] *is also* FO-*expressible.*

*Proof.* We present two different proofs. The first one is independent from Theorem 4.1 and the second one is not.

Assume that the MSO-formula $\varphi(x)$ over $\Sigma_N$ expresses the same node property as the constant iteration depth GNN[$\mathbb{R}$] $(\mathcal{G}, k)$ over $\Sigma_N$, where $k \in \mathbb{N}$ is the iteration depth of $\mathcal{G}$. It is shown in [10] that on every class of graphs of bounded treedepth, MSO and FO have the same expressive power. The class $\mathcal{C}$ of all tree-shaped $\Sigma_N$-labeled graphs of depth at most $k$ has bounded treedepth. We thus find an FO-formula $\vartheta(x)$ over $\Sigma_N$ that is logically equivalent to $\varphi(x)$ on $\mathcal{C}$, i.e., for all $T \in \mathcal{C}$ with root $w$ we have $T \models \vartheta(w)$ iff $T \models \varphi(w)$.

We may manipulate $\vartheta(x)$ into an FO-formula $\widehat{\vartheta}(x)$ such that for any pointed graph $(G, w)$, we have $G \models \widehat{\vartheta}(w)$ if and only if $U_k \models \vartheta(w)$ with $U_k$ the restriction of the unraveling of $G$ at $w$ to elements on level at most $k$. More precisely, to construct $\widehat{\vartheta}(x)$ we do the following:

- First we define an auxiliary formula

$$\psi_{\leq k}(x, y) := \bigvee_{0 \leq \ell \leq k} \exists y_0 \cdots \exists y_\ell \Big( y_0 = x \wedge y_\ell = y \wedge \bigwedge_{0 \leq m < \ell} E(y_m, y_{m+1}) \Big)$$

  which intuitively states that $y$ lies at distance at most $k$ from $x$.

- Then $\widehat{\vartheta}(x)$ is obtained from $\vartheta(x)$ by recursively replacing subformulae of type $\exists y \psi$ with $\exists y(\psi_{\leq k}(x, y) \wedge \psi)$ as follows. First, we simultaneously replace each subformula of quantifier depth 1. Having replaced subformulae of quantifier depth $\ell$, we then simultaneously replace subformulae of quantifier depth $\ell + 1$.

We then have, for every graph $G$, the following where $U$ is the unraveling of $G$ at $w$ and $U_k$ denotes the restriction of $U$ to elements on level at most $k$ (the root being on level 0):

$$G \models \varphi(w) \text{ iff } U_k \models \varphi(w) \text{ iff } U_k \models \vartheta(w) \text{ iff } G \models \widehat{\vartheta}(w)$$

The first equivalence holds because $\varphi$ is expresses the same property as $\mathcal{G}$, the second one by choice of $\vartheta$, and the third one by construction of $\widehat{\vartheta}$.

We also present an alternative proof which takes advantage of Theorem 4.1. By Theorem 3.6 $(\mathcal{G}, k)$ is equivalent to some depth-bounded $\Sigma_N$-formula $\psi$ of $\omega$-GML. On the other hand, by Theorem 4.1 $\mathcal{G}$ is equivalent to some GMSC-program $\Lambda$, since the property expressed by $\mathcal{G}$ is expressible in MSO. By the proof of Proposition 2.6 $\Lambda$ is equivalent to some width-bounded $\Sigma_N$-formula $\psi'$ of $\omega$-GML. Now, it is easy to show that $\psi \wedge \psi'$ is equivalent to some $\Sigma_N$-formula $\psi^*$ of GML, since $\psi$ is depth-bounded and $\psi'$ is width-bounded. This can be seen by transforming $\psi$ and $\psi'$ into disjunctions of (non-full) graded types by applying Proposition B.6 (see also Appendix B.4 for the definition of graded types). Since $(\mathcal{G}, k)$ is equivalent to the $\Sigma_N$-formula $\psi^*$ of GML, it expresses a node property also expressible in FO, since GML is a fragment of FO. □

## C.2 Proof of Theorem 4.1

First we recall Theorem 4.1.

**Theorem 4.1.** *Let $\mathcal{P}$ be a property expressible in* MSO. *Then $\mathcal{P}$ is expressible as a* GNN[$\mathbb{R}$] *if and only if it is expressible in* GMSC.

We recall some details of the proof sketch of Theorem 4.1. We use an automaton model proposed in [31] that captures the expressive power of MSO on tree-shaped graphs. (Note that the automaton model is defined in Section 4.) We then show that the automaton for an MSO-formula $\varphi$ that expresses the same property as a GNN[$\mathbb{R}$] (and thus as a formula of $\omega$-GML) can be translated into a GMSC-program expressing the same property. To do this, we prove the important Lemma 4.7

which shows that for all tree-shaped graphs $T$: the automaton for $\varphi$ accepts $T$ iff there is a $k$-prefix decoration of $T$ for some $k \in \mathbb{N}$. Intuitively, a $k$-prefix decoration of $T$ represents a set of accepting runs of the automaton for $\varphi$ on the prefix $T_k$ of $T$ (the formal definitions are in Section 4). Then we build a GMSC-program that accepts a tree-shaped graph $T$ with root $w$ iff there is a $k$-prefix decoration of $T$ for some $k \in \mathbb{N}$.

We next define in a formal way the semantics of parity tree automata (the definition of a parity tree automaton (PTA) is in Section 4). For what follows, a **tree** $T$ is a subset of $\mathbb{N}^*$, the set of all finite words over $\mathbb{N}$, that is closed under prefixes. We say that $y \in T$ is a **successor** of $x$ in $T$ if $y = xn$ for some $n \in \mathbb{N}$. Henceforth we will call successors **out-neighbours**. Note that the empty word $\varepsilon$ is then the root of any tree $T$. A $\Sigma$-**labeled tree** is a pair $(T, \ell)$ with $T$ a tree and $\ell : T \to \Sigma$ a node labeling function. A **maximal path** $\pi$ in a tree $T$ is a subset of $T$ such that $\varepsilon \in \pi$ and for each $x \in \pi$ that is not a leaf in $T$, $\pi$ contains one out-neighbour of $x$.

**Definition C.2** (Run)**.** Let $G$ be a $\Sigma_N$-labeled graph with $G = (V, E, \lambda)$ and $\mathcal{A} = (Q, \Sigma_N, q_0, \Delta, \Omega)$ a PTA. A **run**[3] of $\mathcal{A}$ on $G$ is a $Q \times V$-labeled tree $(T, \ell)$ such that the following conditions are satisfied:

1. $\ell(\varepsilon) = (q_0, v)$ for some $v \in V$;

2. for every $x \in T$ with $\ell(x) = (q, v)$, the following graph satisfies the formula $\Delta(q, \lambda(v))$:[4]

    - the universe consists of all $u$ with $(v, u) \in E$;
    - each unary predicate $q' \in Q$ is interpreted as the set

        $$\{\, u \mid \text{ there is an out-neighbour } y \text{ of } x \text{ in } T \text{ such that } \ell(y) = (q', u) \,\}.$$

A run $(T, \ell)$ is **accepting** if for every infinite maximal path $\pi$ of $T$, the maximal $i \in \mathbb{N}$, for which the set $\{x \in \pi \mid \ell(x) = (q, d) \text{ with } \Omega(q) = i\}$ is infinite, is even. We use $L(\mathcal{A})$ to denote the language accepted by $\mathcal{A}$, i.e., the set of $\Sigma_N$-labeled graphs $G$ such that there is an accepting run of $\mathcal{A}$ on $G$.

We remark that, in contrast to the standard semantics of FO, the graph defined in Point 2 of the above definition may be empty and thus transition formulas may also be interpreted in the *empty graph*. A transition formula is true in this graph if and only if it does not contain any existential quantifiers, that is, $k = 0$. Note that a transition formula $\vartheta$ without existential quantifiers is a formula of the form $\forall z(\, \mathsf{diff}(z) \to \psi)$, where $\psi$ is a disjunction of conjunctions of atoms $q(z)$ which are unary predicates for the states of the automaton. Such a formula may or may not be true in a non-empty graph. For example, if $\psi$ in $\vartheta$ is a logical falsity (the empty disjunction), then $\vartheta$ is satisfied only in the empty graph.

Let $\mathcal{P}$ be a node property over $\Sigma_N$ which is expressible in MSO and also in $\omega$-GML. Let $\mathcal{A} = (Q, \Sigma_N, q_0, \Delta, \Omega)$ be a PTA that is obtained by Theorem 4.6 from $\mathcal{P}$. If $\psi$ is the $\omega$-GML-formula expressing $\mathcal{P}$, then we may simply say that $\mathcal{A}$ and $\psi$ are equivalent. We identify a sequence $S_1, \ldots, S_n$ of subsets of $Q$ with a graph $\mathsf{struct}(S_1, \ldots, S_n)$ defined as follows:

    - the universe is $\{1, \ldots, n\}$;
    - each unary predicate $q' \in Q$ is interpreted as the set $\{i \mid q' \in S_i\}$.

For a tree-shaped graph $T$, we let $T_k$ denote the restriction of $T$ to the nodes whose distance from the root is at most $k$. An **extension** of $T_k$ is then any tree-shaped graph $T'$ such that $T'_k = T_k$, that is, $T'$ is obtained from $T_k$ by extending the tree from the nodes at distance $k$ from the root by attaching subtrees, but not from any node at distance $\ell < k$ from the root. Note that $k$-*prefix decorations of* $T$ and *universal sets of states* are defined in Section 4.

**Lemma 4.7.** For every tree-shaped $\Sigma_N$-labeled graph $T$: $T \in L(\mathcal{A})$ if and only if there is a $k$-prefix decoration of $T$, for some $k \in \mathbb{N}$.

---

[3]Often semantics for parity tree automata are given with parity games, for the details see for example [14]. Informally, parity games are played by two players called Eloise and Abelard, where Eloise tries to show that the PTA accepts a given graph. Informally, the semantics introduced here represents a winning strategy of Eloise in parity games and similar semantics are used for example in [23].

[4]Note that the graph is empty if and only if $v$ is a dead end in $G$.

*Proof.* "⇒". Assume that $T \in L(\mathcal{A})$. Let $\bigvee_i \psi_i$ be the $\Sigma_N$-formula of $\omega$-GML that $\mathcal{A}$ is equivalent to. Then $T \models \psi_i$ for some $i$. Let $k$ be the modal depth of $\psi_i$. Then $T' \models \psi_i$ for every extension $T'$ of $T_k$.

Next we construct a mapping $\mu\colon V_k \to \mathcal{P}(\mathcal{P}(Q))$ and show that $\mu$ is a $k$-prefix decoration of $(T, w)$. We construct $\mu$ as follows. For all $v \in V$ on the level $k$, we define $\mu(v)$ as the universal set for $\lambda(v)$. Then analogously to the third condition in the definition of $k$-prefix decorations, we define $\mu(v)$ for each node $v$ on a level smaller than $k$. Now, all we have to do is show that the first condition in the definition of $k$-prefix decorations is satisfied, i.e., for each $S \in \mu(w)$, $q_0 \in S$. Then we may conclude that $\mu$ is a $k$-prefix decoration.

Assume by contradiction that there exists a set $S \in \mu(w)$ such that $q_0 \notin S$. (Note that $\mu(v) \neq \emptyset$ for all $v \in V_k$ by the definition of $k$-prefix decorations.) By Condition 3 of $k$-prefix decorations, there must exist sets $S_1^1 \in \mu(u_1^1), \ldots, S_{n_1}^1 \in \mu(u_{n_1}^1)$ (where $u_1^1, \ldots, u_{n_1}^1$ are the nodes on level 1) such that $q \in S$ iff the transition formula $\Delta(q, (\lambda(w)))$ is satisfied in the graph where each $u_i^1$ is labeled $S_i^1$ for each $1 \leq i \leq n_1$. By the same logic, we find such sets $S_1^m \in \mu(u_1^m), \ldots, S_{n_m}^m \in \mu(u_{n_m}^m)$ (where $u_1^m, \ldots, u_{n_m}^m$ are the nodes on level $m$) for each $m \leq k$ (note that these nodes do not necessarily share the same predecessor). Let $T'$ be an extension of $T_k$ that is obtained by attaching to each node $u_i^k$ of $T_k$ on level $k$ some rooted tree $T''$ that is accepted by $\mathcal{A}$ precisely when starting from one of the states in the set we chose for that node, i.e., $Q_{T''} = S_i^k$.

Now, we can demonstrate that there is no accepting run $r_{T'} = (T_{T'}, \ell)$ of $\mathcal{A}$ on $T'$. Any such run has to begin with $\ell(\varepsilon) = (q_0, w)$. Note that the out-neighbours of $\varepsilon$ cannot be labeled with exactly the labels $(q, u_i^1)$ such that $q \in S_i^1$ because $\Delta(q_0, \lambda(w))$ is not satisfied in the graph consisting of the out-neighbours $u_i^1$ of $w$ labeled with $S_i^1$ for each $1 \leq i \leq n_1$. In fact, the out-neighbours of $\varepsilon$ cannot be labeled with any subset of such labels either, because by definition all transition formulae $\vartheta$ are monotonic in the sense that

$$(V, E, \lambda) \not\models \vartheta \implies (V, E, \lambda') \not\models \vartheta \text{ for all } \lambda' \subseteq \lambda.$$

Thus, there is an out-neighbour $x_1$ of $\varepsilon$ in $T_{T'}$ such that $\ell(x_1) = (q, u_i^1)$ where $q \notin S_i^1$, and we may continue this examination starting from $x_1$ in the same way. Inductively, we see that for any level $m$ we find a son $x_m$ of $x_{m-1}$ such that $\ell(x_m) = (q, u_i^m)$ where $q \notin S_i^m$, including the level $m = k$ where we let $\ell(x_k) = (q, u_i^k)$. Thus $q \notin S_i^k$. However, we have $S_i^k = Q_{T''}$ and it is witnessed by the run $r_{T'}$ that $\mathcal{A}$ accepts the tree $T''$ rooted at $u_i^k$ when started in state $q$. Thus, $q \in S_i^k$, a contradiction.

"⇐". Let $\mu\colon V_k \to \mathcal{P}(\mathcal{P}(Q))$ be a $k$-prefix decoration of $T$, for some $k$. We may construct from $\mu$ an accepting run $r = (T', \ell)$ of $\mathcal{A}$ on $T$. For every node $v$ in $T_k$ on level $k$, let $T_v$ denote the subtree of $T$ rooted at $v$. Let a **semi-run** be defined like a run except that it needs not satisfy the first condition from the definition of runs (that is, it need not start in the initial state of the PTA). A semi-run being **accepting** is defined exactly as for runs.

Take any node $v$ in $T_k$ on level $k$. Since $\mu(v)$ is universal for $\lambda(v)$, we find an $S_v \in \mu(v)$ such that $q \in S_v$ if and only if $T_v \in L(\mathcal{A}_q)$. (Note that $\mu(v) \neq \emptyset$ for all $v \in V_k$ by the definition of $k$-prefix decorations.) Consequently, for each $q \in S_v$ we find an accepting semi-run $r_{v,q} = (T'_{v,q}, \ell_{v,q})$ of $\mathcal{A}$ on $T_v$ with $\ell_{v,q}(\varepsilon) = (q, v)$. We now proceed upwards across $T_k$, assembling all these semi-runs into a run. In particular, we choose a set $S_u \in \mu(u)$ for every node $u$ in $T_k$ and, for all $q \in S_u$, a semi-run $r_{u,q} = (T'_{u,q}, \ell_{u,q})$ with $\ell_{u,q}(\varepsilon) = (q, u)$.

Let $v$ be a node in $T_k$ that has not yet been treated and such that its out-neighbours $u_1, \ldots, u_n$ have already been treated, that is, we have already selected a set $S_{u_i} \in \mu(u_i)$ for $1 \leq i \leq n$ along with the associated semi-runs. Due to Condition 3 of $k$-prefix decorations, we find a set $S_v \in \mu(v)$ such that for each $q \in S_v$: $\Delta(q, \lambda(v))$ is satisfied by the graph struct$(S_{u_1}, \ldots, S_{u_n})$. Choose this set $S_v$.[5] As for the semi-runs, let $q \in S_v$. Then $\Delta(q, \lambda(v))$ is satisfied by the above graph. We may thus choose as $r_{v,q}$ the semi-run that is obtained as follows:

1. start with a fresh root $\varepsilon$ and set $\ell_{v,q}(\varepsilon) = (q, v)$;

2. for each $i \in \{1, \ldots, n\}$ and each $q' \in S_{u_i}$, add the semi-run $r_{u_i, q'}$ as a subtree, making the root of $r_{u_i, q'}$ an out-neighbour of the fresh root that we had chosen.

---

[5]Note that if $v$ has no out-neighbours, then $S_v$ is the set of states $q$ such that $\Delta(q, \lambda(v))$ is satisfied in the empty graph.

Note in step 2 that for each $1 \le i \le n$, if $S_{u_i} = \emptyset$, then no semi-run $r_{u_i,q'}$ is added as a subtree. Let $w$ be the root of $T$. In view of Condition 1 of $k$-prefix decorations, it is easy to verify that the semi-run $r_{w,q_0}$ is in fact a run of $\mathcal{A}$ on $T$. Moreover, this run is accepting since we had started with accepting semi-runs at the nodes $v$ in $T_k$ on level $k$ and the finite initial piece that $r_{w,q_0}$ adds on top of those semi-runs has no impact on which states occur infinitely often in infinite paths. $\qquad \square$

By Lemma 4.7, we may finish the proof of Theorem 4.1 by constructing a GMSC-program $\Lambda$ such that for every tree-shaped $\Sigma_N$-labeled graph $T$ with root $w$, we have $T, w \models \Lambda$ iff there is a $k$-prefix decoration of $T$, for some $k$. The definition of $\Lambda$ follows the definition of $k$-prefix decorations. This construction serves as the proof of Lemma C.4 below it.

We define a fresh schema variable $X_S$ for all $S \subseteq Q$. First, a set $P \subseteq \Sigma_N$ of node label symbols can be specified with the formula

$$\varphi_P := \bigwedge_{p \in P} p \wedge \bigwedge_{p \in \Sigma_N \setminus P} \neg p,$$

which states that the node label symbols in $P$ are true and all others are false. Let $\mathcal{Q}_P$ denote the universal set for $P$. For every $X_S$, the program $\Lambda$ contains the following terminal clause, reflecting Condition 2 of $k$-prefix decorations:

$$X_S(0) :- \bigvee_{S \in \mathcal{Q}_P} \varphi_P.$$

Note that if the disjunction is empty, we have $X_S(0) :- \bot$.

We also define a special appointed predicate $A$ that is true when all the head predicates $X_S$ that do not contain $q_0$ are false, reflecting Condition 1 of $k$-prefix decorations. More formally $A$ is the only appointed predicate of the program, and it is defined as follows: $A(0) :- \bot$ and

$$A :- \bigwedge_{q_0 \notin S} \neg X_S.$$

For the iteration clauses of head predicates $X_S$, we need some preliminaries. Let $K$ be the maximum over all $k$ such that $\Delta$ mentions a transition formula

$$\vartheta := \exists x_1 \cdots \exists x_k \left( \mathsf{diff}(x_1, \ldots, x_k) \wedge q_1(x_1) \wedge \cdots \wedge q_k(x_k) \wedge \forall z \left( \mathsf{diff}(z, x_1, \ldots, x_k) \to \psi \right) \right).$$

A *counting configuration* $c \in \mathcal{M}_{K+1}(\mathcal{P}(Q))$ is a multiset of sets of states that contains each set of states at most $K + 1$ times. A sequence $S_1, \ldots, S_n$ of subsets of $Q$ **realizes** the counting configuration $c$ if for each $S \subseteq Q$, one of the following holds:

- $c(S) \le K$ and the number of sets $S_i$ among $S_1, \ldots, S_n$ with $S_i = S$ is $c(S)$;
- $c(S) = K + 1$ and the number of sets $S_i$ among $S_1, \ldots, S_n$ with $S_i = S$ exceeds $K$.

It is easy to prove the following lemma.

**Lemma C.3.** *Let $\vartheta$ be a transition formula mentioned in $\Delta$ and assume that $S_1, \ldots, S_n$ and $S'_1, \ldots, S'_{n'}$ realize the same counting configuration. Then $\mathsf{struct}(S_1, \ldots, S_n)$ satisfies $\vartheta$ iff $\mathsf{struct}(S'_1, \ldots, S'_{n'})$ satisfies $\vartheta$.*

By the above lemma, we may write $c \models \vartheta$, meaning that $\mathsf{struct}(S_1, \ldots, S_n)$ satisfies $\vartheta$ for any (equivalently: all) $S_1, \ldots, S_n$ that realize $c$.

It is easy to see that we can specify a counting configuration $c \in \mathcal{M}_{K+1}(\mathcal{P}(Q))$ in GMSC because we can count out-neighbours; for example, we can write the following formula, which states that for each set $S$ of states there are exactly $c(S)$ out-neighbours where the label includes $S$, unless $c(S) = K + 1$, in which case it permits more such out-neighbours:

$$\psi_c := \bigwedge_{c(S) = \ell \le K} \Diamond_{=k} X_S \wedge \bigwedge_{c(S) = K+1} \Diamond_{\ge K+1} X_S.$$

For each set $P$ of node label symbols and set $S$ of states, we can specify the set of counting configurations in $\mathcal{M}_{K+1}(\mathcal{P}(Q))$ that satisfy a formula $\Delta(q, P)$ if and only if $q \in S$ with the following disjunction:

$$\Psi_{S,P} := \bigvee_{\substack{c \in \mathcal{M}_{K+1}(\mathcal{P}(Q)) \\ c \models \Delta(q,P) \text{ iff } q \in S}} \psi_c.$$

Recall also that the formula $\varphi_P$ specifies the set of node label symbols in $P$. The iteration clause for $X_S$ must state that there is a set $P$ of node label symbols that are true and for which $\Psi_{S,P}$ is true. This can be expressed as below:

$$X_S :- \bigvee_{P \subseteq \Sigma_N} \left( \varphi_P \wedge \Psi_{S,P} \right).$$

We prove that the GMSC-program $\Lambda$ characterizes $k$-prefix decorations associated with $\mathcal{A}$. A **pseudo $k$-prefix decoration** $\mu$ is defined like a $k$-prefix decoration except that it needs not satisfy the first condition of the definition of $k$-prefix decorations (that is, every set in the root given by $\mu$ does not need to contain the initial state). Now, given $\ell \in \mathbb{N}$ and a tree-shaped $\Sigma_N$-labeled graph $T$, let $\mu_T^\ell$ denote the pseudo $\ell$-prefix decoration of $T$. For each $\ell \in \mathbb{N}$ and each set $S$ of states of $\Lambda$, we show by induction on $n \in [0; \ell]$ that for every tree-shaped $\Sigma_N$-labeled graph $T$ and for each node $v$ in $T$ on level $\ell - n$, we have

$$T, v \models X_S^n \iff S \in \mu_T^\ell(v).$$

In the case $n = 0$, the claim holds trivially for the nodes $v$ on level $\ell$ by definition of $\Lambda$, since then $T, v \models X_S^0$ if and only if $S$ is in the universal set for $P$, where $P$ is the set of node label symbols that appear in $v$.

Assume that the claim holds for $n < \ell$; we will show that it also holds for $n + 1$. Let $v$ be a node of $T$ on the level $\ell - (n + 1)$.

First assume that $T, v \models X_S^{n+1}$, that is, for some $P \subseteq \Sigma_N$, we have $T, v \models \varphi_P^n$ and $T, v \models \Psi_{S,P}^n$. Therefore, there is a multiset $c \in \mathcal{M}_{K+1}(\mathcal{P}(Q))$ such that $c \models \Delta(q, P)$ iff $q \in S$ and $T, v \models \psi_c^n$. Thus, for every set $S'$ of states of $\mathcal{A}$ we have that if $c(S') = m \leq K$, there are exactly $m$ out-neighbours $v_1, \ldots, v_m$ of $v$ such that $T, v_i \models X_{S'}^n$ for every $i \in [m]$, and if $c(S') = K + 1$, there are at least $K + 1$ out-neighbours $v_1, \ldots, v_{K+1}$ of $v$ such that $T, v_i \models X_{S'}^n$ for every $i \in [K + 1]$. By the induction hypothesis, for every set $S'$ of states of $\mathcal{A}$ we have that if $c(S') = m \leq K$, there are exactly $m$ out-neighbours $v_1, \ldots, v_m$ of $v$ such that $S' \in \mu_T^\ell(v_i)$ for every $i \in [m]$, and if $c(S') = K + 1$, there are at least $K + 1$ out-neighbours $v_1, \ldots, v_{K+1}$ of $v$ such that $S' \in \mu_T^\ell(v_i)$ for every $i \in [K+1]$. Since $c \models \Delta(q, P)$ iff $q \in S$ and by the definition of pseudo $k$-prefix decorations, we have $S \in \mu_T^\ell(v)$.

Then assume that $S \in \mu_T^\ell(v)$. Let $\{v_1, \ldots, v_m\}$ be the set of out-neighbours of $v$. By the induction hypothesis for each $i \in [m]$ and each set $S'$ of states of $\mathcal{A}$, it holds that $S' \in \mu_T^\ell(v_i)$ iff $T, v_i \models X_{S'}^n$. By the definition of pseudo $k$-prefix decorations there are sets $S_1 \in \mu_T^\ell(v_1), \ldots, S_m \in \mu_T^\ell(v_m)$ such that $q \in S$ iff $\text{struct}(S_1, \ldots, S_m) \models \Delta(q, P)$, where $P$ is the set of node label symbols that are true in $v$. Now, let $c \in \mathcal{M}_{K+1}(\mathcal{P}(Q))$ be a counting configuration realized by $S_1, \ldots, S_m$. Now since $K$ is the maximum over all $k$ such that $\Delta$ mentions a transition formula

$$\vartheta := \exists x_1 \cdots \exists x_k \left( \text{diff}(x_1, \ldots, x_k) \wedge q_1(x_1) \wedge \cdots \wedge q_k(x_k) \wedge \forall z( \text{diff}(z, x_1, \ldots, x_k) \to \psi) \right),$$

we have $c \models \Delta(q, P)$. Therefore, $T, v \models \psi_c^n$ and trivially $T, v \models \varphi_P^n$, and thus $T, v \models X_S^{n+1}$.

Now, if $\Lambda$ accepts a $\Sigma_N$-labeled tree-shaped graph $(T, w)$, then by the result above it means that there is a pseudo $k$-prefix decoration $\mu$ such that $T, w \models A^{k+1}$. Thus in round $k$ we have $T, w \not\models X_S^k$ for all $S$ where $q_0 \notin S$. Therefore $S \notin \mu(w)$ for all $S$ where $q_0 \notin S$, i.e., $\mu$ is actually a $k$-prefix decoration. Similarly if a tree-shaped graph $(T, w)$ has a $k$-prefix decoration, then $T, w \models A^{k+1}$ by the result above. Thus we have proved the lemma below.

**Lemma C.4.** *For every tree-shaped $\Sigma_N$-labeled graph $T$ with root $w$: $T, w \models \Lambda$ iff there is a $k$-prefix decoration of $T$, for some $k$.*

Now we are ready to prove Theorem 4.1.

*Proof of Theorem 4.1.* Note that some of the results needed for this proof are given in this appendix section and some are given outside the appendix. Let $\mathcal{P}$ be a node property expressible in MSO by an MSO-formula $\varphi(x)$ over $\Sigma_N$.

Assume that $\mathcal{P}$ is expressible by a $\Sigma_N$-program of GMSC. By Proposition 2.6 $\mathcal{P}$ is also expressible $\Sigma_N$-formula of $\omega$-GML and thus by Theorem 3.4 as a GNN[$\mathbb{R}$] over $\Sigma_N$. For the converse, assume that $\mathcal{P}$ is expressible as a GNN[$\mathbb{R}$] over $\Sigma_N$. Thus $\mathcal{P}$ is expressible by $\Sigma_N$-formula of $\omega$-GML by Theorem 3.4. Therefore, there is a PTA $\mathcal{A}$ and a $\Sigma_N$-program $\Lambda$ of GMSC such that for any $\Sigma_N$-labeled graph $G$ with root $w$:

$$G \models \varphi(w) \quad \overset{\text{Theorem 4.6}}{\Longleftrightarrow} \quad \text{the unraveling } U \text{ of } G \text{ at } w \text{ is in } L(\mathcal{A})$$

$$\overset{\text{Lemma 4.7}}{\Longleftrightarrow} \quad \text{there is a } k\text{-prefix decoration of } U, \text{ for some } k \in \mathbb{N}$$

$$\overset{\text{Lemma C.4}}{\Longleftrightarrow} \quad U, w \models \Lambda$$

$$\overset{\text{Lemma C.1}}{\Longleftrightarrow} \quad G, w \models \Lambda.$$

Thus, we have proven Theorem 4.1. □

## C.3 Proof of Theorem 4.3

We recall and prove Theorem 4.3.

**Theorem 4.3.** *Let $\mathcal{P}$ be a node property expressible in* MSO*. Then $\mathcal{P}$ is expressible as a* GNN[$\mathbb{R}$] *if and only if it is expressible as a* GNN[F]*. The same is true for constant-iteration* GNN*s.*

*Proof.* Let $\mathcal{P}$ be an MSO-expressible property over $\Sigma_N$. By Theorem 4.1, $\mathcal{P}$ is expressible as a GNN[$\mathbb{R}$] iff it is expressible in GMSC. Thus by Theorem 3.2 $\mathcal{P}$ is expressible as a GNN[$\mathbb{R}$] iff it is expressible as a GNN[F].

For constant-iteration GNNs we work as follows. First assume that $\mathcal{P}$ is expressible as a constant-iteration GNN[$\mathbb{R}$] over $\Sigma_N$. Then by Lemma 4.2, $\mathcal{P}$ is also expressible in FO. Thus, by Theorem 4.2 in [5] $\mathcal{P}$ is expressible by a $\Sigma_N$-formula of GML (trivially, each constant-iteration GNN $(\mathcal{G}, L)$ is trivial to translate to an $L$-layer GNN; see Appendix B.7 for the definition of $L$-layer GNNs). For a GML-formula it is easy to construct an equivalent constant-iteration GNN[F]; the technique is essentially the same as in the proof of Lemma B.5 (omit the clock in the construction) and in the proof of Theorem 4.2 in [5]. The converse direction is trivial. □

# D Appendix: Conclusion

## D.1 A note on other termination conditions

A variant of our proofs gives a counterpart of Theorem 3.2 for recurrent GNNs with a termination condition based on fixed points, studied under the name of "RecGNN" in [26] and closely related to the termination condition in [30, 11].

We start by defining a fixed point acceptance condition for GNNs and for GMSC. Let $\mathcal{G}$ be a GNN over $\Pi$, let $(G, w)$ be a pointed $\Pi$-labeled graph, and let $x_w^t$ denote the feature vector of $w$ in round $t$. We say that $\mathcal{G}$ **fixed point accepts** $(G, w)$ if there is a round $t \in \mathbb{N}$ such that $x_w^T \in F$ for all $T \geq t$.

We define the same accepting condition for GMSC analogously. Let $\Lambda$ be a $\Pi$-program of GMSC, let $\mathcal{A}$ be the set of appointed predicates of $\Lambda$ and let $(G, w)$ be a pointed $\Pi$-labeled graph. We say that $\Lambda$ **fixed point accepts** $(G, w)$ if there is a round $t \in \mathbb{N}$ such that for all $T \geq t$ we have $G, w \models X^T$ for some $X \in \mathcal{A}$.

Next we discuss how the proof of Theorem 3.2 is modified such that the same result holds with fixed point acceptance.

**Theorem D.1.** *The following three have the same expressive power relative to fixed point acceptance:* GNN[F]*s,* GMSC*, and R-simple aggregate-combine* GNN[F]*s.*

*Proof.* We start by showing that each GNN[F] can be translated into an equivalent GMSC-program relative to fixed point acceptance. First we note that it is easy to translate a GNN[F] $\mathcal{G}$ over $(\Pi, d)$

into an equivalent bounded FCMPA $\mathcal{A}$ over $\Pi$ such that the following holds. Let $x_w^t$ denote the feature vector of $\mathcal{G}$ in round $t$ at node $w$ and respectively let $y_w^t$ denote the state of $\mathcal{A}$ in round $t$ at node $w$. Let $F_{\mathcal{G}}$ and $F_{\mathcal{A}}$ denote the accepting states of $\mathcal{G}$ and $\mathcal{A}$ respectively. Then for all pointed $\Pi$-labeled graphs $(G, w)$ and for all $t \in \mathbb{N}$ it holds that

$$x_w^t \in F_{\mathcal{G}} \iff y_w^t \in F_{\mathcal{A}}.$$

Moreover, the construction in the proof of Lemma B.3 shows that we can construct a GMSC-program $\Lambda$ over $\Pi$ for $\mathcal{A}$ with the set $A$ of appointed predicates such that for all pointed $\Pi$-labeled graphs $(G, w)$ and for all $t \in \mathbb{N}$ we have

$$G, w \models X^t \text{ for some } X \in A \iff y_w^t \in F_{\mathcal{A}}.$$

Clearly, for all pointed $\Pi$-labeled graphs $(G, w)$: $\mathcal{G}$ fixed point accepts $(G, w)$ iff $\Lambda$ fixed point accepts $(G, w)$.

Then we show that each GMSC-program $\Lambda$ can be translated into an equivalent R-simple aggregate-combine GNN[F] $\mathcal{G}$ relative to fixed point acceptance. We simply modify the proof of Lemma B.5 as follows: we add to the set of accepting feature vectors all feature vectors $\mathbf{v}$ where $\mathbf{v}_{N+D+1} \neq 1$. Now it is clear for all pointed graphs $(G, w)$ that $\Lambda$ fixed point accepts $(G, w)$ if and only if $\mathcal{G}$ fixed points accepts $(G, w)$. $\qquad \square$

