# OpenReview forum: "Logical characterizations of recurrent graph neural networks with reals and floats"
_NeurIPS.cc/2024/Conference — NeurIPS 2024 poster_

### Official Review · Reviewer_2zSg · 2024-06-12

**Soundness:** 4
**Presentation:** 3
**Contribution:** 3
**Rating:** 6
**Confidence:** 4

**Summary:**

This paper examines the relationship between GNN models based on real numbers, which are predominantly studied in theory, and GNN models based on floating point numbers, which are commonly used in practice. Additionally, the paper advances the state of the art in the study of recurrent GNNs, where the computation is not restricted to a fixed number of layers. Termination in the studied recurrent GNNs is indicated by designated "accepting" feature vectors.

The GNNs studied in this paper feature a termination condition indicated by designated accepting feature vectors. The authors establish connections between the expressive power of these GNNs and logical formalisms such as the graded modal substitution calculus (GMSC) and Monadic Second Order Logic (MSO). Specifically, the authors prove that recurrent GNNs over floats and GMSC are equally expressive and, also, that for properties of Kripke structures definable in MSO, recurrent GNNs over floats and reals have the same expressive power.

**Strengths:**

- The paper advances the state of the art in the theoretical study of the expressive power of graph representation learning formalisms. The authors rightly point out that most theoretical work has been focused on standard GNNs with a fixed number of layers and that only recently in [23] this has been extended to GNNs with some form of fixpoint computation. The separate consideration of floating point numbers in a theory paper is also novel and interesting, as most theoretical studies assume real-valued feature vectors throughout the computation of the GNNs.

- The paper is rigorously written to a high standard of quality and precision.

- The obtained results are non-trivial and can motivate further theoretical study on the connection between graph representation learning models and logic.

**Weaknesses:**

- The paper is accessible only to seasoned logicians, a very small proportion of the NeurIPS community. I would have expected the authors to make an effort to present their results in a manner more accessible to the broader representation learning community. Instead, this feels more like a submission to LICS, and the authors appear not to have considered the suitability of their chosen venue for their work.

- There is no empirical evidence demonstrating that the proposed recurrent GNNs can be successfully trained to accomplish a useful task in practice. Specifically, it remains unclear whether the proposed recurrent model can address (some of) the limitations of standard GNNs in practical applications. The only mention of applications is in the first three lines of the introduction, which are generic statements about standard GNNs. I would have expected the authors to at least attempt to identify some practical needs unmet by standard GNN models, which their proposed recurrent models could potentially fulfil.

- Relevant related work has been cited. However, I would have expected a more in-depth discussion on the contributions of this paper with respect to [23] and also some discussion on the connection between GMSC and recent logics proposed to capture standard GNNs with a fixed number of layers beyond the framework of First-Order logic, such as the use of counting terms or Presburger quantifiers.

- The fixpoint termination condition proposed by the authors, based on a set of designated final feature vectors feels indeed natural from an automata-theoretic perspective; it is, however, unclear to me how these vectors could be learnt, as suggested by the authors. Concerning termination, it would appear to me that the recurrent application of these GNNs may not terminate at all (e.g., if the acceptance vectors are never reached during recurrent computation). I would have expected the authors to comment on the implications of this possibility.

- It is unclear whether the expressivity results also provide an effective procedure for computing, given a trained recurrent GNN, the corresponding GMSC theory. What would be the size of such a theory based on the size of the GNN representation?

**Questions:**

- Please comment on potential applications of this work
- Please provide a more detailed account on the differences of this work wrt [23] as well as on the connection between GMSC and logics with counting terms and/or presburger quantifiers.
- Please comment on the possibility of learning the acceptance vectors and the possibility of non-termination.

**Limitations:**

The authors have indicated some limitations of their work in the paper.

---

> ### Author Rebuttal · Authors · 2024-08-06
>
> We first address the weaknesses.
>
> We will improve the accessibility of the paper by lightening the preliminaries by moving non-essential parts to the appendix and adding examples (including a GMSC-program for reachability, to complement the corresponding GNN example) and illustrating remarks.
>
> About the remark on empirical evidence, we view our GNN model as a basic yet expressive version of recurrent GNNs. Please note that the upper bound of expressivity of our model with reals is ω-GML (Theorem 3.4), and thus one can conceive this upper bound as a sequence of GML-formulas of increasing modal depth. Barceló et al. [5] showed that GML-formulas correspond to constant-time GNNs, so our model relates to sequences of constant-time GNNs with the constant gradually increased. Also, our GNN[F]s can express all properties in the mu-fragment of the graded mu-calculus (cf. the response to all reviewers). We expect that typical properties relevant in practice fall into that class. However, we indeed did not perform practical experiments, concentrating instead on theory. We will add a remark on this to the limitations section.
>
> The weaknesses regarding relevant work, learning acceptance vectors and non-termination are addressed later below.
>
> Concerning the size of a GMSC theory corresponding to a recurrent GNN and its computability: the translation GNN[F]s -> GMSC is effective with only polynomial blowup. It is indeed good to point this out.
>
> We then address the questions.
>
> Regarding potential applications, studying the expressivity of GNN models, and characterizing GNNs via logic, is an active research theme, and our main aim is to contribute to that. While results of this kind have no immediate applications to practical learning, they are useful for understanding (also the practical) limitations of GNNs.
>
> Concerning whether our recurrent GNN model, with its specific termination condition, has practical applications: note that there is no standard termination condition for recurrent GNNs, and in publications on practical applications of recurrent GNNs, the termination condition is often not made explicit. We believe our condition is natural. Please note also that our GNNs contain the mu-fragment of graded mu-calculus; we expect that typical properties relevant in practice fall into that class. Also, as noted in the conclusion and appendix, our characterization in Theorem 3.2 also applies to the fixpoint acceptance condition of [23] if GMSC is modified accordingly. We will make this more explicit.
>
> We will improve the discussion on [23] and [6]. Some central points:
>
> [23] differs from our setup in three crucial ways: (i) [23] studies GNNs with reals (not floats), and (ii) with unrestricted (possibly even non-computable) aggregation and combination functions, and (iii) with different termination conditions.
>
> The setup of [23] can express all (even undecidable) bisimulation-invariant properties and yields potentially extremely powerful GNNs. Our main results concern the case of floats and R-simple GNNs not studied in [23]. The ensuing logical characterizations are quite different:
>
> - the background logic of [23] is LocMMFP (a version of monadic least fixpoint logic) while we use the more expressive MSO.
>
> - our characterization is via GMSC while theirs is via the (two-way) graded mu-calculus; as pointed out in our paper, these logics are orthogonal in expressivity. This is due to the differences (i) to (iii).
>
> [6] studies constant iteration-depth GNNs, not recurrent ones. In the case of eventually constant activation functions, equivalence to a certain Presburger modal logic is shown; this does not extend to activation functions that are not eventually constant. Our results enable any activation function, except for R-simple GNNs which use truncated ReLU. The logic ω-GML contains Presburger modal logic as a proper fragment.
>
> We then comment on the possibility of learning acceptance vectors. With "termination can be learned in the training phase" in the intro, we only meant the following: specify a fixed set F of accepting feature vectors before training and try to train the model so that termination is achieved on all training examples. If this doesn't work, declare failure or modify F by making it smaller or larger, and try again.
>
> Regarding the possibility of non-termination, for many acceptance conditions, such as ours or conditions based on fixed point constructs, training and applying a recurrent GNN brings questions of termination. For GNNs over reals with our accepting condition, any node that accepts is classified "accepting" in a finite number of rounds. Non-accepting nodes might not be classified "non-accepting" in any finite number of rounds (in the general setting with no additional assumptions). The same issue is true of, e.g., the node-local fixed-points of the GNNs in [23], where the issue concerns both accepting and non-accepting nodes. Now, with float GNNs---at least in a certain theoretical sense---everything conceivably "terminates" due to GNNs always reaching a global attractor (meaning global configurations ultimately start repeating). But this is indeed a theoretical remark. Also, the mu-fragment contained in GMSC (please see above) is based on local fixpoints with better convergence properties, so many scenarios are more favourable with suitable assumptions.
>
> From a less theory-oriented perspective, if gradient descent is used for training, one needs to repeatedly evaluate the model with its current parameterization on training examples, possibly facing non-termination. In practice, this could be dealt with by declaring non-termination after a certain number of steps, e.g., via a function of the size of the training example. We will note in the limitations section that the we do not address training of GNNs, which is a fair point. When applying a trained GNN, one may also face non-termination. Again, in practice this could be dealt with by stopping after a certain number of steps.

---

> > ### Comment · Reviewer_2zSg · 2024-08-09
> > **Thanks for the clarifications**
> >
> > Thanks for the clarifications.
> >
> > I would appreciate it if you could make the suggested changes to the paper as they would certainly improve readability and accessibility. I feel that termination issues are important, and so are practical considerations related to training.

---

### Official Review · Reviewer_TUPQ · 2024-06-18

**Soundness:** 4
**Presentation:** 3
**Contribution:** 4
**Rating:** 8
**Confidence:** 5

**Summary:**

The paper presents logical characterizations of recurrent GNNs in terms of logical formalisms, following the line of previous work by Barceló et al. It is shown that recurrent GNNs have the same expressive power as the infinitary extension of graded modal logic, when arbitrary precision is allowed for feature vectors, and as the graded modal substitution calculus (GMSC), when only fixed precision is admitted. In addition, arbitrary precision boils down to finite precision if the property that is being considered can be expressed in monadic second order logic (MSO).

**Strengths:**

This is one of the most important papers to have appeared in the area of the expressiveness of GNNs over the last years. While several of these results can be considered kind of folklore in hindsight, I really appreciate the fact that the authors have spelled out all these connections in full detail and clarified several missing points. In particular, the study of finite precision GNNs is very interesting and the connection to GMSC of great importance.

I have no doubts about the pertinence and technical quality of the article, and I think it should be accepted to NeurIPS.

**Weaknesses:**

I see no weakness on the paper. This is a comprehensive and solid piece of work that should clearly be accepted at the conference.

**Questions:**

This is something that intrigues me: Is it possible to establish a connection between the GMSC and some version of (partial) fixed-point logics?

Btw, in his survey on GNNs, Grohe mentions the following question:

Question 2. Let Q be a unary query expressible in a suitable modal (2-variable) fixed-point logic with counting. Is there a
recurrent GNN (with global readout) expressing Q?

Do you have an answer to this question?

**Limitations:**

The authors have correctly addressed the limitations of their work.

---

> ### Author Rebuttal · Authors · 2024-08-06
>
> Concerning question 1, GMSC translates into partial fixed-point logic with choice (see, e.g., David Richerby, Logical Characterizations of PSpace (CSL 2004)): it is easy to see that GMSC has PSpace data complexity upper bound (one simply keeps in memory a subset of the graph domain for each schema variable), and partial fixed-point logic with choice captures PSpace on all (including unordered) models. It is also interesting to note that GMSC of course does not translate to any monadic fixed-point logic contained in MSO (simply because GMSC and MSO are orthogonal in expressive power).
>
> Concerning question 2, we probably do not have a definite answer, but several relevant observations:
>
> * The mu-fragment of graded modal mu-calculus translates into GMSC (cf. the response to all reviewers). Thus our results imply that every property expressible by the mu-fragment of graded mu-calculus is expressible by a recurrent floating-point GNN (without global readout).
>
> * On the other hand, non-reachability is expressible in the mu-calculus, but it is not even in omega-GML, which is easy to show as follows. If non-reachability of p was expressible by omega-GML, then a node w in a small cycle, from where we cannot reach any node satisfying p, would satisfy one of the disjuncts φ of some modal depth n. Consequently, a node w' in a larger cycle with an isomorphic local environment up to modal depth n would satisfy φ also, even if p was reachable from w' at a distance greater than n. This is a contradiction.
>
> * The situation with the expressive power of GMSC changes with global readouts added. In GMSC with global readouts, non-reachability becomes expressible, because we can "detect" fixed points. Let Chi be a program such that the head predicate X will reach a fixed point containing precisely those nodes from where a node satisfying p is reachable; this program is definable already in the mu-fragment. Now, additionally, we can use auxiliary head predicates to keep track of interpretations of X at any two successive stages of computation. To detect when the fixed point of X is reached, we essentially simply use a global readout and the auxiliary head predicates to state that X has remained unchanged globally for two successive rounds. Once we know that the fixed-point has been reached, we accept all nodes in the complement of the fixed point of X.

---

### Official Review · Reviewer_vmsJ · 2024-07-08

**Soundness:** 4
**Presentation:** 3
**Contribution:** 4
**Rating:** 8
**Confidence:** 4

**Summary:**

This paper introduces a new logical characterization for recursive Graph Neural Networks (GNNs) following the aggregation-combine or message-passing paradigm. Among other topics, it primarily focuses on understanding the differences between GNNs in the typically considered theoretical setting—where an arithmetic with unlimited representation sizes (i.e., all reals are representable) is assumed—and practical settings, where GNNs operate using floating-point arithmetic.

In particular, the paper demonstrates that:
- GNNs working with floats, even those with a “simple” aggregation-combine structure, are equally expressive as the graded modal substitution calculus (GMSC).
- GNNs working with reals are equally expressive as graded modal logic with infinite disjunctions.
- Relative to queries expressible by monadic second-order logic (MSO), GNNs working with reals and those working with floats are equally expressive.

As a byproduct, the arguments employ automata-theoretical tools, providing a concise characterization of GNNs through the concept of counting message-passing automata (CMPA).

**Strengths:**

This work aligns with a current trend in characterizing the expressiveness of GNNs from various perspectives, specifically through logical frameworks. This direction of research is by now well-established, making this work highly relevant.

The quality of the paper is exceptionally high, particularly in the formal sections dealing with logics, automata, and their theoretical intricacies. Every statement, including lemmas, propositions, and theorems, is precisely articulated, leaving no room for doubt or misinterpretation.

The presented characterizations are novel and intriguing within the context of related work on logical characterizations, making this a valuable and noteworthy contribution to the field.

**Weaknesses:**

I have a single, but rather general, weakness to point out:

- The accessibility of the topic could be improved. The paper expects readers to be very familiar with certain theoretical topics, particularly logics, which I assume are non-standard for most readers. Fortunately, I have a sufficient understanding of these subjects, but I believe that readers less familiar with them will find it challenging to grasp the key ideas of this work. This is not to suggest that the paper is misplaced; I firmly believe there is room for such specialized work in this venue. However, the main part of the paper could benefit from more examples, illustrations, or intuitive explanations, rather than a heavy focus on preliminaries. These extensive preliminaries could also be shortened and major parts be placed in the appendix, to make room for other stuff.

**Questions:**

- If I am not mistaken, in your definition of floating-point arithmetic, you only describe how to handle rounding situations, but what about overflows? As far as I understand, you assume a saturating scenario, correct? If so, could you provide a brief comment on whether handling overflow with wrap-around would make a difference?

**Limitations:**

The clearly stated results of this paper also imply their clear limitations. Additionally, the authors added open questions, which I count as a valueable statement for limitations as well.

---

> ### Author Rebuttal · Authors · 2024-08-06
>
> Concerning the weakness mentioned, we will make the preliminaries somewhat lighter---possibly moving some non-essential parts to the appendix---and instead add more examples. For example, we will include a GMSC program for reachability in order to complement the corresponding GNN example. We will also add more illustrative remarks. However, there is probably a limit to what we can achieve, as we still would not like to compromise rigor in relation to the definitions. But some illustrations and examples can surely be added, and excess details removed.
>
> Concerning the question about floats, we indeed use saturating floating-point systems. We will point this out. The wrap-around case is different: using that approach, Proposition 2.3 would not hold and thus a GNN with floating-point numbers would not be bounded.

---

> > ### Comment · Reviewer_vmsJ · 2024-08-08
> >
> > Thank you very much for the clarification and your agreement on the addition of illustrative examples and so on.
> >
> > I stand with my initial review and strongly recommend this paper to be accepted.

---

### Official Review · Reviewer_3ziF · 2024-07-12

**Soundness:** 4
**Presentation:** 4
**Contribution:** 3
**Rating:** 8
**Confidence:** 5

**Summary:**

The authors analyze the expressive power of recurrent graph neural networks (GNNs) through the lens of logic, focusing on uniform expressibility, which refers to functions expressible over all input graphs. Their findings are as follows:
- Expressive Equivalence: GNNs with floating-point numbers and the graded modal substitution calculus (GMSC) exhibit equal expressive power. The GMSC, a concept recently introduced in the context of circuits and distributed computation, is shown to be equivalent in expressiveness to simple GNNs—those that employ truncated rely and summation as aggregation functions.
- Reals vs. Floats: When floats are replaced with real numbers, an infinitary logic is required.
- MSO Definable Properties: For properties definable by monadic second-order logic (MSO), the distinction between reals and floats becomes irrelevant. Consequently, when considering MSO properties, recurrent GNNs over reals are equivalent to the finitary GMSC logic.

**Strengths:**

**S1 Relevant Research Question:** The investigation into the expressive power of recurrent GNNs is pertinent and interesting, despite their less widespread use compared to non-recurrent GNNs.

**S2 Precise Characterization:** The results offer a robust and precise characterization of the power of recurrent GNNs, using either floats or reals, from a logical perspective. These findings surpass the seminal work by Barceló et al., achieving results without assuming a background logic. When MSO is considered, the results extend those of Pfluger et al., and they also allow for the recovery of the first-order logic case as discussed by Barceló et al.

**S3 Well-Written and Interesting Techniques:** The paper is well-written, and the proofs involve intriguing techniques, yielding non-trivial results.

**Weaknesses:**

**W1 Related Work:** While the authors cite Grohe [12], Benedikt et al. [6], and Pfluger et al. [23], the relationship to Grohe’s work is not well-elaborated. A more detailed description of related work would enhance the context.

**W2 Type of Recurrent GNN:** The initialization function π\piπ can be arbitrarily chosen, such as an indicator vector for the initial vertex in reachability. This approach appears non-standard, as recurrent GNNs are not typically used once per vertex. Clarification is needed on how the analysis would change if the initialization were fixed (e.g., an all-ones vector). The use of accepting feature vectors also seems tailored to achieve desired results. Furthermore, the relationship to recurrent GNNs, interpreted as a fixed point equation, should be elucidated.

**W3 Distributed Automata:** The paper’s introduction mentions characterizing GNNs in terms of distributed automata, but this aspect is unclear in the main text. Making this contribution more explicit would benefit readers from the machine learning community.

**Questions:**

- Please comment on **W1**, **W2** and **W3**.

- Please comment on **Proof Sketch of Theorem 3.2**: In the proof sketch  you seem to assume/define a specific floating-point system? However,  the results are applicable to arbitrary floating-point systems, right? Could you clarify this point?

**Limitations:**

This has been addressed in a satisfactory way by the authors.

---

> ### Author Rebuttal · Authors · 2024-08-06
>
> Concerning **W1**: Grohe's setup is different from ours in both [11] and [12]; most notably, Grohe uses non-recurrent GNNs rather than recurrent ones. Also, the articles use reals and dyadic rationals. We will add further discussion on Grohe's work as well as on [6] and [23].
>
> Concerning the first part of **W2**, our approach is equivalent to that used in related work on the topic, in particular that of Barceló et al. [5], where the labeling of the nodes with proposition symbols is itself viewed as the initial feature vector in the GNN computation. It is easy to simulate the initialization of [5] in our setting and vice versa. The main difference is that we have explicitly included a separate initialization function (based on the propositions true at a node) to the definition of GNN.
>
> We suppose that the initialization being fixed (e.g., an all-ones vector) means that we run GNNs in graphs without node labels. In that setting, our results remain true as this scenario is the special case of our approach where the propositional vocabulary is empty.
>
> Concerning the second part of **W2**: There are different kinds of semantics for recurrent GNNs in the literature, including ones with different kinds of fixed points. For example, the pioneering GNN paper [26] uses global fixed points; a global fixed point means that a computation step of the GNN would not change the feature vector of any node in the graph. On the other hand, the RecGNN semantics of [23] uses local fixed points, that is, only the node in question must ultimately keep repeating within an appointed set of feature vectors. This is less restrictive than the case with global fixed points. The exact relation between different recurrent GNN formalisms depends on many parameters including whether reals or floats are used, which aggregate-combine functions are admitted, and whether global readouts are available. We will add some first observations to the paper, but leave a thorough investigation for future work. Please note that the issue of investigating only one acceptance condition is also discussed in the limitations section of our paper, so it was in that sense not in the primary scope of the current submission.
>
> Also, as pointed out in the conclusion, we obtain a counterpart of Theorem 3.2 for floating-point GNNs with the same accepting condition as in [23] by modifying the acceptance condition of GMSC accordingly. Furthermore, it is interesting to note that the mu-fragment of graded mu-calculus translates into GMSC (cf. the response to all reviewers), bringing its least-fixed point acceptance to the picture.
>
> Concerning **W3**, we agree that the connection between distributed automata and GNNs should be made more clear, which we will do. Our main result on this is Proposition 3.1, which links finite-state distributed automata to GMSC, and GMSC is further linked to floating-point GNNs via Theorem 3.2. Also, Theorem 3.3 links unrestricted distributed automata to omega-GML, and omega-GML is also further linked to GNNs with reals via Theorem 3.4. Please note also that with these equivalences, Theorem 4.3 gives various links between distributed automata and GNNs. Concerning Proposition 3.1, it states expressive equivalence between GMSC and the distributed algorithm class MB (cf. [15]) when based on finite-state distributed automata on directed, node-labeled graphs. Intuitively, an algorithm in MB reads multisets of a node's neighbours' messages and thereby determines its next state, also taking into account its own previous state.
>
> Regarding the question about the **Proof Sketch of Theorem 3.2**, please note that the choice of the floating-point system used in the GNN depends on the constants k that occur in the counting modalities of the GMSC program. So the resulting GNNs are actually equipped with different floating-point systems, depending on the GMSC program that they encode. Please note that fixing a single floating-point system would in some sense trivialize the computing model, because only a finite number of functions could be defined. It is indeed good to mention this more explicitly in the paper, and we shall do so.

---

> > ### Comment · Reviewer_3ziF · 2024-08-08
> > **Thanks for the clarifications!**
> >
> > I have the read the rebuttal. The authors address all my comments/questions in a very satisfactory. This is really a **very nice** paper and I will boost my score since it will be an **excellent addition to the conference program**.  And yes, I would really appreciate it if the authors expand the paper (or possible online version) with the promised additional explanations mentioned in the rebuttal.

---

### Author Rebuttal · Authors · 2024-08-06

We thank the reviewers for the reviews, all of which help clarify the paper and summarize it.

We point out the following fact concerning our responses to three of the reviews: It follows from the results in [21] (by essentially the same argument as the one justifying Proposition 7 of that article) that the mu-fragment of the graded modal mu-calculus translates into GMSC. The mu-fragment is the one with only least fixed-point operators and no greatest fixed-point ones (when presented in negation normal form).

---

### Decision · Program_Chairs · 2024-09-25

**Decision:**

Accept (poster)

**Comment:**

All reviewers agree that the submission made a significant contribution on characterizing the expressiveness of GNNs from various perspectives, specifically through logical frameworks. Although the work is purely theoretical, the presentation is good enough to make the submission comprehensible by audients having some logical basis. Thus a spotlight acceptance is recommended.